# *Drosophila* motor neuron boutons remodel through membrane blebbing coupled with muscle contraction

Andreia R. Fernandes[1], João P. Martins[1,2], Edgar R. Gomes [2], César S. Mendes[1] & Rita O. Teodoro [1]✉

Wired neurons form new presynaptic boutons in response to increased synaptic activity, however the mechanism(s) by which this occurs remains uncertain. *Drosophila* motor neurons (MNs) have clearly discernible boutons that display robust structural plasticity, being therefore an ideal system in which to study activity-dependent bouton genesis. Here, we show that in response to depolarization and in resting conditions, MNs form new boutons by membrane blebbing, a pressure-driven mechanism that occurs in 3-D cell migration, but to our knowledge not previously described to occur in neurons. Accordingly, F-actin is decreased in boutons during outgrowth, and non-muscle myosin-II is dynamically recruited to newly formed boutons. Furthermore, muscle contraction plays a mechanical role, which we hypothesize promotes bouton addition by increasing MN confinement. Overall, we identified a mechanism by which established circuits form new boutons allowing their structural expansion and plasticity, using trans-synaptic physical forces as the main driving force.

The establishment of synapses enables the wiring and communication of neurons into circuits, that when activated allow processes like perception, learning, memory, or locomotion. Synapses assemble within synaptic boutons, which are conserved round specializations of axonal membrane[1]. Boutons are plastic and can undergo rearrangements in size and number, therefore contributing to adjustments in neuronal output and/or synaptic rewiring[1,2]. Hence, it is not surprising that a link has been made between presynaptic-malfunction and neurodevelopmental or neurodegenerative disease etiology[3,4]. Despite this, the understanding of the mechanism(s) of bouton formation in wired neurons is far from complete. During embryogenesis, neurons extend highly dynamic structures, called growth cones (GCs), comprised of filopodia and lamellipodia, to navigate towards their targets, where upon arrival differentiate into round boutons[5,6]. After this period, neurons increase bouton number in response to developmental cues and to external stimuli like synaptic activity[7–9],

but the mechanism by which boutons form during these later stages remains puzzling.

The *Drosophila melanogaster* larval neuromuscular junction (NMJ) —a synapse formed between motor neurons (MNs) and skeletal muscle fibers, critical for the control of muscle contraction, has been widely used as an in vivo model in which to study the machinery that regulates bouton formation[10,11]. During larval growth, synaptic boutons are added to the neuronal arbor through developmental and activity-dependent processes, although it is not entirely understood whether these processes are regulated by common mechanisms. By mimicking elevated activity, protocols based on patterned high-K$^+$ depolarization[12–15] have successfully been used to induce the acute plastic changes at the NMJ, including formation of new boutons, called ghost boutons (GBs). GBs are immature structures that lack post-synaptic machinery despite having some presynaptic markers (including synaptic vesicles, SVs, and occasionally active zones, AZs).

[1]iNOVA4Health, NOVA Medical School|Faculdade de Ciências Médicas, NMS|FCM, Universidade Nova de Lisboa, Lisboa, Portugal. [2]Instituto de Medicina Molecular João Lobo Antunes, Faculdade de Medicina da Universidade de Lisboa, Avenida Professor Egas Moniz, 1649-028 Lisboa, Portugal. ✉e-mail: rita.teodoro@nms.unl.pt

After being formed, GBs can either mature[12] or, if they fail to stabilize, be retracted or degraded via muscle- and glia-mediated mechanisms[16]. Many studies in *Drosophila* and in mammals have used increased extracellular $[K^+]$ to depolarize neurons with the aim of increasing neuronal activity[17–21]. However, perhaps due to the small size of boutons and limitations of optical microscopy, little is known about activity-dependent bouton formation in mammalian systems. In contrast, *Drosophila* NMJs allow clear identification of single boutons, providing a good model for such studies[10,11], which resulted in the identification of several conserved signaling pathways important for activity-dependent bouton outgrowth, including Wnt-, BMP-, and synapsin/PKA-dependent pathways[13–15,22]. Additionally, live imaging of NMJs has shown that in response to elevated activity new boutons form within minutes, mostly during muscle contraction, and emerge by budding off the membrane from the presynaptic arbor[14], distinct from the embryonic GC. Also, there is evidence that this type of bouton growth is coupled to synaptic transmission, as it is associated with actin dynamics[14,23] and requires proteins that regulate the SV machinery[13,15]. Moreover, recent studies have indicated that NMJ plasticity is regulated by trans-synaptic signaling and requires factors involved in physical interactions between motor neurons (MNs), muscle and glial cells[11–14,24–26]. However, few of these have addressed the mechanisms used by neurons to (1) coordinate activity with the cytoskeleton to mediate the morphologic changes that initiate bouton addition and (2) synchronize these changes with the intercellular microenvironment.

In vivo, neurons develop in complex 3-D environments where they are subjected not only to chemical but also to mechanical signals, such as the stiffness of the environment and compression or traction forces exerted on neurons by the surrounding cells[27]. For example, the stiffness of the extracellular environment has been shown to influence the movement and complexity of neurites and astrocytic processes[28]. Also, throughout their lifetime neurons are exposed to heterogeneous stiffness and several mechanical cues, which, in addition to modulating neuronal differentiation and development, likely influence the plasticity of neuronal circuits. Indeed, mechanical force is emerging as a key factor for the regulation of axon guidance, growth, synapse formation, and plasticity[27–30]. Several studies have suggested that synaptic changes driven by activity involve dynamic cell interactions, which are mediated by molecules coupled to the cytoskeleton and governed by trans-synaptic forces[25,31–35]. Altogether, there is accumulating evidence supporting that neuronal structure and function depend on mechanical interfaces between neurons and their 3-D microenvironments, but the general mechanisms by which force acts remain largely unknown.

Here, using live imaging of NMJ and *Drosophila* genetics, we found that, activity-dependent bouton addition occurs quickly by membrane blebbing, a pressure-driven mechanism well characterized in 3-D cell migration, but to our knowledge, never reported to occur in neurons. Moreover, we showed that this mechanism of acute bouton formation requires muscle contraction around the MN membrane, indicating that mechanical forces may cooperate with biochemical signals to coordinate activity-dependent bouton formation in vivo.

## Results

### Bouton formation in response to acutely increased activity and at rest is fast and mechanically resembles blebbing migration

To study activity-dependent bouton formation, we used *Drosophila* 3rd instar larvae NMJs, where developmental bouton addition is nearly complete and well-established patterned high-K+ depolarization is known to induce fast bouton formation[12,13]. New boutons induced by this form of acute stimulation, classically called ghost boutons (GBs)[12], are thought to represent immature or precursors of functional boutons[15,36]. To dissect how GBs are added to wired neurons, we imaged every few seconds NMJs of live semi-dissected larvae after induction of acute structural plasticity with high-K+ (Supplementary

Fig. 1). We did not anesthetize the larvae during imaging based on early observations that boutons are formed during larval contraction[14]. We expressed UAS-CD4-Tomato (CD4-Tom) in all neurons under the control of NSyb-Gal4 to label the presynaptic membrane and observed that membrane dynamics accompanying the formation of new boutons was clearly distinct from those of embryonic GC. As reported[14], acute stimulation by high-K+ leads to the quick formation of GBs during the stimulation pulses. However, boutons continue to form after the stimulation period, which allowed us to identify by live imaging two main types of bouton appearances (Fig. 1). First, bursting growth characterized by very fast membrane expansion was frequently associated with clear muscle contractions (Fig. 1a and Supplementary Movie 1); and second, gradual extension sustained by a slow membrane flow, seen with more subtle muscle movements (Fig. 1b and Supplementary Movie 2). Analysis of the movies revealed that boutons developed primarily from preexisting boutons, which split or elongate, and rarely extent from axons (Supplementary Fig. 2a, b). Importantly, we never observed boutons emerging from lamellipodia and/or filopodia precursors. Instead, boutons formed as large spherical expansions of the MN membrane, and their growth was often rapid, ranging from less than 1 min to a few minutes (Fig. 1c; median: 5 min 38 s). There was a wide variation in the formation time, probably representing two (or more) modes of bouton addition, which we did not separate because the n was not sufficiently high to divide in clusters. In addition, on several occasions, a bright membrane punctum was visible at the base of newly formed boutons, suggesting local intracellular vesicle dynamics. To assure that the transmembrane tag used, CD4-Tom, did not interfere with bouton formation, we compared this genotype with wild-type (*w1118*) and neuronal expression of cytosolic LexAop-mCherry under the control of DVGlut-LexA[37] (Supplementary Fig. 2c, d) and showed that both GB frequency and area after stimulation was identical (Supplementary Fig. 2c, d, g, h). Interestingly, the time of bouton formation was slightly increased in neurons expressing CD4-Tom (Supplementary Fig. 2e, f) suggesting that CD4-Tom, an external tag, may delay bouton growth. Therefore, we may be overestimating the time, as boutons may form even faster.

Given that high-K+ stimulation could be considered an extreme method to induce neuronal depolarization[38,39], we also used optogenetics as an alternative means to visualize bouton outgrowth (Supplementary Fig. 3 and Supplementary Movies 3 and 4). Using optogenetic stimulation of larval MNs we observed the formation of GBs displaying the same morphological features as those formed by high-K+ (Supplementary Fig. 3a, b). Moreover, when we imaged larval preparations at rest, without applying any stimulation method, we also observed bouton formation with the same characteristics as with high-K+ (Fig. 1d and Supplementary Movie 5). To further examine GBs during normal larval NMJ development, we imaged fixed larvae from the 2nd through 3rd instar stages and observed that these immature boutons lacking Dlg were naturally present in vivo, despite in low numbers (Supplementary Fig. 4). Although non-stimulated animals would represent the most physiological scenario to study the mechanisms of bouton formation systematically, the frequency at which boutons are added to the NMJ is much lower when compared with high-K+ stimulation, making it difficult to use this paradigm for our studies. Overall, GBs formed in naturally growing larvae or by external stimulation appear identical with both displaying less differentiation in the synaptic transmission machinery, compared to mature boutons.

Previous studies have shown that GBs in intact undissected larvae have maturation potential[12,15], whereas other GBs have been shown to be eliminated by Draper-dependent phagocytosis by both glia and muscles[16]. Therefore, GBs are generally considered immature precursors of synaptic boutons. However, the fate of GBs formed by external stimulation with high-K+ has not been investigated. Thus, we analyzed whether boutons induced by this method of depolarization could mature ex vivo, as it has been suggested to occur in vivo[12],

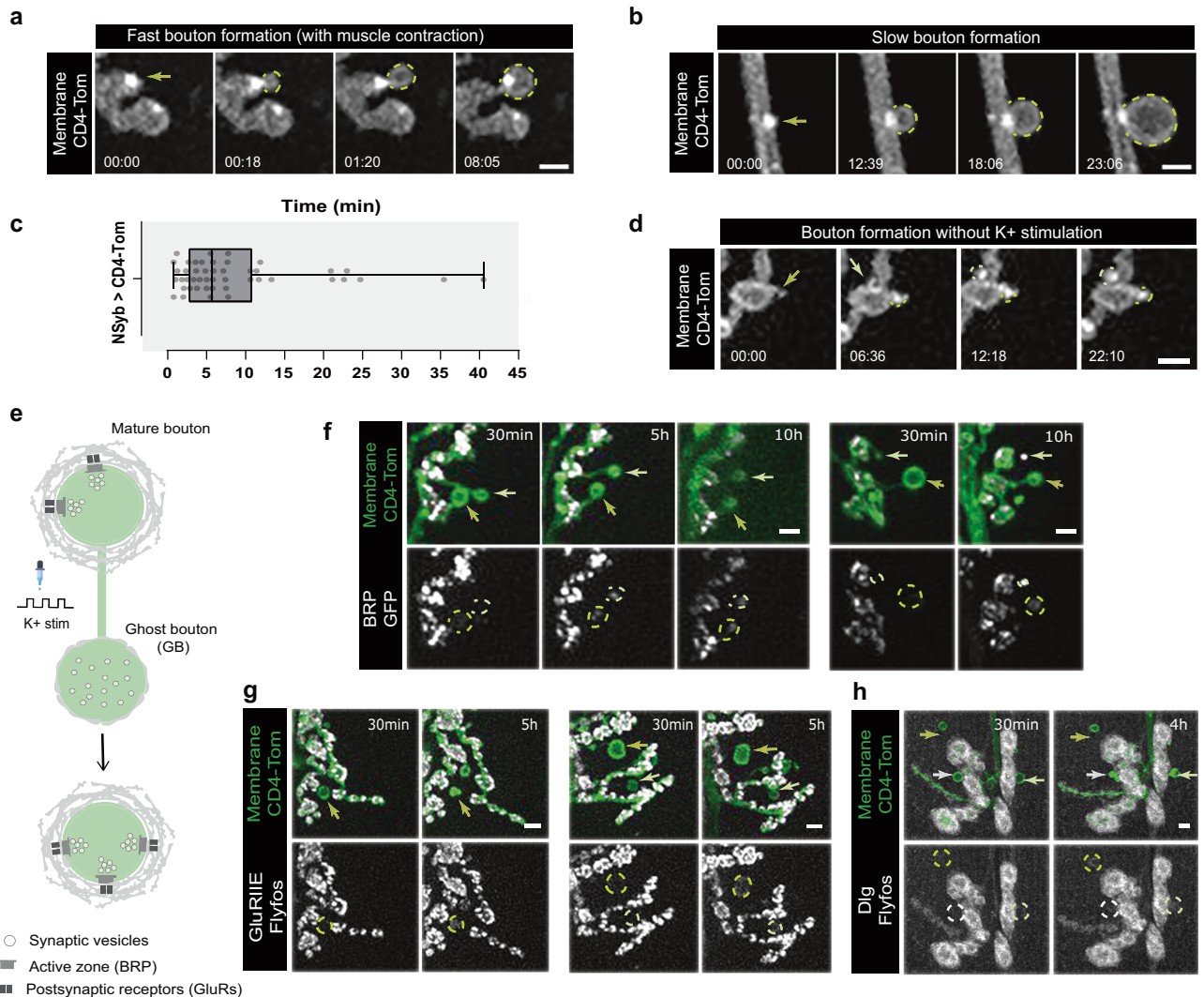

**Fig. 1 | Ghost boutons form by acute induction of activity and at rest, having maturation potential ex vivo. a**, **b** High temporal time-lapse images of bouton formation after high-K⁺ stimulation. Neuronal membranes were labeled with UAS-CD4-Tomato (CD4-Tom) under the control of NSyb-Gal4 (pan-neuronal driver). Scale bar is 2 μm. Arrow indicates where the bouton emerges. **a** Examples of fast bouton formation was frequently observed, with noticeable muscle contractions. **b** Example of slow bouton formation. **c** Boxplot (min to max) of the time of bouton formation across experiments. Boxplots show 25th–75th percentiles, the median, and whiskers from minimum to maximum. Median is 5 min 38 s. All the data points are presented. $n = 13$ larvae and 46 boutons. **d** Representative example of a time-lapse of bouton formation without stimulation (at rest), in HL3.1 with 0.5 mM Ca²⁺.

Scale bar is 2 μm. Arrow indicates where the bouton emerges. **e** Schematic of activity-induced bouton formation and subsequent maturation progression. Representative of $n = 10$ NMJs, 7 larvae. **f**, **g**, **h** Representative time-lapse images showing ghost bouton (GB) maturation (in Schneider's medium) by acquisition of the presynaptic active zone scaffold protein Bruchpilot (BRP, f) and postsynaptic markers such as Glutamate receptor IIE (GluRIIE, g) and Discs-large (Dlg, h) enriched at postsynaptic membrane specializations surrounding mature boutons – Subsynaptic reticulum (SSR) – 5-10 hours post high-K⁺ stimulation. Scale bar is 2 μm. Arrows indicate boutons undergoing maturation. **f** $n = 37$ NMJs, 6 larvae; **g** $n = 41$ NMJs, 6 larvae; **h** $n = 105$ NMJs, 23 larvae. Source data provided as a Source Data file.

following their fate by time-lapse imaging. We expressed UAS-CD4-Tomato in neurons using NSyb-Gal4 together with UAS-BRP-GFP, or with protein traps Dlg-GFP or GluRIIE-GFP and imaged 30 min after stimulation through 6–10 h post stimulation. We observed ex vivo that GBs can acquire synaptic markers (Fig. 1e–h) like the presynaptic active zone protein BRP and postsynaptic density proteins Dlg (scaffold) and GluRIIE (core Glutamate receptor subunit). Maintenance of preparation viability for extended periods was very variable, preventing a quantitative analysis of the frequency of GB maturation. In sum, we provide evidence that GBs are physiological structures present throughout larval development, whose formation can be acutely induced by external stimulation and that have the capacity to mature, supporting the hypothesis that GBs represent at least a fraction of precursors of synaptic boutons.

Because of the morphological resemblance between bouton formation and yeast budding, an analogous mechanism has been proposed to explain activity-dependent bouton formation at the *Drosophila* NMJ[14]. However, budding is a slow process that takes hours to occur and can be as short as 1.5 h under optimal conditions[40]. Our data suggested that bouton formation is a faster process that strongly resembles cellular blebbing. Blebs are round membrane protrusions extruded by intracellular hydrostatic pressure that some cells use, in combination with or as an alternative to lamellipodia, to migrate in 3-D environments[41,42]. Blebbing has been shown to occur in migratory cells during development, tissue homeostasis, immune surveillance, and pathological conditions, such as cancer metastasis and inflammation[43]; however, to the best of our knowledge, it has never been reported to occur in neurons. Blebbing is in a large extent a physical, rather than a

biochemical process, occurring over time- and length-scales that have been seldom investigated in animal cells and in vivo[43–46]. A characteristic feature of blebs is the rapid change in cell shape at the site where the protrusion emerges[45].

Since our movies revealed rapid dynamics and morphological features reminiscent of membrane blebbing, we sought to test whether *Drosophila* MNs adapted this mechanism to remodel under conditions of intense neuronal activity. A large difference between blebs and other membrane protrusions is that bleb growth is not dependent on actin polymerization, but is rather a pressure-driven mechanism where cytosolic fluid flow pushes the membrane through a local weakening (rupture or detachment) in the cell actomyosin cortex[41]. Therefore, the main hallmark of blebs is that they have reduced actin when they first form, which can subsequently assemble to reform the actin cytoskeleton to stabilize or retract[45,47]. Nonmotile cell blebs undergo fast expansion-retraction cycles (Fig. 2a), while migrating cells rarely retract and usually form persistent and even sequential blebs that eventually become stabilized[48]. To test whether boutons formed with low actin, we expressed Lifeact-Ruby under the control of distinct Gal4 lines, with different strengths, to visualize F-actin, and observed that the frequency of GB formation after stimulation remained unchanged (Supplementary Fig. 5a–d). Since blebs have low actin levels when they form, we chose the strongest line, NSyb-Gal4, to ensure that if actin was not observed in the newly formed boutons, it was not due to low expression levels. Expression of CD4-GFP and Lifeact-Ruby in neurons did not alter GB formation frequency after stimulation (Supplementary Fig. 5e, f), and was therefore used to follow membrane and F-actin dynamics after high-K$^+$ stimulation and in unstimulated rest conditions (Fig. 2b–g and Supplementary Figs. 6–8 and Supplementary Movies 6–10). Real time imaging showed that the new boutons have membrane growth without prominent F-actin, in both stimulated and unstimulated conditions, a hallmark of blebs. Furthermore, we observed bouton formation with (Fig. 2b and Supplementary Figs. 6a, b and 7) and without retraction (Fig. 2c, f and Supplementary Figs. 6c and 8a) identical to the bleb types previously described in non-neuronal cells[43].

We quantified actin content in the new boutons comparing actin levels at the base and at the edge of these boutons, in the beginning and after their expansion (Fig. 2h, i and Supplementary Fig. 9a) and showed that actin content in the edge of growing boutons was lower than base (92% boutons at beginning, median −43.5%; 83% boutons after expansion, median −31%). Furthermore, the actin distribution at bases or edges between the beginning and expansion suggested that low amounts of actin were recruited during the outgrowth of new boutons (Fig. 2j and Supplementary Fig. 9). Overall, actin intensity appears very low when boutons are first formed and remains low throughout the expansion process, analogous to bleb formation. We also analyzed the dynamics of bouton formation and its relationship with actin (Supplementary Fig. 9) and showed that both new bouton size and time of bouton formation were independent of the actin content. Our data showed various scenarios related to actin content, but we never observed accumulation of actin at the edge of actively growing boutons.

To support the live data, we analyzed fixed larvae expressing Lifeact-GFP under the control of NSyb-Gal4, which confirmed that GBs lack actin (Supplementary Fig. 10a, c). In our movies we observed F-actin puncta at the base of new boutons in 26% of events (Fig. 2c and Supplementary Fig. 9l), which were possibly related with some actin flow (Supplementary Fig. 9m, n). This finding was consistent with a previous report showing that spots of GFP-Moesin are associated with sites of bouton growth[14]. Interestingly, our analysis indicated that these actin puncta occurred preferentially in cases where threads of boutons formed, rather than with single events (Supplementary Fig. 9p, q), suggesting that local actin rearrangements may occur at the place where the bouton emerges to sustain rapid and successive

bouton formation, as it happens in cells forming persistent or sequential blebs[49,50]. On the other hand, pre-synaptic actin puncta have also been associated with places of active endocytosis[51]; therefore it is possible that these puncta represent endocytic regions. Interestingly, in addition to more canonical functions, endocytosis has also been shown to regulate the localization of factors required for the control of membrane blebbing[52]. Also, a recent study reported three types of presynaptic actin structures: (1) a branched actin mesh at the active zone that regulates access to the plasma membrane for exocytosis; (2) linear actin rails which help vesicles move between the exocytosis/endocytosis zones and intracellular pools; and (3) peri-synaptic actin corrals that function as scaffolds for the vesicular pool and could also help at endocytic zones[53]. Altogether, we suggest that these actin puncta are likely related with local membrane trafficking events that can contribute to the formation of boutons, but how and whether they represent endo- or exocytic events requires further investigation. Also supporting that boutons form with a weaker actin structure, identical to membrane blebs[48,54], we observed that GBs were depleted of filamin (Supplementary Fig. 10b, d), which crosslinks the membrane and intracellular proteins to actin[55]. After bouton growth stalls, we observed a few cases of increased levels of Lifeact-Ruby within the new boutons, which is suggestive of increased actin dynamics (Supplementary Fig. 8b–e and Supplementary Movies 11 and 12). This dynamic accumulation of Lifeact-Ruby at the membrane was concomitant with a reduction in bouton size, suggesting that actin recruitment leads to bouton stabilization. Importantly, in our live and fixed imaging, bouton formation occurred without visible actin (independent of the presence of actin puncta at the base of the forming bouton), corroborating that actin polymerization is not the main driving force for bouton extension, in contrast to what occurs when other protrusions are formed[43,56]. Altogether, our data suggests that new boutons are generated by a blebbing mechanism.

## Manipulation of actin dynamics regulates GB size

The actin machinery plays a critical role in blebbing, low levels at initial phases of bleb formation and increasing as growth stalls. To test how actin dynamics impacts bouton growth, we pharmacologically manipulated actin during bouton formation (Fig. 3). We used Latrunculin B (LatB) to induce actin depolymerization and Jasplakinolide (JAS) as an actin stabilizing drug. We measured number and intensity of F-actin puncta observed in boutons attached to the main axonal arbor, to monitor the effect of the drugs on actin dynamics. We observed that application of LatB (Fig. 3a, c) to the stimulation solutions rapidly dispersed actin puncta in NMJs, while treatment with JAS (Fig. 3b, d) maintained or caused the formation of actin puncta, without changing the total intensity per area (Fig. 3e, f). Within each punctum, the intensity of actin was reduced with LatB and increased in the presence of JAS (Fig. 3g, h), again suggesting that actin dynamics is altered towards a more depolymerized or polymerized state, respectively. To further analyze the efficacy of JAS in inducing actin polymer stabilization we performed Fluorescence Recovery After Photobleaching (FRAP). We showed that Lifeact-Ruby and GFP-labelled monomeric Actin5C, which can be incorporated into filamentous structures, displayed similar recovery dynamics[57] after photobleaching, in neurons expressing both actin probes (using NSyb-Gal4) (Supplementary Fig. 11 and Supplementary Movie 13), validating Lifeact use in FRAP experiments as a readout for F-actin stability, analogous to previous studies[58]. Using a Lifeact reporter, we showed that JAS treatment increased the half-time of recovery compared with control conditions (HL3.1 and DMSO vehicle) (Supplementary Fig. 12 and Supplementary Movie 14), and decreased total fluorescence recovery and mobile fractions (Supplementary Fig. 12) supporting JAS effect on F-actin stabilization.

After confirming the effect of the drugs on actin polymerization and stabilization, we treated MNs with LatB and observed that favoring actin depolymerization did not change GB frequency after high-K$^+$

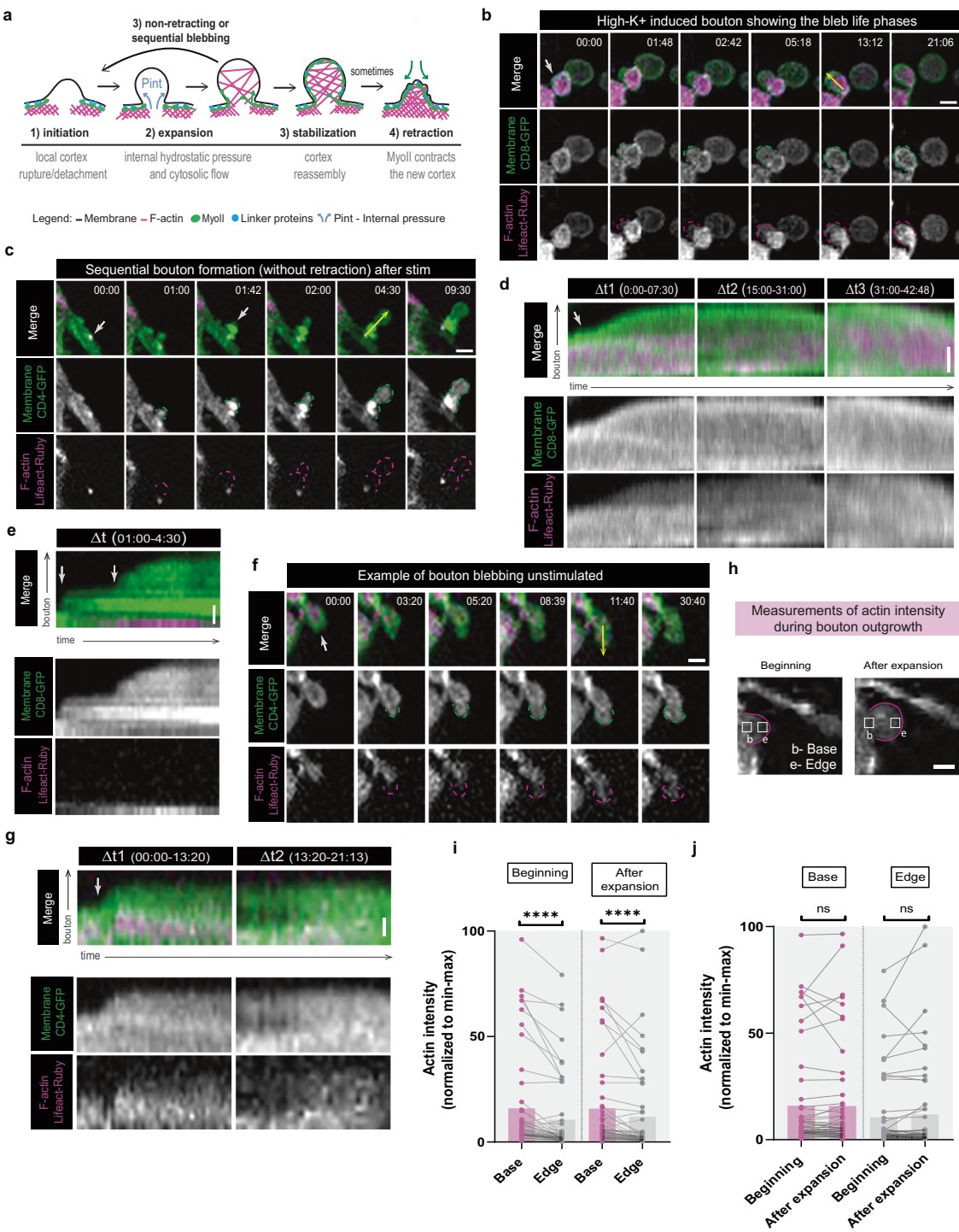

stimulation, compared to HL3.1 and DMSO controls (Fig. 3i, j and Supplementary Fig. 13a) contrary to what was previously observed for the same concentration of Latrunculin A, a more potent inhibitor of actin polymerization[14]. However, GB size was increased (Fig. 3k and Supplementary Fig. 13c). This result suggests that mild disruption of actin polymerization is not sufficient to impair acute structural plasticity, possibly because actin is not essential during the initial stages of bouton formation. Accordingly, local application of LatB to cells in culture does not change bleb frequency but leads to global increase in bleb size[54]. Additionally, in the aforementioned study, slower bleb dynamics were observed within the treated area, with retraction resuming only after the removal of LatB, driven by actin polymerization[54]. Conversely, treating MNs with JAS, revealed that GB

formation after high-K+ stimulation was also not different from controls (Fig. 3l, m and Supplementary Fig. 13b). However, we noticed a conspicuous phenotype in the NMJs that responded to the stimulation, which was the presence of very small GBs, usually clustered in discrete regions. While JAS did not change GB frequency after stimulation, we found that GB size was smaller after JAS treatment (Fig. 3n and Supplementary Fig. 13d). Our results suggest that bleb size is determined by a balance between the initial growth rate and the time needed for actin to repolymerize at the bleb membrane (Supplementary Fig. 13e), which can be changed according to the prevailing local actin dynamics. While JAS has been reported to either cause or block blebbing[59,60], one hypothesis is that in cases where the flow rate of the cytoplasm is insufficient to outpace local actin polymerization, it

**Fig. 2 | Activity-dependent bouton formation mechanistically resembles blebbing migration. a** Schematic of bleb life-cycle. Non-motile cells show: (1) initiation, (2) rapid expansion, (3) stabilization, and (4) slow retraction (a few min). Retraction is rarely observed in motile cells. **b**, **c** Time-lapse images of ghost boutons (GBs) induced by high-K⁺ stimulation showing bleb life phases. Neuronal membrane and F-actin were labeled with UAS-CD4-GFP (or CD8-GFP) and UAS-Lifeact-Ruby, both under the control of NSyb-Gal4. **b** Example of bouton formation and subsequent stabilization. **c** Example of sequential bouton formation without retraction. **d**, **e** Kymographs of (**b**) and (**c**). Kymographs for distinct $\Delta t$ values to highlight the dynamic nature of actin during bouton outgrowth. **d** Initially actin is low ($\Delta t1$) but subsequently assembles inside boutons and accumulates when bouton size stabilizes ($\Delta t2$ and $\Delta t3$). **e** actin is absent in the forming bouton, but an F-actin punctum persists at the base. **f** Time-lapse images of bouton formation under unstimulated conditions also show bleb life phases and morphological characteristics. Example of bouton formation and stabilization. **g** Kymograph from

**f** displaying actin dynamics found in boutons formed without stimulation, analogous to boutons induced by high-K⁺, $N = 7$ larvae/7NMJs. **b**–**g** White arrow indicates where bouton emerges, yellow arrow indicates ROI used for kymograph generation. **h** Time-lapse images showing the location where actin intensity was measured: squares of $1\,\mu m^2$ were positioned at the base (**b**) and edge (**e**) of the bouton in the beginning and at the end of expansion. **i**, **j** Bar plots (mean) with normalized values (min−max normalization) for F-actin intensity measured in new bouton edges and corresponding bases at the beginning or after expansion. **i** Actin levels at the edges were significantly lower than those of basal actin at both growth stages. **j** No significant actin flux occurred at the base or edge during bouton outgrowth. All data points are represented, and the lines connect the paired points. $n = 14$ larvae/53 boutons. Statistical significance determined using non-parametric Wilcox on matched-pairs signed rank test; ****$p < 0.0001$, ns is not significant. Scale bar, $2\,\mu m$ for time-lapse images (**b**, **c**, **f**, **h**) and $1\,\mu m$ for kymographs (**d**, **e**, **g**). Source data provided as a Source Data file.

could result in small or no blebs, whereas cases where actin filaments are highly stabilized would also restrict bleb formation. Supporting this hypothesis, smaller boutons frequently appeared together or in nearby regions, suggesting that they were formed during the same event. Altogether, we provide evidence that GBs show the actin cytoskeletal hallmark and dynamics (Figs. 2 and 3) compatible with bona fide blebs, which supports the hypothesis that MNs adopted blebbing as a mechanism to rapidly modulate bouton formation in response to increased activity.

### MyoII, a master regulator of blebbing, contributes to GB growth

Non-muscle myosin II (MyoII) is considered a master regulator of blebbing since, even though this process can be triggered in various ways, bleb formation is preferred under high cortical tension conditions, which is generated mainly by MyoII[41,44,54] (Supplementary Fig. 14a, b). To study MyoII distribution at the NMJ, under basal conditions and post high-K⁺ stimulation, we used an antibody against the heavy chain of MyoII Zipper (Zip) (Fig. 4a and Supplementary Fig. 14c, d) and two protein traps for MyoII, GFP-tagged Spaghetti Squash-Sqh (regulatory light chain) and Zip (heavy chain) (Fig, 4b and Supplementary Fig. 14e, f). MyoII expression pattern was identical when the antibody or protein traps where used, supporting that it represents endogenous protein localization. We observed that MyoII is localized in the MN terminal, sometimes visibly outlining the boutons, in addition to being abundant in muscle and trachea (Supplementary Fig. 14c). Noteworthy, MyoII was present in GBs induced either by stimulation (Fig. 4a,b and Supplementary Fig. 14) or in GBs present in non-stimulated NMJs (Supplementary Fig. 15), displaying noticeable accumulations at the base (mostly in the form of a strong punctum) or at the edge of the boutons (partially or completely outlining the bouton membrane cortex). This distribution is consistent with a role of MyoII in bleb initiation (possibly by contributing to the formation of a weak spot at the membrane from where the bleb will emerge), and at later stages, in stabilization/retraction phases (aiding in reforming the cortex) (schematic Fig. 4c).

To characterize MyoII dynamics during activity-dependent plasticity at the NMJ, we followed both bouton growth and MyoII dynamics using live imaging. We used larvae expressing cytosolic mCherry under the control of the neuronal LexA driver DVGlut and a GFP-tagged Sqh protein trap rescue of a Sqh-null mutant ($sqh^{AX3}$) to guarantee that all MyoII was tagged with GFP. From these movies we observed a myriad of dynamic MyoII behaviors, which we summarize here. We observed MyoII puncta at the base of the boutons preceding formation (Fig. 4d, f and Supplementary Fig. 16a and Supplementary Movie 15) and a flow of MyoII into new boutons after they started growing (Supplementary Fig. 17 and Supplementary Movie 16). We also show two examples of MyoII accumulation during bouton morphological remodeling, leading to shrinkage (Fig. 4e, g, h and Supplementary Fig. 16b, c and Supplementary Movie 17). This is in accordance with a previous study in

other cell type, suggesting that MyoII recruitment to the bleb cortex drives stabilization/retraction[48].

Analogous to actin, we quantified MyoII content in the new boutons base and edge in the beginning and after expansion. Akin to what we saw for actin, MyoII was lower at the edge of the bouton both in the beginning of the expansion and at maximum size, when expansion ceased, despite less pronounced (Fig. 4i and Supplementary Fig. 18a): 70% boutons at beginning, median −11.9%; 72% boutons after expansion, median −15,5%. Furthermore, MyoII content did not change significantly at base or edge throughout bouton outgrowth (Fig. 4j and Supplementary Fig. 18d). These data indicate that in the majority of the events MyoII was more prominent, and remained, at the base of the growing bouton in accordance to what was shown in other cells, where MyoII is recruited to the bleb cortex at later stages and later than actin[47]. However, when we analyzed bouton dynamics and its relationship to MyoII (Supplementary Fig. 18), we found that bouton formation was significantly faster in boutons formed with low MyoII levels (Supplementary Fig. 18e), compared with boutons formed with high MyoII, suggesting that MyoII recruitment to boutons can change their growth dynamics by slowing growth. Taken together, our data of fixed and live MyoII distribution after high-K⁺ stimulation, suggests that the spatio-temporal recruitment of MyoII to GBs is tightly regulated, and comparable to actin recruitment: (1) early MyoII accumulates at base to facilitate initiation and initial stages of expansion (supporting this we find MyoII puncta at the base of 35% new boutons); and (2) at later growth stages, MyoII moves towards the new cortex at the edge where it accumulates possibly during the stabilization phase (as observed in Fig. 4g, h).

To better understand the role of MyoII in GB formation, we took a knockdown (KD) approach (by RNAi), to decrease the levels of MyoII specifically in neurons. For this, we expressed different RNAi lines against both subunits of MyoII Sqh (regulatory light chain) and Zip (heavy chain) under the control of the neuronal Gal4 driver NSyb (Fig. 5 and Supplementary Fig. 19) and confirmed that Zip levels were reduced in GBs by antibody staining (Supplementary Fig. 20). Additionally, the RNAi line used in Fig. 5 has been previously validated in the *Drosophila* retina[61]. Under these conditions, MyoII-dependent contractility is reduced in MNs but not in muscle or other cells. We discovered that neuronal decrease of the light or the heavy chain of MyoII increased MNs capacity to form GBs after high-K⁺ depolarization (Fig. 5a–c and Supplementary Fig. 19c–e), rather than blocking it. Accordingly, MN expression of a validated[61] non-phosphorylatable dominant negative (DN) form of Sqh under the control of NSyb-Gal4 also increased activity-dependent bouton frequency (Supplementary Fig. 21). Overall, decrease or inactivation of MyoII in neurons shifted the distribution of GB number to increased values after stimulation, which was not seen in unstimulated NMJs, suggesting that these changes were not due to increased developmental bouton addition (Fig. 5d,e and Supplementary Fig. 21c, e). On the other hand, when we

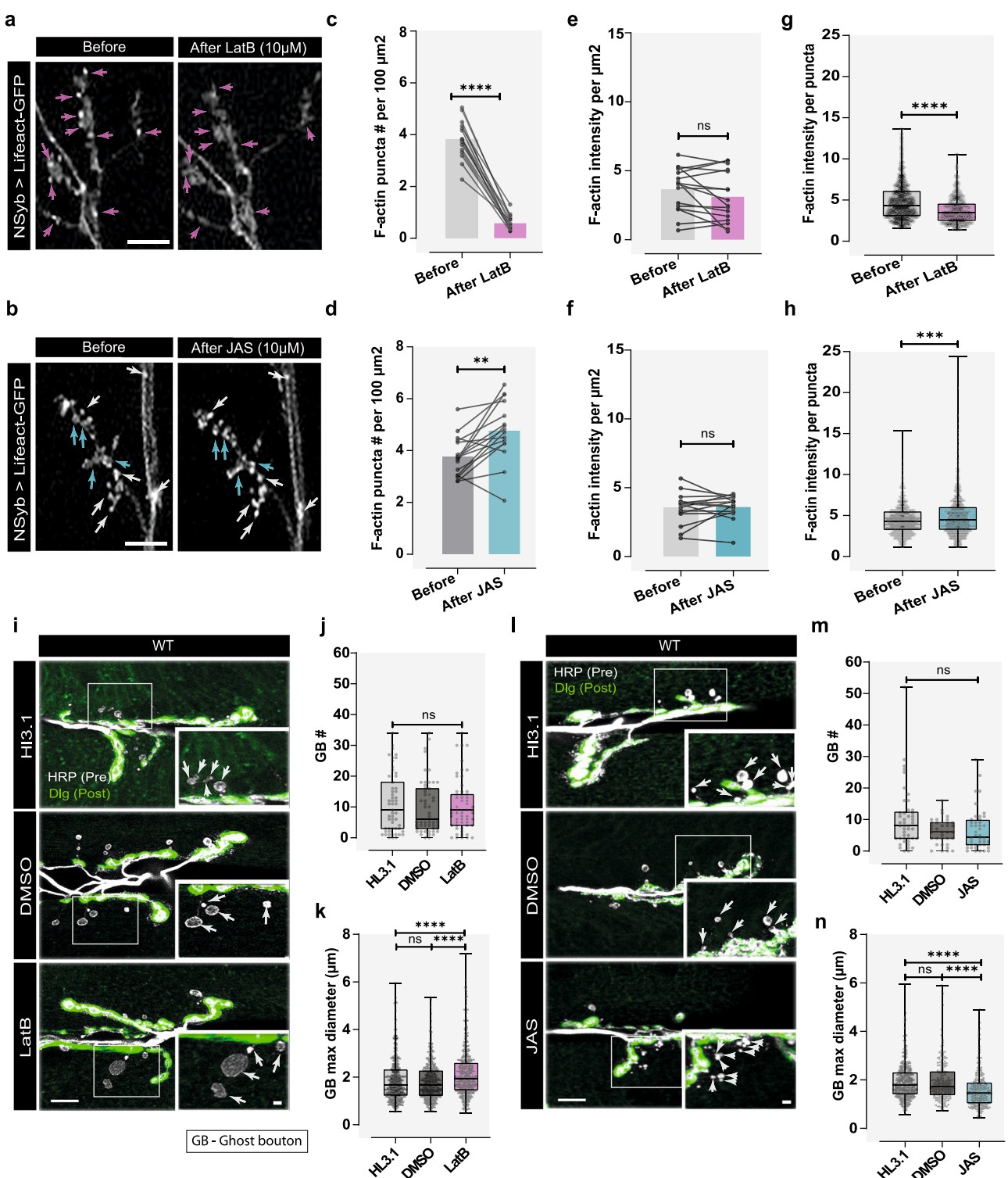

expressed a phosphomimetic constitutively active (CA) form of Sqh, we did not observe a significant difference in GB frequency after stimulation, when compared to control (Supplementary Fig. 21b, d), showing that general increases in presynaptic MyoII are not sufficient to increase the formation of GBs.

At first the results obtained after MyoII KD were unexpected because MyoII depletion or inhibition strongly reduce bleb frequency in cells in 2-D[54,62], but not necessarily in a 3-D environment (see below). To further analyze the dynamics of bouton formation when MyoII is reduced in MNs, we performed live imaging of larvae expressing both CD4-Tom and Sqh-RNAi under the control of the neuronal NSyb-Gal4

driver (Fig. 5f). We observed that MyoII KD NMJs were not only more plastic but also formed boutons faster (Fig. 5g and Supplementary Fig. 19a and Supplementary Movie 18), while bouton area was unchanged (Fig. 5h and Supplementary Fig. 19b). Interestingly, in our analysis of the dynamics of MyoII distribution during bouton formation (Supplementary Fig. 18), we observed that boutons that were formed with low MyoII at the base and edge, a scenario of low MyoII levels somewhat equivalent to Sqh-RNAi (despite being local), corresponded with the pool of boutons that formed faster (Supplementary Fig. 18g, h, j), akin to MyoII reduction using RNAi. Additionally, we detected that in control animals (NSyb-Gal4>CD4Tom) the variation

**Fig. 3 | Manipulation of actin dynamics during acute stimulation regulates GB size. a, b** Treatment of larvae with Latrunculin B (LatB) or Jasplakinolide (JAS). Images before/after drug administration. F-actin visualized by UAS-Lifeact-GFP expression under NSyb-Gal4, and actin puncta in main arbor-attached boutons were used as readout. Scale bars, 10 μm. **a** LatB application led to F-actin puncta dispersion (pink arrows). **b** After JAS treatment, F-actin puncta persisted/became brighter (white arrows); new puncta appeared in some regions (blue arrows). **c, d** Bar plots (mean) showing F-actin puncta relative frequency per 100 μm$^2$ after LatB and JAS treatment, decreased and increased, respectively ($p < 0.0001$ and $p = 0.0013$). **e, f** Bar plots (mean) showing F- actin intensity/μm$^2$ of NMJ after LatB and JAS treatment. Lines connect before and after values. **c–f** Statistical significance by parametric $t$-test (paired); ****$p < 0.0001$, **$p < 0.01$ and ns is not significant. $n = 16$NMJs, from 5(LatB) and 8(JAS) larvae. **g, h** Boxplots showing F-actin intensity/ puncta after LatB and JAS treatment, decreased and increased, respectively ($p < 0.0001$ and $p = 0.0005$). $N$ of Actin puncta LatB: Before = 848/After = 483 from $N = 16$NMJs. Actin puncta JAS: Before = 865/After = 1032 from $N = 16$NMJs. Statistical significance by Mann–Whitney test (two-tailed). **i** Images of WT animals ($w^{1118}$)

treated with HL3.1, DMSO, or 10 μM LatB after high-K$^+$ stimulation. **j** Boxplot showing ghost bouton (GB) number and **k**, area, after high-K$^+$ stimulation. HL3 $n = 59$NMJs/779 boutons, DMSO $n = 56$NMJs/629 boutons and LatB $n = 55$ NMJs/779 boutons. Statistical significance determined by non-parametric Kruskal–Wallis test (two-tailed); ****$p < 0.0001$, ns is non-significant. **l** Images of WT animals ($w^{1118}$) treated with HL3.1, DMSO, or 10 μM JAS after high-K$^+$ stimulation. **i, l** Scale bar, 10 μm and 2 μm (inset), presynaptic and postsynaptic membranes labeled with antibodies against horseradish peroxidase (HRP-gray) and Discs-large (Dlg-green), respectively. GBs (white arrow) zoom corresponding to squares. **m** Boxplot (min to max), data showed that GB number was unaffected by JAS treatment. **n** Boxplot (min to max), data showed decreased GB maximum diameter with JAS treatment. HL3 $n = 44$NMJs/471 boutons, DMSO $n = 35$NMJs/214 boutons and JAS $n = 46$ NMJs/ 307 boutons. Statistical significance determined by non-parametric Kruskal–Wallis test (two-tailed); ****$p < 0.0001$, ns is non-significant. Boxplots show 25$^{th}$–75$^{th}$ percentiles, lines at the median, and whiskers from minimum to maximum. Source data provided as a Source Data file.

observed in the formation times of boutons was only partially explained by their size (Fig. 5i), suggesting that other elements may regulate bouton expansion dynamics at the NMJ. In addition, we observed a clear dynamic shift in MyoII KD larvae (Fig. 5j), where the linear relationship between area and time of formation was almost lost, possibly because another factor had a larger effect on growth dynamics. Remarkably, in MyoII KD NMJs we only observed bouton formation associated with evident muscle contractions (Supplementary Movie 18), contrary to what was observed in the controls (Fig. 1a, b). Hence, we hypothesized that the increase in GB formation after stimulation observed in MyoII neuronal KD is dependent on muscle activity.

## Muscle contraction regulation of MN confinement is necessary for acute structural plasticity

It has been shown that WT cells increase bleb frequency, and even myosin-null cells recover competence to bleb, when compressed between two agar layers[63], and that blebs formed by cells in highly confined 3-D environments do not always depend on MyoII contractility[64]. Based on these results[63,64] and on our MyoII KD data, we hypothesized that muscles play a mechanical role in regulating activity-dependent plasticity (Fig. 5k, l), possibly adjusting GB number by providing physical confinement to the MNs. To test this, we uncoupled MN excitability from muscle contraction during stimulation (Supplementary Fig. 22a–c) either by using a Glutamate receptor (GluR) inhibitor to block muscle activation, or by mechanically stretching the larvae, reducing their contractibility.

As a strategy to reduce muscle contraction, we first used 1-Naphytlacetyl Spermine Trihydrochloride (NASPM), a GluR antagonist, which was shown to be effective at the *Drosophila* NMJ[65]. To find the best condition where the use of this drug resulted in effective blockade of muscle contraction, we tested a range of concentrations (200–500 μM) (Supplementary Fig. 23a, b), and chose to use 300 μM NASPM to examine the role of muscle contraction on activity-induced bouton formation. After stimulation in the presence of NASPM, we observed a strong reduction in bouton formation, both in wild-type and in Sqh-RNAi or Sqh-DN (but not in Sqh-CA), suggesting that bouton formation is dependent on muscle activity (Supplementary Figs. 23c–e and 24a–c). However, even though NASPM treatment reduced muscle contraction, it also affected postsynaptic signaling through GluRs activation and, consequently, muscle-MN communication, which is known to have a role in the regulation of activity-dependent bouton formation[14,66].

Therefore, to directly assess the effect of muscle contraction on the capacity of MNs to add boutons to their arbors in response to high-K$^+$ depolarization, we developed a method to mechanically stretch larvae by using extra dissection pins to minimize muscle tearing during

stimulation (Supplementary Fig. 22b). To validate the effectiveness of this technique in reducing muscle contraction, we imaged live larval fillets that were successively subjected to different stretching and stimulation conditions (Fig. 6a, top; see "Methods" for more details): (1) relaxed and unstimulated, (2) relaxed and stimulated, (3) stretched and stimulated. Expression of a tropomyosin protein trap in muscles (Tropomyosin-GFP under the control of the MHC promoter) permitted visualization of body-wall muscles and spinning disk microscopy allowed fast acquisition of muscular movements (500 ms). We show contraction kymographs for ventral muscles between segments A2–A4 for each condition (Fig. 6a, bottom and Supplementary Movies 19). We plotted the length of muscle 6 in segment A3 over time, which showed that mechanical stretching not only reduced the amplitude of muscle contractions but also prevented rhythmic contractions (Fig. 6b). Moreover, quantification of the mean and maximum displacement of the muscle during the imaging, within the same muscle, showed that both mean and maximum contractility, under high-K$^+$ conditions, were significantly decreased by larval stretching (Fig. 6c, d). Having a good method to block muscle contraction (Supplementary Fig. 22b), we tested whether this biophysical parameter influenced activity-dependent bouton addition in WT and neuronal MyoII KD larvae. When we mechanically blocked muscle contraction, GB frequency with high-K$^+$ stimulation was significantly decreased in both control and Sqh-RNAi NMJs (Fig. 6e–g); and the same was observed for neuronal expression of Sqh-DN, but not for Sqh-CA (Supplementary Fig. 24d–f). To ensure that the result was not due to destruction of the preparation by excessive stretching, we imaged the body wall muscles of larvae after stimulation, relaxed and stretched, and without stimulation (Supplementary Fig. 25), which confirmed that the muscle integrity was comparable between the larvae that were stimulated, relaxed or stretched.

In conclusion, our results (Figs. 5 and 6) showed that the increase in activity-induced bouton formation observed upon MyoII reduction or inactivation is dependent on muscle contraction, since the increase in GBs observed after MyoII KD disappeared when contraction was blocked. Furthermore, we also observed a significant reduction in the number of high-K$^+$ induced GBs in WT larvae, which suggests that at the *Drosophila* NMJ, muscle contraction is required for normal bouton addition in response to elevated activity.

## Acute structural plasticity is coupled to increased synaptic activity and muscle contraction

To further evaluate the contribution of muscle contraction for activity-dependent plasticity in WT, we tested the effect of distinct degrees of freedom for the muscle to contract (Fig. 7a)—relaxed, semi-stretched and stretched—while avoiding too much stretching of larvae, which leads to occasional muscle rupture. This manipulation resulted in a

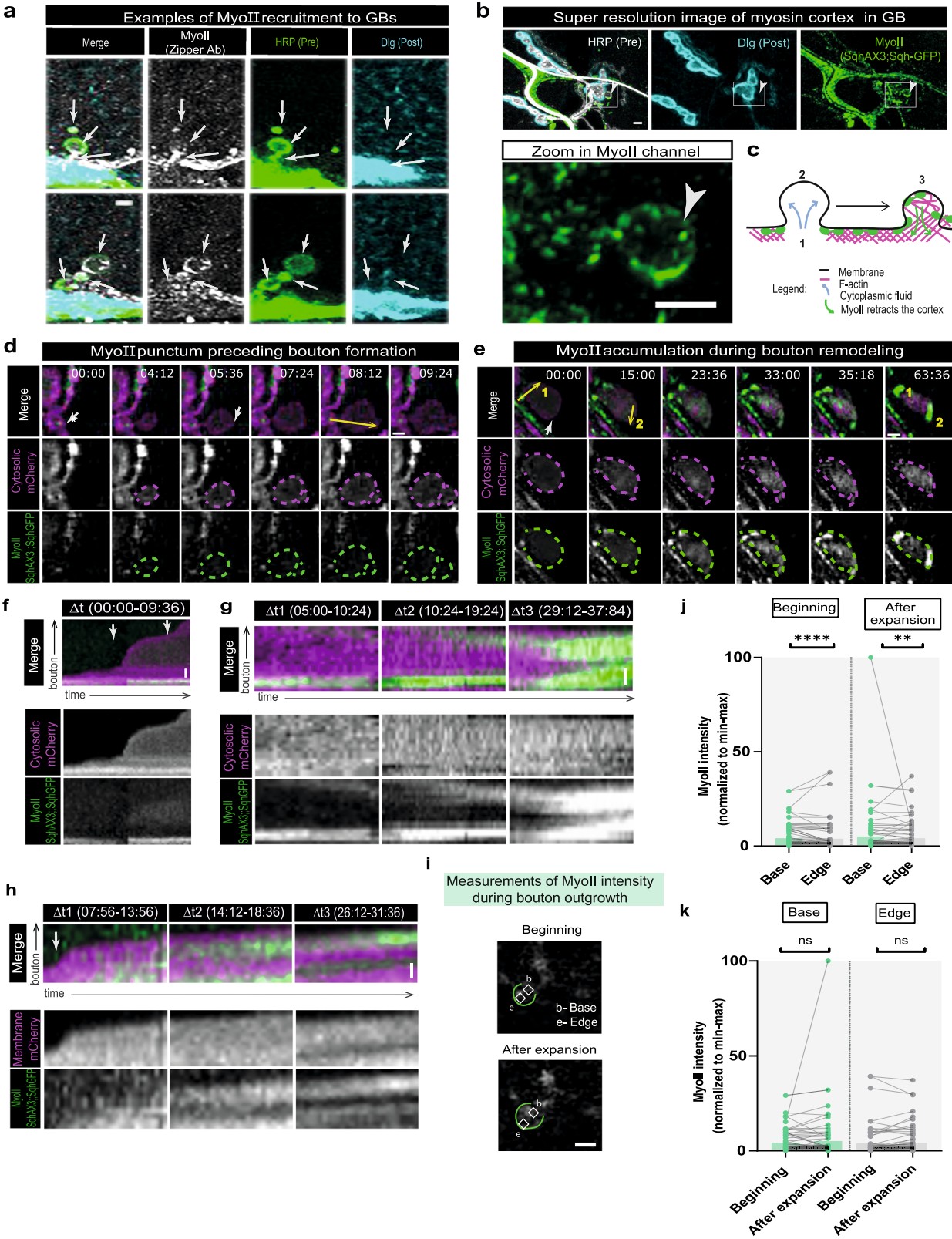

correspondent gradient of bouton formation, with an increased number of GBs being observed with a higher degree of muscle contractibility (Fig. 7b, c), which demonstrates that muscle contraction levels can change the extent of structural plasticity at the NMJ.

Furthermore, to test whether bouton formation is a result of muscle contraction due to increased synaptic activity, we quantified bouton formation after stimulation with increasing external $Ca^{2+}$ saline

concentrations (from 0 mM to physiological levels of ~1.5 mM $Ca^{2+}$). We observed a pronounced reduction in GB formation in $Ca^{2+}$-free saline, as has been previously shown[12], and here we show that both GB number and size steadily increase with incrementally higher $Ca^{2+}$ concentrations (Fig. 7d–f), concomitant with higher levels of both neuronal and muscle activity. Altogether, our experiments showed that, coupled with synaptic activity, muscle contraction plays a

**Fig. 4 | Recruitment of MyoII, a master regulator of blebbing, to GBs is tightly regulated in space and time. a** Post-stimulation non-muscle myosin-II (MyoII) heavy chain (Zipper antibody, gray) is recruited to ghost boutons (GBs) with visible accumulation at the base and/or edge. Presynaptic and postsynaptic membranes labeled with antibodies for Horseradish peroxidase (HRP-green) and Discs-large (Dlg-cyan), respectively. n = 32NMJs. **b** 2-D super-resolution image of MyoII regulatory light chain (Spaghetti squash(Sqh)-GFP, green). Zoom of MyoII corresponds to square in the lower magnification image. n = 10NMJs. **a**, **b** GBs (white arrow) identified by lack of Dlg. **c** Schematic of MyoII localization during bleb phases. **d**, **e** Time-lapse images of MyoII dynamics during bouton formation. Bouton growth visualized with LexAop-monomeric-Cherry (mCherry, cytosolic) under control of LexA driver DvGlut, to label MyoII we used GFP-tagged Sqh in a Sqh-null background ($sqh^{AX3}$). Scale bars, 2 μm. **d** Example of MyoII punctum at base preceding bouton formation. **e** Example of MyoII accumulation during bouton remodeling. 1-corresponds to (**f**), 2-corresponds to (**g**). **f**–**h** Kymographs of (**d**) and (**e**).

Scale bar, 1 μm. Kymographs for distinct Δt to highlight MyoII dynamics. Dotted lines along bouton edge (**e**) and base (**b**). **f** MyoII persists at base during bouton growth. **g** MyoII clustering coincident with bouton size reduction (Δt2−Δt3). **h** Small bouton forming at edge (arrow) of parental bouton (Δt1) and MyoII clustering as bouton gets smaller (Δt2−Δt3). **d**–**h** White arrow: where bouton emerges, yellow arrow indicates ROI used for kymograph generation. **i** Time-lapse images showing location where MyoII intensity was measured: squares of 1 μm² were positioned at base (**b**) and edge (**e**) of the bouton in the beginning and end of expansion. Scale bar, 2 μm. **j**, **k** Bar plots (mean) with normalized values for MyoII intensity. All data points represented, lines connect paired points. n = 9 larvae/92 boutons. Statistical significance by non-parametric Wilcoxon matched-pairs signed rank test (two-tailed); **j** MyoII levels at the edges were lower than basal myosin-II in both growth stages. ****$p < 0.0001$,**$p < 0.01$ ($p < 0.0001$ and $p = 0.0023$) **k** Myosin flow remained unchanged at the base or edge during bouton expansion. ns not significant. Source data provided as a Source Data file.

mechanical role in bouton formation. We suggest that in conditions of intense activity, muscle contraction increases MN confinement (an important biophysical factor in promoting cellular blebbing) to boost bouton formation at the *Drosophila* NMJ. Furthermore, we propose that, at the *Drosophila* NMJ, in addition to biochemical signaling, mechanical forces are coordinated between MNs and the muscle to regulate neuronal remodeling by blebbing (Fig. 7g, h).

## Discussion

Dissection of the mechanisms that regulate activity-dependent bouton formation is critical to understand remodeling strategies required for neuronal growth and wiring. Here, using live imaging of the *Drosophila* larval NMJ, a system in which 3-D intercellular and biophysical interactions are preserved, we showed that bouton formation occurred by membrane blebbing, a mechanism widely used in 3-D cellular migration but, to our knowledge, never reported to be used by neurons to remodel. Our study showed that new boutons induced by elevated activity, also called ghost boutons (GB), are bona fide blebs, whose growth does not rely on actin polymerization but is rather pressure driven. Additionally, we observed that activity-dependent bouton formation was frequently associated with muscle activity and that blocking muscle contraction significantly decreased GB frequency after stimulation. Moreover, the manipulation of synaptic activity and/or muscle contraction resulted in predictable changes in bouton formation in response to stimulation, with a progressive increase in new bouton numbers with higher levels of synaptic activity and/or muscle contraction.

Our results suggest that muscle contraction plays a mechanical role in activity-dependent bouton formation. At the NMJ, the MN is deeply imbedded in the muscle, which by contracting can directly alter neuronal confinement. We hypothesize that with elevated activity, and in response to a still unclear pre and/or postsynaptic signal to initiate bouton outgrowth, MNs add new boutons by membrane blebbing, and muscle contraction is required to increase neuronal confinement. This confinement results in increased pressure onto the MN membrane, powering bouton outgrowth at places putatively primed for bouton formation. In accordance, it is known that blebs are favored in conditions of high confinement, namely in tissues where cells encounter increased mechanical resistance. Furthermore, cells in compressive 3-D environments naturally go through stiffness gradients, which can polarize the recruitment of molecules required to or that facilitate blebbing in spots of high membrane tension. Likewise, at the NMJ it is possible that regions of higher activity produce more signals for inducing synaptic growth, which is supported by previous studies that showed that synaptic plasticity at the NMJ, including GB formation, requires activity-dependent secretion of postsynaptic signals such as BMP ligands (Maverick and Gbb) and $Ca^{2+}$-sensitive vesicle regulator Syt-IV, presynaptic Wg and Syt-I mediated NT release and postsynaptic glutamate receptor function[13–15,66–68]. It will be interesting to investigate

whether activation of pathways downstream of these factors can converge to determine sites primed for bouton initiation with synaptic activity, by recruiting initiation factors and/or changing cytoskeletal dynamics at these regions.

Cells migrating in confined environments typically display rounder morphologies and use hydrostatic blebs for movement. Interestingly, it has been shown that neurons from mouse central nervous system (CNS) growing in 3-D matrices, rather than 2-D surfaces, displayed GCs that had an amoeboid-like morphology[69], characterized by very low adhesive interactions and extensive rounded deformations of the body-wall, challenging the paradigm of the lamellipodia based GC. In the brain, even though there are no muscle compressive forces, synaptic boutons are equally confined and surrounded by other neuronal, astrocytic, or microglial processes. Interestingly, a recent study showed that in the mammalian brain, enlargement of dendritic spines produces mechanical pressure onto boutons thereby enhancing NT release, thus contributing to synaptic strength (a force reported to be comparable to that of muscle contraction)[70]. This exciting discovery is in line with our hypothesis that mechanical force directly regulates synaptic function, or as in the case of our study, formation of new boutons. It will be interesting to study how neurons can sense and respond to mechanical signals provided by the cellular environment, and if pathways used for mechanosensing and transduction, likely regulating adhesion and contacts with extracellular partners, are coupled with synaptic transmission machinery during plasticity.

While the molecular identity of the signal that leads to bouton initiation remains to be discovered, it is well established that blebs nucleate when a small patch of membrane is detached from the actin cortex[71]. This can happen either as a direct consequence of buildup in hydrostatic pressure (that detaches membrane to cortex binding proteins) or by formation of local gaps, resulting from rupture of the cortex in regions of high membrane energy (where accumulation of MyoII helps to weaken the cortex)[71]. Our data suggests that, although at the NMJ the two mechanisms may coexist, with elevated activity boutons tend to form mainly because of compressive forces exerted by the muscle. Supporting this, we found that activity-induced bouton formation was usually fast and correlated with visible muscle contraction, while events with low or no visible muscle contraction were slower and occasional. Additionally, post-stimulation, even though we observed MyoII puncta preceding bouton formation in ~35% of events, we found boutons forming with low MyoII levels and decreasing or inactivation of MyoII did not prevent bouton formation, which suggests that MyoII was not required to propel bouton growth induced by NMJ stimulation (although this result can not directly be extended to a null scenario). Interestingly, MyoII was recruited to ghost boutons in stimulated and unstimulated NMJs, which may explain cases where boutons formed with lessened muscle contractions. The fact that boutons always formed without filamin, which links actin to

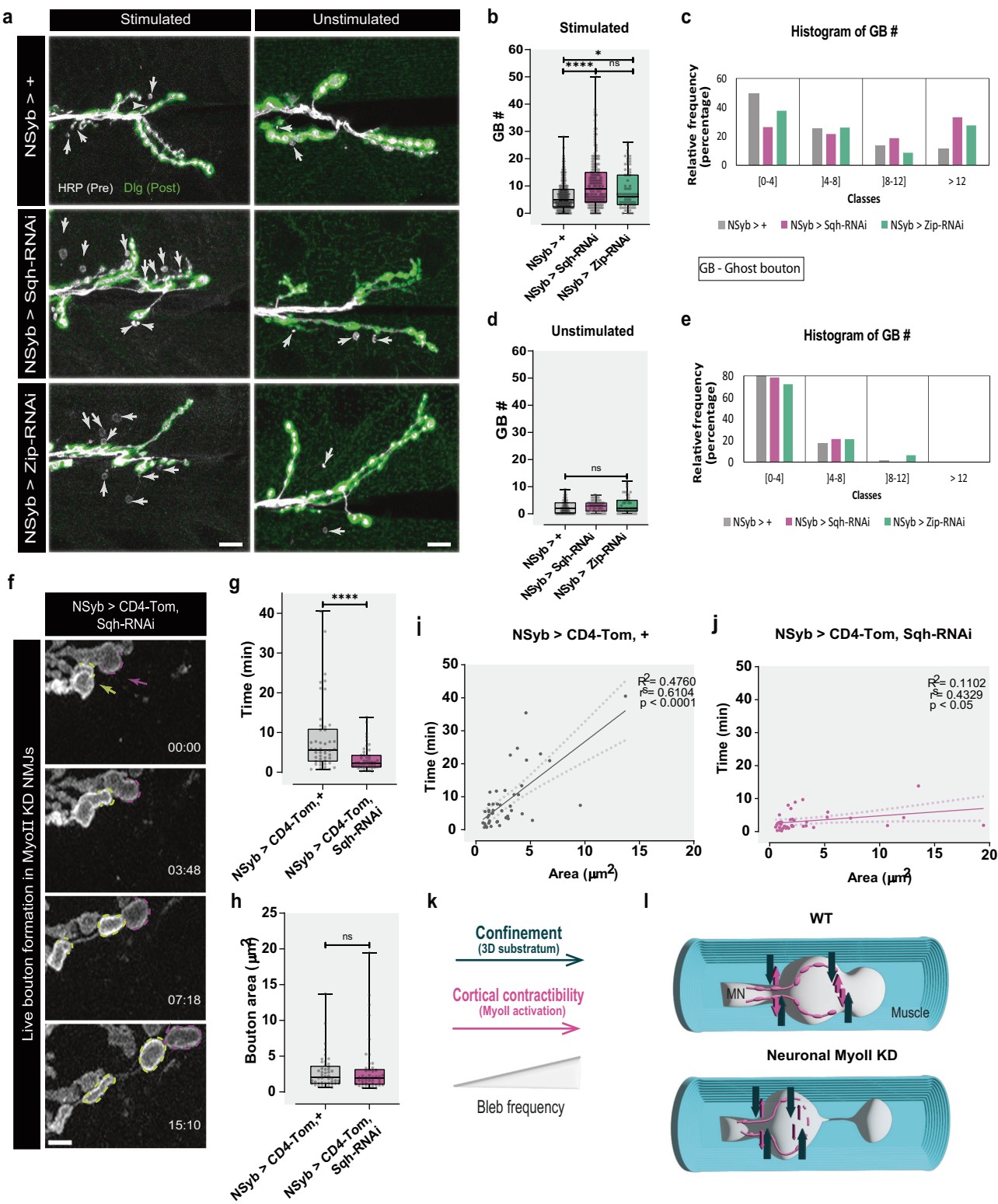

membrane, further supported that actin polymerization is not required for bouton initiation. Altogether we favor a model in which boutons form in regions of high pressure by detachment of membrane from the actin cortex and that local MyoII activation can facilitate the weaking of the cortex at these sites. Importantly, regardless of the exact signal used to initiate activity-dependent bouton formation, we showed that growth and stabilization of these boutons requires both actin and MyoII local rearrangements analogous to what is

reported for cellular blebs: weakened actomyosin cortex in early-stage boutons and enrichment after expansion[48].

Our model explains the formation of new synaptic boutons by a physical process—membrane blebbing, which in addition to the more studied transcriptional or biochemical factors, allows fast, on demand expansion of the NMJ. Whether axons are always competent for bleb-dependent bouton expansion or require priming by cytoskeletal regulation remains to be elucidated. Moreover, considering the

**Fig. 5 | Interfering with MyoII alters MNs capacity to form GBs. a** Representative images of control and MyoII K/D animals with (left) or without (right) high-K+ stimulation. We used NSyb-Gal4 to express RNAi against the two subunits of MyoII (Spaghetti squash – Sqh, regulatory light chain, and Zipper – Zip, heavy chain) in neurons. Presynaptic and postsynaptic membranes labeled with antibodies for Horseradish peroxidase (HRP, gray) and Discs-large (Dlg, green), respectively; ghost boutons (GBs) identified by white arrows. Scale bar, 10 μm. **b** Boxplot (min to max) showing that neuronal RNAi against Sqh and Zip increased GB frequency after stimulation. **c** Histogram showing relative frequency of GBs (%) in each class for the stimulated NMJs. Control n = 313NMJs/50 larvae, Sqh-RNAi *n* = 187NMJs/40 larvae, Zip-RNAi *n* = 69NMJs/20 larvae. **d** Boxplot (min to max) showing that neuronal RNAi against Sqh and Zip did not change GB frequency during development (unstimulated control). **e** Histogram showing frequency of GBs (%) in each class for unstimulated NMJs. Control *n* = 85NMJs/22 larvae, Sqh-RNAi *n* = 56NMJs/14 larvae, Zip-RNAi *n* = 47NMJs/14 larvae **f** Time-lapse images of bouton formation when MyoII

was K/D in neurons. NSyb-Gal4 used to drive UAS-CD4-Tom (membrane label) and UAS-Sqh-RNAi expression. Scale bar, 2 μm. Arrows indicate where boutons emerge. N = 13 NMJs/larvae (control), 5 NMJs/larvae (Sqh-RNAi). **g** Boxplot (min to max) reduced bouton formation time in neuronal MyoII K/D. **h** Boxplot (min to max) showing that GB area remained unaltered in neuronal MyoII K/D. *N* = 13 larvae/46 boutons (for control), 5 larvae/46 boutons (for Sqh-RNAi). Statistical significance determined using non-parametric two-tailed tests: Kruskal–Wallis (**b**, **d**) or Mann–Whitney (**g**, **h**); *$p < 0.05$ ($p = 0.0254$), ****$p < 0.0001$, ns not significant. Boxplots show $25^{th}$–$75^{th}$ percentiles, lines at the median, and whiskers from minimum to maximum. All data points are represented. Plots with linear regression between time and area for driver control (**i**) and Sqh-RNAi (**j**). 95% confidence bands of the best-fit line are indicated by dotted lines. $R^2$, Spearman's correlation value ($r^s$), and respective *p* values, are shown in the graphs. **k** Schematics of the factors that promote bleb formation. **l** 3-D schematic of muscle contraction effect on neuronal MyoII K/D. Source data provided as a Source Data file.

conservation of presynaptic cytoskeletal components and bouton ultrastructure throughout evolution[72], we postulate that this mechanism of synapse remodeling can be present in other organisms, including vertebrates. Overall, our finding that bouton addition at the NMJ occurs as result of a MN-muscle physical interplay highlights the importance of intercellular cooperation during plastic changes. Circuit remodeling as a response to experience requires rapid modulations of bouton number. We speculate that the regulation of confinement by non-neuronal cells (muscle or glia) can be a mechanism widely used by the nervous system to coordinate local activity-dependent structural changes in neurons with its surrounding 3-D cellular microenvironment. Future studies will elucidate whether the understanding of the biochemical and mechanical relations between neurons and their neighboring cells during structural plasticity can help design new strategies to remodel neuronal circuits that have been impaired by neurodevelopmental or neurodegenerative diseases.

## Methods

### *Drosophila* culture and stocks

*Drosophila melanogaster* stocks were maintained at 25 °C in standard medium, except for RNAi experiments where they were maintained at 29 °C. Crosses were set at a minimum of 5–10 virgin females and 3-5 males of the appropriate genotype. Adults were removed 7–8 days after each cross to ensure segregated generations. For larval collection, the eggs were laid and grown in apple juice vials at 25 °C. For RNAi experiments, RNAis-expressing strains and their controls were set up at 29–30 °C to maximize knockdown efficiency. TM6b- or GFP-expressing balancer chromosomes were used to facilitate genotyping of larvae (non-tubby or non-fluorescent larvae were selected). The *w*[1118] line was used as the wild-type (WT) genotype, as the stocks used were of this genetic background. For tissue-specific transgene expression, NSyb-Gal4 (pan-neuronal driver) was used to drive UAS-construct expression in neurons. As driver control, we crossed NSyb-Gal4 with *w*[1118].

The following *Drosophila* lines were used in this study: Stocks obtained from the Bloomington Drosophila Stock Center (BDSC): UAS-Zip-RNAi (#36727, #37480) UAS-Sqh RNAi (#32439, #31542, #7916, #38222), UAS-Sqh-CA (#64411), UAS-Sqh-DN (#64114), UAS-CD4-Tom (#35837), UAS-CD4-GFP (#35836), NSyb-Gal4 (from Thomas L. Schwarz Lab, #51635), OK6-Gal4 (from O'Kane lab, #64199), UAS-Act5C-GFP (#7309), UAS-BRP-GFP(#36292), Tropomyosin-GFP trap (#51537), sqhAX3;;SqhGFP (#57144), 13XLexAop-6XmCherry-HA (#52271), UAS-Lifeact-Ruby (Ivo Telley lab gift), UAS-Lifeact-GFP (Ivo Telley lab gift). From the Vienna Drosophila Resource Center (VDRC): GluRIIE (flyfos) VDRC#318061, Dlg (flyfos) VDRC#318133, Sqh (flyfos) VDRC#318487. From the Kyoto Resource Center: Zip-GFP (trap) Kyotto#115082. DV-Glut-LexA described in this study, see below for details. This information is also provided in a table format in Supplementary Table 1 and 2.

### Generation of transgenic flies

The DVGlut-LexA[37] driver was derived from the pCaSpeR VGlut » LexA construct[73] by removing the FRT cassette using KpnI restriction enzymes. Transgenic lines were generated by standard P-element-mediated transformation procedures in a *y,w* background. Random insertions were selected based on strength and specificity. 13XLexAop-6XmCherry-HA (attP2) from Bloomington, BL52271.

### Larval dissections and immunocytochemistry

Third instar larvae were dissected in HL3.1 saline (in mM: 70 NaCl, 5 KCl, 0.1 CaCl₂, 4 MgCl₂, 10 NaHCO₃, 5 Trehalose, 115 Sucrose, 5 HEPES-NaOH, pH 7.2-7.4) as described by Brent *et. (*2009). The gut and fat body were removed, while the CNS remained intact until after fixation. The resulting larval fillets were fixed in Bouin's fixative (saturated picric acid + formaldehyde + glacial acetic acid) or PFA (4% paraformaldehyde diluted in 1× PBS) at room temperature for 5 and 20 min, respectively, and extensively washed in PBT (1× PBS + 0.3%Triton). Unspecific binding was minimized by incubating for 30 min-1 h with 5% normal goat serum (NGS) dissolved in PBT. The primary antibody incubation was performed overnight at 4 °C, in blocking solution. Subsequently, the larvae were extensively washed using PBT, blocked for 30min-1h, and incubated for 2 h with the secondary antibody at room temperature. After extensive washing with PBT, larvae were transferred to 50% glycerol in PBS for 5 min and then mounted in DABCO medium (in 90% glycerol).

Primary antibodies: mouse anti-Dlg (4F3;1:250; Developmental Studies Hybridoma Bank; University of Iowa, Iowa, IA, USA, AB_528203), α-Actinin (2G3-3D7; Developmental Studies Hybridoma Bank), AB_2721943, rabbit anti-GFP (1:1000; Thermofischer A11122). The following antibodies were used and obtained as gifts since they are not commercially available: rabbit anti-Dlg (1:10000; Vivian Budnik lab), rabbit anti-Myosin-II, Zipper (1:1000, Christine Field); rat anti-Filamin (1:800; Mirka Uhlivora lab). All antibodies were previously validated (see reporting summary).

All secondary antibodies used in this study were purchased from Jackson Immunoresearch and used at 1:500 (initial dilution in 50% glycerol, according to manufacturer indications): A488/RhRx/A647 donkey anti-mouse (used for mouse anti-Dlg and mouse anti-α-Actinin), A488/A647 donkey anti rabbit (used for GFP or Myosin-II), A488 donkey anti-Rabbit (used for anti-mouse Dlg in filamin stainning) and Cy3 donkey anti-Rat (used for filamin). All with minimum cross-reactivity for all relevant species. Also from Jackson Immunoresearch Horseradish peroxidase (HRP) conjugated to Cy3, Alexa488, Alexa647 or Alexa405 was used to label neuronal membrane (1:500). All antibodies and conditions are summarized in Supplementary Table 3.

### High-K+ stimulation of larval NMJs for fixed and live imaging

Fixed: Third instar larvae were pinned onto Sylgard-coated plates using insect pins and partially dissected in HL3.1 saline solution (in

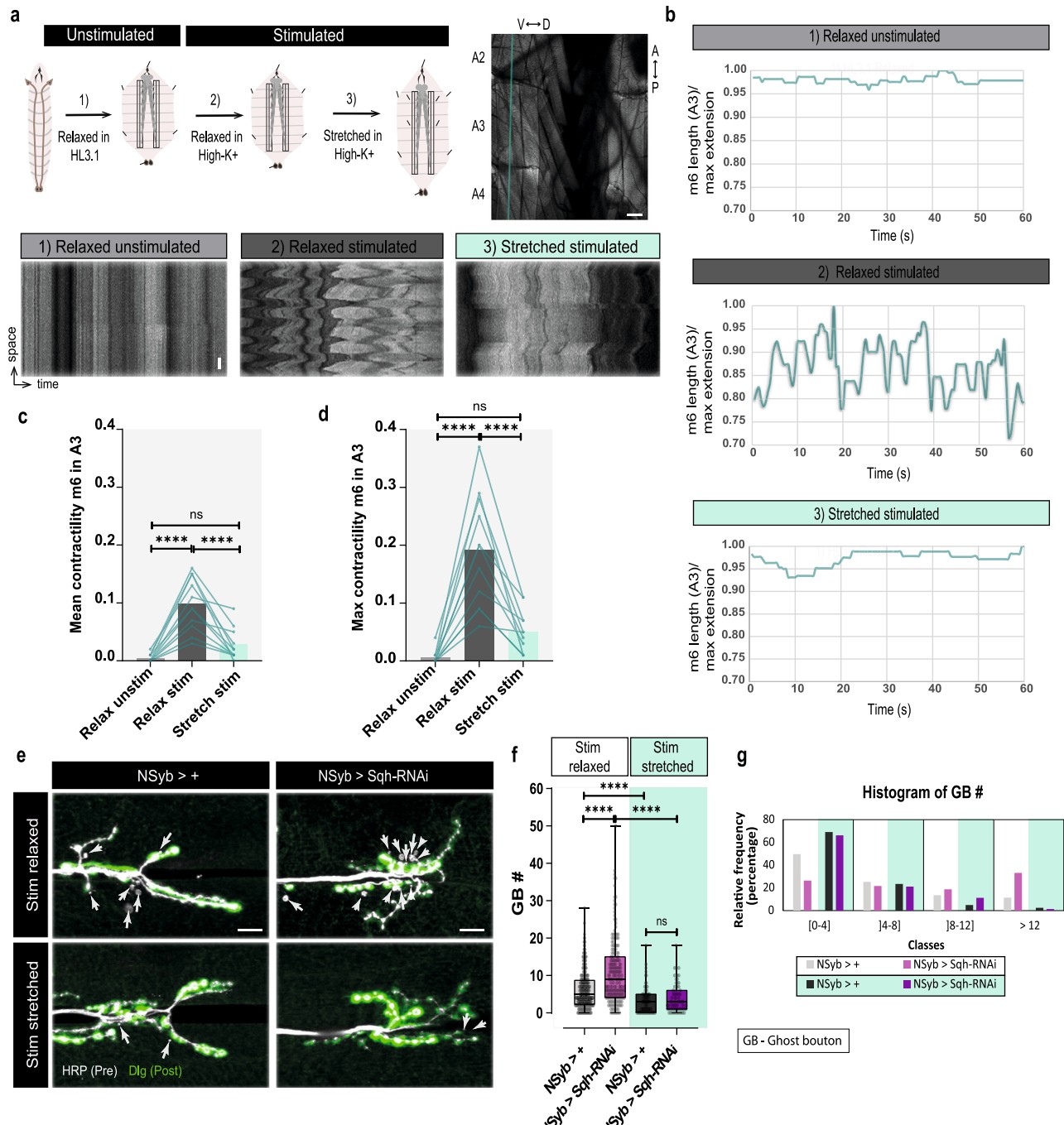

**Fig. 6 | Muscle contraction regulation of MN confinement is necessary for acute structural plasticity. a** Analysis of muscular inhibition by mechanical stretching. Schematic (left) of the protocol applied to larvae to achieve different degrees of stretching, where the same larvae is dissected to three successive conditions (1) HL3.1 relaxed (unstimulated), (2) high-K⁺ relaxed (stimulated relaxed) and (3) high-K⁺ stretched (stimulated stretched). Example of an image (right) of body-wall muscles visualized by expression of Tropomyosin-GFP, focusing on ventral muscles in segments A2-A4. The plotted line indicates the location used to create the kymographs shown below, which display the movement of the ventral muscles in each condition throughout time. Scale bar, 50 μm in body-wall muscles image and kymograph (over 300 sec). **b** Representative plot of muscle 6 length in segment A3 over time for each condition. Graphs were normalized by dividing the muscle length by maximal muscular extension. Top: unstimulated relaxed, middle: sti-mulated relaxed, bottom: stimulated stretched conditions. **c**, **d** Plots of mean and maximal contractility of muscle 6/segment A3. Lines connect the values obtained for individual larvae under each condition. $n = 11$ larvae. Statistical significance

determined by one-way parametric analysis of variance (ANOVA); ****$p < 0.0001$, ns non-significant. **e** Images of control and MyoII-K/D animals in which muscle con-traction was either allowed (relaxed fillets) or restricted (stretched fillets) after high-K⁺ stimulation. Presynaptic and postsynaptic membranes labeled with anti-bodies against Horseradish peroxidase (HRP-gray) and Discs-large (Dlg-green), respectively. Ghost boutons (GBs) (white arrow) are identified by the lack of Dlg. Scale bar, 10 μm. **f** Boxplot (min to max) showing that mechanically blocking muscle contraction by stretching, reduced GB number in Sqh-RNAi and control groups. Relaxed: Control $n = 313$ NMJs/50 larvae, Sqh-RNAi $n = 187$ NMJs/40 larvae (as in Fig. 5b), Stretched: Control $n = 81$ NMJs/15 larvae, Sqh-RNAi n = 71 NMJs/15 larvae, **g** Histogram showing relative frequency (%) of GBs in each class for relaxed or stretched animals. Statistical significance determined using non-parametric Krus-kal-Wallis test (two-tailed); ****$p < 0.0001$, **$p < 0.01$, ns is not significant. Boxplots show 25th–75th percentiles, line at median, and whiskers from minimum to max-imum. All data points are represented. Source data provided as a Source Data file.

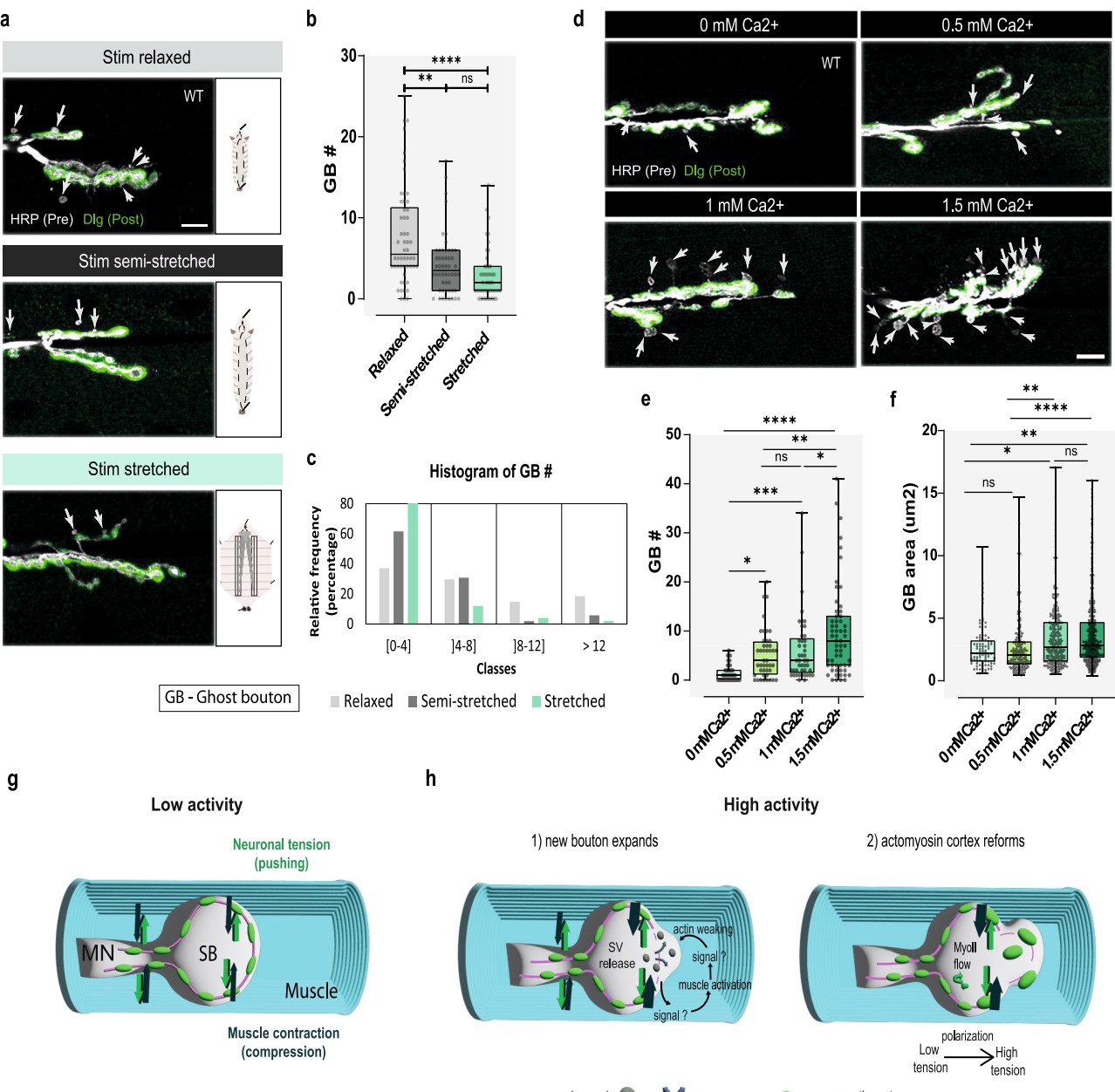

**Fig. 7 | Acute structural plasticity is coupled to increased synaptic activity and muscle contraction. a** Representative examples of WT (*w[1118]*) larvae allowing distinct degrees of muscle contractibility (relaxed, semi-stretched, stretched) after high-K+ stimulation. Scale bar, 10 μm. **b** Boxplot (min to max) showing that ghost bouton (GB) number progressively decreased by gradually restricting muscle contractibility. **c** Histogram showing relative frequency (%) of GBs in each class with distinct levels of muscle contractibility. *n* = 54NMjs (stim+relaxed), *n* = 50NMjs (stim + semi-stretched), *n* = 52NMjs (stim+stretched). **d** Images of WT larvae displaying progressively higher degrees of plasticity by incrementally increasing external Ca2+ concentration in saline from 0 mM to physiological levels of 1.5 mM Ca2+. Scale bar, 10 μm. **e** Boxplot (min to max) showing GB number is strongly reduced in Ca2+-free saline and steadily increases with enhanced external Ca2+ levels. **f** Boxplot (min to max) displaying GB size decreases with lower external Ca2+ levels and increases when further Ca2+ is added to the saline solution. **e**, **f** 0 mM Ca2+: *n* = 65 NMJs/89 GBs; 0.5 mM Ca2+: *n* = 44 NMJs/224 GBs; 1 mM Ca2+: *n* = 45 NMJs/227 GBs; 1.5 mM Ca2+: *n* = 59 NMJs/556, from 15, 12, 11 and 19 larvae, respectively.

Statistical significance by parametric analysis of variance (ANOVA); ****$p < 0.0001$, **$p < 0.01$, *$p < 0.05$, ns is not significant. Exact *p* values in "Statistics and Reproducibility". Presynaptic and postsynaptic membranes labeled with antibodies against Horseradish peroxidase (HRP-gray) and Discs-Large (Dlg-green), respectively. GBs (white arrow) identified by lack of Dlg. Boxplots show 25th–75th percentiles, line at median, and whiskers from minimum-to-maximum. All data points represented. Source data provided as a Source Data file. **g, h** Model proposed for activity-dependent bouton formation at the *Drosophila* NMJ. **g** Schematics of motor neurons (MNs) deeply embedded in the muscle, which is contractile, where synaptic boutons (SB) form. **h** Schematics of the effect of our model where increased activity (and in response to a yet unidentified signal) induces a local actin weakening, and the release of synaptic vesicles (SVs), priming places where MNs add new boutons by membrane blebbing. During this process, we hypothesize that muscle contraction can act through the compression of MNs to increase their confinement and cortical tension, facilitating bouton formation.

mM:70 NaCl, 5 KCl, 0.1 CaCl$_2$, 4 MgCl$_2$, 10 NaHCO$_3$, 5 Trehalose, 115 Sucrose, 5 HEPES-NaOH, pH 7.2-7.4) at room temperature. Importantly, prior to stimulation, the dissection pins were moved inward to the same guide shape at ~50% of the original size of each larva to allow for muscle contraction. We used the stimulation protocol described by Vasin et al.[13], which used a spaced high-K$^+$ depolarization paradigm to induce patterned activity and rapid bouton formation in *Drosophila* larval NMJ. Briefly, relaxed fillets were incubated in high K$^+$ (90 mM) and high Ca$^{2+}$ (1.5 mM) HL3.1 adjusted for osmolarity changes with 2-, 2, 2 min pulses each separated by 10 min incubation in normal HL3.1. For immunocytochemistry, the stimulated larvae were fixed 30 min after the 3$^{rd}$ stimulus to maximize bouton formation in the respective paradigms. Control larvae (non-stimulated) were dissected as described above but with only normal HL3.1.

Live imaging: The protocol for acute stimulation of boutons used for live imaging was identical to that described above unless otherwise stated. Third instar larvae were dissected and glued directly onto a Sylgard-coated slide (with GLUture topical tissue adhesive, Zoetis Inc.), and imaged in HL3.1 saline right after stimulation for 30 min to 1.5 h. Larvae should be stretched as little as possible to avoid muscle tearing and to allow muscles to contract around MNs, which we show here to be critical for GB addition. However, it is important to note that larvae contractibility is still somewhat restricted during the live imaging, which difficulted bouton formation and observation of events. For live imaging, the ventral nerve cord and CNS were maintained intact (and stretching was minimized as much possible). While most GB formation occurs during the stimulation, we were able to observe bouton outgrowth in ~30% of NMJs, which occurred mostly during self-generated muscle contractions.

Additionally, for live imaging, we also used a mass stimulation procedure of 10- or 16-min high-K$^+$ incubations developed by Martin et al.[74]. Specifically, paradigms of 3xK$^+$ and massed 10 or 16-min stimulations (mostly 16-min) were used to obtain data presented in Fig. 1c and Supplementary Fig. 2. Paradigms of 3xK$^+$ and massed 16-min stimulation were used for live experiments conducted in Figs. 4i, j; 5g–j; and Supplementary Fig. 18.

**Drug administration and manipulation of muscle contractibility**
**Actin manipulation.** The actin depolymerizer Latrunculin B (Focus Biomolecules) and actin stabilizer Jasplakinolide (ChemCruz) were used. These reagents were prepared as 25 mM and 3 mM stocks in DMSO, respectively, and diluted in HL3.1 and in high-K$^+$ high-Ca$^{2+}$ HL3.1 solutions to the desired concentration. Drug treatments were performed by pretreating dissected larval preparations with HL3.1 solution containing 10 μM Latrunculin B or 10 μM Jasplakinolide for 15 min. Stimulation was performed using HL3.1 and high-K$^+$ high-Ca$^{2+}$ HL3.1 solutions containing either 10 μM Latrunculin B or 10 μM Jasplakinolide. Each experiment was always performed side by side with a normal stimulation control, using HL3.1, and a solvent control, using DMSO (dilution used for LatB or JAS from stock).

**Muscle inactivation with NASPM.** To prevent muscle contraction, we prepared a 100 mM solution of 1-Naphtylacetil spermine trihydrochloride (NASPM) in H$_2$O, which was diluted to the desired concentration (mostly 300 μM NASPM solution) in HL3.1, and the same was done for high-K$^+$. The solution HL3.1 containing NASPM was incubated in the dark for 30 min prior to stimulation. We also used NASPM (Sigma, N193) in the stimulation protocol. During stimulation, the microscope light was turned off to minimize light exposure to the drug, and the larvae were fully stretched with 6 pins to minimize residual contraction and to verify drug efficacy (NMJs that contracted significantly exhibited visible muscle damage or tearing). Additionally, to validate the effectiveness of the drug in reducing muscle contraction, we stimulated relaxed fillets incubated with NASPM. We started by choosing the lower NASPM concentration to be effective in GluR

inhibition – 100 μM NASPM[75]. However, using this concentration post high-K$^+$ stimulation most muscles were destroyed, suggesting that larvae were still able to contract resulting in muscles pulling away from the body wall and destruction of the preparation. To find a condition where the use of this drug resulted in effective blockade of muscle contraction, we tested a range of concentrations (200–500 μM). At these higher concentrations, even though, some contraction was observed during the stimulation protocol, muscles were not so damaged indicating that muscle activity was attenuated. We chose to use 300 μM NASPM to test a role for muscle contraction on bouton formation induced by activity.

**Muscle inactivation by mechanical stretching.** To directly assess the role of muscle contraction during bouton formation, larvae were mechanically stretched and incubated with HL3.1 only (mock 30 min treatment and during stimulation). To block muscle contraction and simultaneously avoid preparation of tearing, larvae were stretched with 10 pins (additional 4 pins were placed surrounding the NMJ 6/7 between A2–A4). To minimize protocol variations, each experiment (same for NASPM) was always performed side-by-side with a normal stimulation control, with stimulation starting after a mock 30 min treatment in HL3.1. Additionally, to test whether different degrees of contractility can result in discrete plasticity outputs, larvae were divided into 3 groups: relaxed (2 pins, half size), semi-stretched (2 pins, full size), and stretched (6 pins not fully stretched to avoid muscle tearing). In this case, relaxed larvae were used as normal stimulation controls (without mock pre-treatment).

**Fluorescence imaging**
Confocal images were obtained using a laser scanning confocal microscope (LSM 710 mainly; LSM 980 was used for super resolution images) with a 40 × 1.3 NA water-immersion objective or a 63 × 1.3 NA oil-emersion objective (Carl Zeiss). Images were processed in ImageJ/FIJI3 (National Institutes of Health) and Adobe Illustrator (2022 version 26.0.1) and Adobe Photoshop (2020 version 21.2.1) software. The live imaging experiments were performed with a spinning disk confocal microscope (Andor) using a 60 × 1.3 NA oil immersion objective (Carl Zeiss), equipped with a heating stage heated to 25 °C. Quantification of the bouton number was performed at NMJ 6/7, abdominal segments A2–A4 were analyzed. In general, at least 12 (fixed) or 10 (live) NMJs of each genotype were analyzed for each experiment/time-point. The Software packages used were: Black 2011 SP1 for LSM 710, ZEISS ZEN 3.3 (blue edition) for LSM 980 and Andor iQ3 (3.6) for Spinning disk. Quantitative and video analyses were performed using maximum intensity projections from the z-stacks on the image. Images were mounted using Adobe Illustrator and Photoshop.

**GB quantification and size analysis**
As in previous studies[12,14], GBs were identified in fixed preparations by the presence of presynaptic HRP labeling (outlining the boutons) and the absence of postsynaptic Dlg staining (which is absent until ~3 h post bouton formation[15]). We considered dividing boutons, where constriction was clearly seen, as two. We counted GBs that were or were not attached to the main arbor, as the connecting filaments are known to be sensitive to fixation. To measure the GB size, we used ImageJ/FIJI 3 for the maximum projection images. We outlined GB with segmented lines and measured their area. We measured the maximal diameter using a straight line across the GBs.

**F-actin puncta and fluorescence intensity quantification**
F-actin puncta were quantified before and after a 15 min incubation with HL3.1, vehicle (DMSO concentration), 10 μM Latrunculin B or 10 μM Jasplakinolide to determine total number of puncta and puncta per NMJ area. Quantification of F-actin puncta was performed using ImageJ/FIJI 3 and maximum intensity projections from Z-stacks. The

actin puncta were defined as regions enclosing a local intensity maximum. The maximum luminance points were identified and measured as follows:

(1) Process > Filters > Maximum > Radius 0.1
(2) Process > Find maxima (adjust prominence to fit observable puncta before treatment and use the same prominence to adjust puncta after treatment)

<u>Output type</u>: point selection (displays a multi-point selection with a point at each maximum).

In short, we used the maxima function to detect local maxima within our images. Using this function and by defining a specific prominence, the maxima will only be counted when they stand out from the surroundings by more than the defined value for the prominence. We obtained NMJ area following these steps:

(1) Image > Adjust threshold (Actin channel)
(2) Process > Smooth
(3) Process > Binary > Make binary
(4) Process > Binary> Options > 1 iteration
(5) Process > Dilate
(6) Select the NMJ and measure the area

To calculate the total actin intensity at the NMJ, we performed Process > Image calculator: image with actin fluorescence AND image with NMJ area (obtained as shown above). Then, integrated intensity was measured. Additionally, we measured the integrated intensity for each actin puncta (identified as described above).

## Actin/MyoII kymographs analysis

Kymographs were used to visualize actomyosin cytoskeleton dynamics during live bouton formation. Specifically, to show motion or recruitment of actin or MyoII into boutons during a selected Δt – while these boutons were growing or remodeling. Since our movies show a lot of movement (due to the muscle contractions of the larval preparation), prior to generation of kymographs, videos were divided into discrete smaller Δt values. In the cases in which this strategy was not sufficient to remove the movement, we also corrected by rigid registration (2d/3d + t) in ImageJ/FIJI 3; membrane channel was aligned or adjusted to actin or MyoII. Kymographs were build using Multi Kymograph macro from ImageJ/FIJI 3. To create these graphs, we drew a ROI that extended from the region of bouton insertion and crossed the bouton for the chosen Δt. To define the starting point, we used a reference frame prior to outgrowth or remodeling of boutons and selected a point localized in the region underneath the bouton of interest, often in a mother bouton and less frequently in axons. From this starting point we drew a line that passed within the bouton in all times of the Δt for which the changes occurred, with the ending point placed at the bouton border whenever possible (using the frame when expansion or remodeling stopped as reference). Then, we plotted a profile for the selected line ROI encoding the intensity in grayscale level (or color for color images) for all slices in the Δt. In the kymographs, as they are oriented in the figures, the x-axis reports time (from beginning) and y-axis reports space (distance between region of bouton insertion and the expanding or remodeling bouton). After kymographs were generated, to measure changes that occurred within the bouton and at the insertion region, we traced for membrane and actin/MyoII a line along these locations and plotted the fluorescence intensity values (normalized for min and max values observed in each time interval) for these lines. In these plots, x-axis reports distance (along the line that we draw) and y-axis is fluorescence intensity for each line during this time window. In these graphs we show distances assigned to the bouton of interest and its insertion region to allow comparison between corresponding membrane or actin/MyoII levels. On top of the panels, we indicated the approximate time interval at which the changes occurred.

## Actin/MyoII intensity analysis

Maximum intensity projections from Z-stacks acquired in our movies were used to probe the fluorescence intensity of actin and MyoII at the base and edge of growing boutons. For each event, the intensity (integrated density) was measured in a $1 \times 1\,\mu m$ square ROI placed at the base (approximate point of origin of the new bouton) and in the edge of new bouton (farthest point parallel to the base). Base-edge pairs were defined, and measured for actin or MyoII intensity, in the beginning of bouton expansion (the frame when the bouton started to emerge) and after expansion ceased (maximum size frame). Whenever a bright actin or MyoII puncta was found at the base of new boutons these structures were included in the analysis (which represented 26% of cases for actin and 35% of cases for MyoII). To discount the photobleaching effects, we also measured the intensity at the main branch of the NMJ in the initial frame and when the bouton reached maximum size. A $1 \times 1\,\mu m$ square as the ROI. Actin or MyoII intensity was normalized by feature scaling (min–max), which preserves the relationships among the original data values, and multiplied by 100 to scale the data in the range of 0–100.

Additionally, to further analyze variation in actin or MyoII content, using absolute values, we subtracted intensity measured in the edge from the intensity measured in the base (base-edge). If base-edge = 0, edge equals to base; if base-edge > 0, edge is lower than base; and if base-edge <0, edge is higher than base. The % of variation inside boutons was calculated using actin or MyoII at the base as a 100% reference; the value was corrected for photobleaching effects by subtracting the percentage of change observed in the main branch (intensity at initial frame – intensity at maximum size). To quantify actin or MyoII, fluxes were measured by subtracting the intensity measured in the beginning from the intensity measure at maximum (beginning – maximum size). If beginning-maximum size = 0, maximum size equals beginning; if beginning – maximum size > 0, maximum size is lower than beginning; if beginning – maximum size <0, maximum size is higher than beginning. The % of variation was calculated using the actin or MyoII at maximum size as 100% reference; the value was corrected for the photobleaching effect by subtracting the percentage of change observed in the main branch (intensity at initial frame – intensity at intensity at maximum size).

## FRAP assay

FRAP experiments were performed using a Zeiss LSM 980 confocal microscope with a $25 \times 0.8$ NA water immersion objective. Lifeact-GFP positive MNs from 3rd instar *Drosophila* larvae A2/A3 segments were used. Each MN was imaged 5 times before bleaching; the photobleaching was performed using 50 iterations (0.5 ms each) of the 488 nm laser at 80% intensity in a circular ROI with 4 μm diameter. Imaging continued after photobleaching, at 1 frame per second for a duration of 300 s. To determine half-time of recovery ($t_{half}$), the time between bleaching and the time point when fluorescence recovery reaches half of its final intensity value ($t_{1/2}$), it is necessary to determine the half-recovered intensity ($I_{1/2}$), which corresponds to $t_{1/2}$ and can be calculated by the following equation:

$$(I_{1/2} = (I_E - I_0)/2) \tag{1}$$

where $I_E$ is the final intensity value, and $I_0$ is first post-bleaching intensity value. With $t_{1/2}$, $t_{half}$ is calculated as:

$$t_{half} = t_{1/2} - t_0 \tag{2}$$

where $t_0$ is the first time point after bleaching. Videos with preparatory movements and significant ROI shifts were excluded. FRAP data analysis was performed using Zeiss ZEN software blue edition, GraphPad Prism 8.01 or 8.0.2, Microsoft Excel v.16.70 and ImageJ (FIJI 3).

We present time-lapse images showing Lifeact-GFP expression in synaptic boutons during FRAP in Supplementary Fig. 12. The image signal intensity of these images was normalized by multiplication (to share a similar grayscale range) and they were represented in thermal LUT (look-up table) for better visualization.

To validate the use of Lifeact as a reporter for F-actin stabilization by applying FRAP analysis, we showed that after fluorescent photobleaching the half-time of Lifeact correlated with the half-time of actin in cells simultaneously expressing both actin probes. For this experiment, we analyzed larvae expressing Lifeact-Ruby and Actin5C-GFP (a fluorescent monomeric version of actin that can be incorporated into filamentous structures[57]) in boutons under the control of NSyb-Gal4. Photobleaching was performed using a 488 nm laser with similar settings to those described above.

### Optogenetic experiments – larva rearing

Crosses for optogenetic experiments (for live imaging and stimulation quantification) were reared at 25 °C in vials containing cornmeal fly food supplemented with 40 mM all-*trans* retinal (ATR – from Sigma). The ATR control cross (OK6-Gal4>20xUAS-IVS-CsChrimson::mVenus) was reared at 25 °C in vials containing ATR-free cornmeal fly food. Prior to the experiment, all crosses were reared in the dark to prevent unwanted light activation.

### Optogenetic plasticity stimulation assay

We used transgenic 3^rd instar larvae expressing UAS-CsChrimson in MNs using OK6-Gal4, to perform spaced depolarization, mimicking the high K+ stimulation protocol previously described. Briefly, open larvae were subjected to patterned light activation, achieved with an Arduino-controlled system containing 6 deep red (655 nm) LEDs from Luxeonstar (Brantford, ON, Canada) mounted on a circular arena placed ~1 cm from the sample with an intensity of 0.06 mW/mm$^2$ (within the published range of 0.02–1 Mw/mm$^2$)[76]. The following stimulation protocol was used (adapted from Maldonado-Diaz et al.[39]): 3 cycles of 2 min of light activation and 2 cycles of 10 min rest. Each light activation cycle consisted of 3 s with LEDs turned on and 3 s turned off repeatedly for the duration of the cycle. Larvae were partially dissected prior to stimulation, as described in the high-K+ stimulation protocol. Partial dissection was performed in HL3.1 containing 0.1 mM Ca$^{2+}$ and light activation in HL 3.1 containing 1 mM Ca$^{2+}$. After the third light activation cycle, all larvae were dissected in HL3.1 with 0.1 mM Ca$^{2+}$ and fixed in PFA 4%, 30 min after the end of the program. For live imaging, larvae expressing CsChrimson and CD4-Tom in MNs were dissected on a sylgard-coated slide, subjected to the same optogenetics protocol but imaged in a spinning disk microscope immediately after the 3^rd stimulation pulse.

### Long-term live imaging to assay bouton maturation

To assess whether GBs induced by high-K+ stimulation underwent maturation, we imaged the distribution of pre- and postsynaptic markers in these structures several hours post-stimulation. We tested two solutions to maintain our ex vivo preparations (dissected larvae) during prolonged periods. HL6 solution, more similar to larval haemolymph, initially preserves neuronal excitability but does not maintain the health of the preparation over 5–6 h. As an alternative, we used Schneider's insect medium (Thermo Fisher), which is known to support long-term culturing of the *Drosophila* brain[77] *and* larval motor axons[78], allowing more than 10 h of viability. Larvae were dissected as previously described using GLUture topical tissue adhesive (Zoetis Inc.) as a replacement for the dissection pins, and then subjected to an incubation period in Schneider's post-stimulation (30 min, 3, 5 and 10 h). Time-lapse imaging was performed at each timepoint with larvae submerged in Schneider medium, using a spinning disc confocal microscope (Andor) with a

60 × 1.3 NA oil immersion objective (Carl Zeiss). The same NMJs were imaged throughout each timepoint, which were matched retrospectively.

### Imaging and quantification of muscle contraction

We developed a method to block muscle contraction during stimulation, which consisted of mechanically stretching the larvae with 10 dissection pins (6 pins used in normal dissection plus 4 pins between segments A2–A4) to minimize muscle tearing during stimulation and maximizing the degree of contraction. To validate the effectiveness of this technique in reducing muscle contraction we imaged live larval fillets that were successively subjected to different muscle-stretching conditions: (1) relaxed and unstimulated, (2) relaxed and stimulated, (3) stretched and stimulated. We used larvae expressing a tropomyosin protein trap in muscles (Tropomyosin-GFP under the control of the MHC promoter) to visualize body-wall muscles and spinning disk microscopy, which allowed very fast acquisition of muscular movements (500 ms). First, the larvae were marginally reduced to ~70% of their original size and dissected with four small pins onto Sylgard-coated slides to allow live imaging of the abdominal muscles (A2–A4). We then imaged this ROI in HL3.1, a scenario equivalent to unstimulated, in which no or very few contractions were observed. Next, we exchanged HL3.1 with a high-K+ solution used for stimulation and clearly observed patterned contractions of the body wall muscles. Following this, we replaced larvae in HL3.1 and carefully stretched them to their original size, added 4 extra pins between segments A2–A4, and exchanged HL3.1 with high-K+, and imaged this condition. In this case, contractions were visibly attenuated. Importantly, a significant difference between the real experiment and this assay is that relaxed larvae do not have to be fully dissected during the procedure, therefore can be reduced to smaller sizes (~50% of the original size), allowing higher contraction levels. Therefore, it is likely that the level of muscle blockade achieved by mechanical stretching the larval preparations in our experiments is higher than what we have shown with this assay.

Kymographs showing the movements of the ventral muscles between A2–A4 for each condition were obtained in a way similar to described in Hiramoto et al.[79]. Kymographs were build using Multi Kymograph macro from ImageJ/FIJI 3. We drew a line ROI from the middle of muscle 6, encompassing the imaged segments (A2–A4), and plotted this profile for all slices in the videos (corresponding to 1 min in each video). In the kymographs, as they are oriented in the figures, the x-axis reports time (from beginning) and y-axis reports space (muscle 6 from A2 to A4).

To quantify muscle contraction and relaxation, we manually measure the length of m6 in segment A3 in each frame during the recordings. We then plotted muscle length, which was normalized by dividing this length by the maximal muscular extension, over time. In these graphs, the local minimum points (valleys) represent contractions, while the local maximum points (peaks) represent relaxation. For each time point the contractility level was obtained as follows:

1 – (muscle length/maximal muscular extension)

*1 is when muscle length = maximal muscular extension

From this, we calculated both mean and maximum contractility for each condition.

### Statistical analysis

Statistical analyses were conducted using GraphPad Prism Version 8.0.1 and version 8.0.2 (GraphPad Software). First, we performed a descriptive analysis of the data to study the distribution and variance associated with each condition/genotype sampled. Furthermore, all data sets were tested for normality using Shapiro−Wilks normality test. We opted to perform a non-parametric based inferential statistical analysis since we found the data to be asymmetrically distributed, high

covariance levels (superior to 60%) and frequently data sets did not pass normality. Statistical significance in two-way comparisons was determined by Mann–Whitney test, while Kruskal–Wallis analysis was used when comparing more than two datasets. In both cases we performed two-tailed tests. When comparisons were made between paired measurements, we performed non-parametric Wilcoxon matched-pairs signed rank test, as indicated in the figure legend. In figures data is presented as median (interquartile range) unless otherwise stated; ****$p < 0.0001$;***$p < 0.001$; **$p < 0.01$; *$p < 0.05$, n.s. not significant for the cases where datasets passed the normality test, statistical significance in two-way comparisons was determined by a Student's $t$-test, while ANOVA analysis was used when comparing more than two datasets. Statistical comparison is made between all groups. Sample size is presented in figure legends and "Methods" section "Statistics and reproducibility".

## Statistics and reproducibility

Data was collected using Excel version 17.71, and all statistical tests were performed in Prism (Graph Pad Software, Inc v8.0.1 or 8.0.2.) version 8.0. For each experiment, data distributions were tested for normality using a Shapiro-Wilk test (which is a test that can also be used for small samples). Comparisons of means between different conditions were tested with a Student's $t$-test (two-way comparisons) or with an ANOVA (when comparing more than two data sets) if the data passed the normality test. Otherwise, statistical significance in two-way comparisons was determined by non-parametric Mann-Whitney test, or Kruskal–Wallis test when more than two data sets were tested. When comparisons were made between paired measurements, we performed non-parametric Wilcoxon matched pairs signed rank test, as indicated in the figure legend. In all cases two-tailed tests were used. Sample $P$ value summaries: *$p < 0.05$, **$p < 0.01$, ***$p < 0.001$, ****$p < 0.0001$. ns not significant ($p$ value > 0.05). Statistical comparison is made between all groups.

**Independent biological experiments (per figure).** These represent independent with a minimum of 5–10 females and 3–5 males per cross. All independent experiments were performed with a minimum of 4 different animals per experiment (normally 6 larvae per genotype per experiment). Figure 1c−10 experiments, 1d−2 experiments, 1f, g−2 experiments, 1 h−7 experiments; Fig. 2i, j−9 experiments, 2f, g−1 experiments; Fig. 3c, e−2 experiments, 3d, f−2 experiments, 3g, h−2 experiments, 3j, k, m, n−3 experiments; Fig. 4a−4 experiments, 4b −1 experiment, 4j, k−5 experiments; Fig. 5b, c−10 experiments (control), 4 experiments (Sqh-RNAi and Zip-RNAi, for each), Figs. 5d, e−7 experiments (control), 3 experiments (Sqh-RNAi and Zip-RNAi, for each), 5g−j−10 experiments (control), 5 experiments (Sqh-RNAi); Figs. 6a–d−3 experiments, Fig. 6f, g−10 experiments (relaxed), 5 experiments (stretched); Fig. 7a, c−3 experiments, 7e, f−4 experiments.

All $p$ values with **** correspond to <0.0001 and are not given as exact $p$ values, ns corresponds to a $p$ value > 0.05, the exact $p$ values for the experiments with significance of *, **, *** are the following (most of these values are also provided in the figure legends): Fig. 3d, $p$ value = 0.0013; Fig. 3h, $p$ value = 0.0005; Fig. 4j, $p$ value = 0.0023; Fig. 5b, $p$ value = 0.0254; Fig. 7b, $p$ value = 0.0044; Fig. 7e, $p$ value (0 vs 0.5 mM Ca2+) = 0.0108, $p$ value (0 vs 1 mM Ca2+) = 0.0008, $p$ value (0.5 vs 1.5 mM Ca2+) = 0.0026, $p$ value (1 vs 1.5 mM Ca2+) = 0.0236; Fig. 7f, $p$ value (0 vs 1 mM Ca2+) = 0.0287, $p$ value (0 vs 1.5 mM Ca2+) = 0.0022, $p$ value (0.5 vs 1 mM Ca2+) = 0.0018.

## Reporting summary

Further information on research design is available in the Nature Portfolio Reporting Summary linked to this article.

## Data availability

Source data are provided with this paper as supplementary material. All other data associated with this manuscript is available upon request.

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

## Acknowledgements

We would like to thank Telmo Pereira from the Microscopy Facility for technical support, the Fly Facility at Nova Medical Research; CONGENTO: consortium for genetically tractable organisms. We thank the Developmental Studies Hybridoma Bank, Bloomington Drosophila Stock Center and VDRC for antibodies and fly stocks. This work was supported by PTDC-01778/2022- NeuroDev3D to R.O.T. GEMiNI and PTDC/BIA-COM/0151/2020 to C.S.M and European Research Council H2020-GA 810207-ARPCOMPLEXITY to E.R.G.. A.R.F. is supported with a PhD scholarship from Fundação para a Ciência e Tecnologia, Portugal, reference SFRH/BD/144488/2019, and J.P.M. with a reference SFRH/BD/130920/2017. This work also supported by iNOVA4Health (UIDB/04462/2020 and UIDP/04462/2020), and LS4FUTURE (LA/P/0087/2020).

## Author contributions

A.R.F and R.O.T designed and executed the experiments. A.R.F. analyzed the data. A.R.F. and R.O.T. wrote the manuscript. J.P.M executed and analyzed the optogenetic experiments, contributed for FRAP and muscle stretching experiments (Fig. 6a–d), and edited the manuscript. C.S.M. built the LexA-DVGlut driver and the optogenetic setup, contributed with intellectual input and edited the manuscript. E.R.G. contributed with intellectual input and edited the manuscript.

## Competing interests

The authors declare no competing interests.
