## [Peer Review File · Nature Communications]

REVIEWER COMMENTS

Reviewer #1 (Remarks to the Author):

Fernandes et al., 2021 Motor neuron boutons remodel through membrane blebbing

This manuscript makes the interesting proposal that synaptic bouton addition through a process they term 'blebbing' is intimately tied to muscular contraction and synaptic activity at the *Drosophila* larval NMJ terminal. Synaptic boutons are a fundamental organizational unit of both invertebrate and vertebrate peripheral and central synaptic terminals so understanding the mechanisms that govern their development is intrinsically important. However, the manuscript does not study normal bouton development, but rather 'ghost boutons', an artificially induced phenomenon whose relationship to normal bouton development is unclear and could be argued to be a pathological phenomena. While the authors may argue, as other studies have done, that ghost boutons could be perhaps be precursors of 'real boutons', without definitive establishment that this is actually the case, the import and findings of the manuscript is greatly diminished.

Major issues.

1. Artificially induced ghost boutons may bear no relationship to normal bouton formation. The authors interchangeably and incorrectly confuse terminology between 'boutons' and 'ghost boutons'. Synaptic boutons develop pre and postsynaptic specialisations including active zones for synaptic vesicle release, discrete accumulations of synaptic vesicles, postsynaptic glutamate receptor puncta and 'SSR' elaborations. In contrast, 'ghost boutons' induced by KCl or other treatments have not been shown to produce any of these structures. The blebbing the authors study appears to occur only in ghost boutons which do not show many of the features of normal boutons outlined above suggesting they are not functional and could be an aberrant artificial phenomena. If the manuscripts ultimate findings are that induced extreme hyperactivity causes an increase in non-functional membrane elaborations that never progress to become functional participants in neurotransmission, the relevance is not clear. To address the issue the authors should aim to develop a preparation where normal bouton development can be examined and manipulated. This seems challenging in vivo, but perhaps an ex-vivo preparation could be used, with a number of solutions that allow long-term ex-vivo maintenance of larval preparations (e.g. HL6 etc.) that might allow normal bouton addition to progress and be manipulated. Without data of this nature, the manuscript findings would need to be specifically (and less interestingly) restricted ghost boutons.

2. Analysis of ghost bouton formation. Although stimulation method the authors used has previously been published on several occasions, they should recognise their preparation is highly artificial due to (i) physiological saline composition and (ii) methodology of stimulation. I would like to see direct interrogation that the bouton addition they observe is not at least in part attributable to their methodological approach and clearly presented. Suggested experiments would include – (a) Observation of the blebbing process at NMJs without addition of KCl and report if they observe similar or lower levels of blebbing – this data is presented in fig.4 as a consequence of another manipulation but would be better tackled directly for validation of the technique in their conditions. (b) Quantify blebbing at incrementally increasing external Ca²⁺ saline concentrations (from 0 mM to physiological levels ~ 1.5mM). If blebbing is a result of muscle contraction due to increased synaptic activity, the authors should observe no or heavily reduced blebbing in Ca²⁺ free saline which should

increase with increased Ca²⁺ concentration (c) KCl addition is a far more extreme stimulation than what would be experienced through endogenous motor activity I would suggest authors ideally visualise blebbing through alternative approach, either direct nerve stimulation in line with parameters for known endogenous activity or through pulsed ChR stimulation, live or analysed post hoc to confirm increased blebbing with alternative stimulation parameters.

3. Stretching assays are imprecise. The classification of semi stretched, stretched and relaxed is very subjective in my opinion. Can the authors be sure that excessive stretching does not cause ruptures to muscles and thus changes in membrane resistance? How did they ensure that the stretching force is the same across samples? Tools such as GtACR1 or GtACR2 could be used to express in muscles to counteract muscle contractions to independently validate their findings. Do the authors observe any of the reported effects if the cuticle is also stretched?

Additional issues.

Figure 2. Page 10 – line 2 onwards . Authors appear to blame high variability for a lack of effect in JAS treatment on new bouton number. However, in fig.2j, HL3 saline has the highest variability, I would suggest they remove misleading interpretation of data ('although we could observe fewer boutons, we also observed unaffected boutons') this is likely just a description of biological variability in their system and should not be confused with any statistical comparison or real effect.

Figure 3. It is not intuitive to me why or how the authors chose to study muscle myosin ii despite reading their rational, the logic seems counterintuitive to the most interesting aspect to the paper – muscle force mediated bouton addition. Regardless, fig 4d shows bouton addition occurs when unstimulated, do they see similar myosin II localisation when boutons form without being stimulated? Visualisation of a similar mechanism when unstimulated will help address the above major critique. Furthermore – would the authors comment on why blebbing is observed when not stimulated and muscles are not contracting?

Figure 4. As a further control for SqhDN and SqhCA experiments, authors should upregulate WT Sqh for comparison, expression level and presence of a mutant are confounding in some experiments. Has RNAi-sqh been published before? If not, the efficiency and specificity should be shown, e.g. with a western blot.

Figure 5. Philanthotoxin is more commonly used to block drosophila GluRs, can authors rationalise the use of NASPM vs. PhTx. As the authors note, inhibiting GluRs is not an ideal way to neutralize muscle contractions because glutamate receptor inhibition has been shown to inhibit ghost bouton formation through other mechanisms (see Ref 21 in the manuscript) i.e. by affecting the retrograde signaling from the muscle.

Text. The logical flow of the manuscript could also be improved. For example, it is confusing for the reader to understand whether actin is or is not involved.

Reviewer #2 (Remarks to the Author):

In this paper, Fernandes and co-workers describe the mechanism by which motoneuron (MN)-muscle interactions generate new boutons in the *Drosophila* larval nervous system, allowing MNs to form membrane blebbing as a strategy for local remodeling of presynaptic structures. The authors use a combination of genetics and microscopic approaches to explore bouton genesis at the NMJ of muscles 6/7. They report that an amount of F-actin is relatively low inside new boutons, suggesting that actin polymerization is not essential for bouton formation. They argue in addition that the recruitment of myosin-II to new boutons slows down the growth of new boutons by classical genetic analysis (albeit primary with RNAi and gain-of-function analyses). They further argue that mechanical forces from muscle contraction promote bouton formation at the NMJ. However, the data supporting this contention is not compelling.

I think this paper provides an essential step towards the mechanistic understanding of bouton genesis. However, it requires additional experiments to provide insight into the molecular mechanisms regulating bouton genesis. I have listed my specific concerns about this paper below.

Specific comments:

[Major comments]

1. Introduction: Given that the trans-synaptic forces mediate synaptic changes (refs 23, 28-35), this must be intensively discussed in the introduction. The findings in the references established through genetics and imaging analyses that a variety of cellular proteins are involved in bouton genesis. Fernandes et al. extend these works to the molecular function of actomyosin networks and make significant contributions towards understanding bouton formation.
2. Page 6. The authors point out the potential involvement of a budding mechanism. A quick google search shows an intriguing paper (PMID: 12160747). In the paper, Pennetta et al. have demonstrated that microtubules play an essential role in initiating the budding of boutons. Because there has been an alternative scenario, the authors need to discuss and ideally test it experimentally.
3. Page 6. Can the F-actin distribution along the length of boutons (Fig 1 d-e) be quantified with a couple of more examples?
4. Page 6, 7. Because the authors state "(blebbing) is driven by a flow of cytosolic fluid across a local weakening (rupture or detachment) in the cell actomyosin cortex," they should provide examples of boutons that show rupture or detachment in the actomyosin cortex using a higher resolution imaging approach.
5. Page 7. What are the controls of overexpression of Lifeact-Ruby? Where is the transgene inserted, and what control transgenes are used? These should be shown at least in Supplementary material. Levels of Lifeact's expression need to be carefully calibrated by proper controls; however, data were not presented to establish this point.
6. Page 9. Please comment on the localization of actin puncta with JAS (Fig 2a). Furthermore, argue how those are correlated with the position of blebbing sites.

7. Page 13. The authors state, “we opted to use RNAi to decrease ... myosin-II levels”. Are there data establishing that the protein level is reduced in these neurons?

8. Page 13. An essential complement to the RNAi analysis would have been assessing the requirement of myosin-II with loss-of-function mutants. The author should use the MARCM system to study the function of myosin-II in these neurons.

9. Page 16, 17, 18. The addition of physical confinement to MNs is a critical experiment. A more detailed assessment of the frequency, strength, and duration of muscle contractions should be included to confirm that the effect is direct rather than indirect.

10. Page 18. The last paragraph of the Results overstated the implication of the results in this paper. They state that “... a balance of mechanical forces is tightly coordinated between NMs and the muscle to regulate remodeling at the NMJ...”. The results have not indicated this. They have put together a solid story that myosin-II acting with actin regulates bouton genesis. However, a weakness of the paper is that they have not provided credible evidence of whether muscle contraction locally provides mechanical force to the NMJ.

[Minor comment]

11. No explanation is included for Fig 2h, 3b, and 5j in the main text.

Reviewer #3 (Remarks to the Author):

This manuscript provides important mechanistic insight into the basis of acute structural synaptic plasticity at a widely used model synapse, the *Drosophila* neuromuscular junction. A multitude of studies over the last 10+ years have used acute stimulation to delineate signaling pathways important for new synaptic bouton formation in this system, without understanding the cell biological nature of new bouton growth. The current study uses a combination of live imaging and *Drosophila* genetics to suggest that rapid new bouton addition occurs by blebbing, which has been extensively characterized as a mechanism for cell motility, but to my knowledge not previously considered as a mechanism for synapse morphogenesis. The authors go on to show that muscle contraction plays an important role in formation of these blebs, indicating that mechanical forces cooperate with biochemical signals to drive structural plasticity.

The overall message will be of broad interest to the field and is generally supported by the data (though see below for specific questions and concerns). However, the manuscript requires a number of changes before it is accessible to readers and properly acknowledges its impact and place in the field. In particular, the authors fail to provide enough background of previous work studying this paradigm of rapid new bouton addition, and how their results fit in with these previous studies.

Major points:

1. The new boutons quantified in the live imaging studies seem extremely heterogeneous. In addition to the expected budding of boutons from pre-existing boutons (Fig 1d) some of them appear to originate from naked axon rather than from existing NMJ (e.g. Fig 1a bottom panel,

305108_0_video_893854_qqj956.avi) or seem to form from boutons splitting or just changing shape (e.g. Fig 1a top panel and Figure 1e, 305108_0_video_893855_qqj9j6.avi). The authors mention that some form quickly and some form slowly (as previously described by Piccioli et al., 2014). Since all these types of events are lumped together in the live imaging analysis without clarifying their distribution, it is unclear whether certain types of events dominate the quantifications. Also, due to this heterogeneity, it is not clear what fraction of boutons assessed by live imaging relate to the Dlg-negative, HRP-positive structures characterized in this paper and in the literature as “ghost boutons” in fixed, immunolabeled samples. It is particularly hard to imagine that “splitting” boutons in the middle of a string are Dlg-negative.

The authors should explain why it is appropriate to pool these apparently different types of events together, and describe the distribution is between these types of events. They should also more directly relate their live and fixed imaging, by quantifying from two-color live imaging experiments how many presynaptic events (either type of new bouton according to the categories above, or live actin behavior) lack a corresponding Dlg signal. There is a Dlg knockin available that may be suitable. For this experiment, please ensure that these presynaptic events are identified blind to the Dlg channel.

2. One of the major arguments that the structures are blebs is that they have reduced F-actin and filamin compared to the parent bouton. I do believe the structures are blebs based on their speed of formation, shape, regulation by myosin, and role of external mechanical forces. However, I am less enamored of the actin localization arguments. The authors do not know the identity or function of the major F-actin or filamin structures in the parent boutons, which provide the baseline for their measurements, and this could really matter. For example, if most F-actin in the parent bouton is at sites of compensatory endocytosis (as suggested by the observation that FM dye uptake upon stimulation is actin-dependent (Wang 2010)), then it would not be surprising that these structures are absent from ghost boutons, which only very rarely mature enough to recycle synaptic vesicles (Vasin 2014). On the other hand, any kind of actomyosin cortical cytoskeleton that forms the edges of a bleb or eventually enters into a new bouton might be inherently much less abundant relative to endocytic or other structures. It might be helpful to see filamin and Lifeact co-staining, to demonstrate what proportion of the presynaptic actin co-labels with filamin. The authors should also note that there isn't currently experimental evidence for a cortical actomyosin cytoskeleton in synapses, comparable to the cell cortex of motile cells, so such an apparatus remains hypothetical. Overall, the authors should either justify the importance of the actin localization data, or explain this major caveat and play down the importance of the actin localization evidence in the manuscript.

3. For the actin drug experiments in Fig 2, which are important for the blebbing argument, the authors state that they measure “the number of F-actin puncta” to assess the effectiveness of the drugs at destabilizing or stabilizing actin filament structures. There are a few arrows on n=1 NMJs per treatment, but no quantification is shown as they claimed. This is required. Since we don't know what these structures are, lifeact intensity, number of puncta/NMJ area, and coefficient of variation of lifeact would be an appropriate measure, rather than raw number. For the Jas samples, a FRAP assay showing slower turnover may also be useful if steady-state quantifications are unclear. The data in Figure 2 are uninterpretable without these measurements, especially given the lack of change in ghost bouton number which is the opposite of the observations in Piccioli et al (reduced ghost boutons in both LatA and Jas). These quantifications should be combined with a repeat of the main results of Figure 2 in parallel, to ensure that the drug treatment worked in the same samples as the bouton size measurements are made.

4. While I appreciated many of the explanatory diagrams, other parts of the data presentation and analysis were extremely difficult to understand, and must be clarified for a general readership.

- The language describing kymograph generation was not understandable to me, though I tried very hard. “We draw a line ROI from the base that passed through the bouton reaching the edge and plotting this profile, encoding the intensity in grayscale level (or color for color images) for all slices in the Δt . To adjust the line position, we select a frame where the bouton is still small and draw a line ROI preferentially from the center through the bouton. However, sometimes this was not possible. In these cases, we used the max size frame as reference and draw a line ROI that extended from base to edge during bouton growth.”

What I did understand made the selection of base in particular seem subjective. Is the base at the point of origination of the new bouton (as seems to be the case in 1e, though the “b” end of the line seems arbitrary), or is it the opposite side of the pre-existing bouton (e.g. Fig 1d)? For the $1\mu\text{m} \times 1\mu\text{m}$ boxes used for quantification, it would make a huge difference if it were placed on a bright actin punctum or not. The authors should rewrite these sentences with attention to grammar and clarity, or draw a better diagram to describe how what they did was consistent from bouton to bouton.

- The authors use the label “distance” on the line scans (e.g. 1d’ and 1e’). Methods suggest that this represents “distance” along the time axis in the kymograph, which is an extremely confusing thing to call a time measurement (“In these plots, x-axis reports time (along the line that we draw) and y-axis is fluorescence intensity along this time window.”). Also, is it normalized since the times are different in different boutons? “Distance” in some figures (eg 3c’ and 7a’ is 10x larger than in 1d’, 1e’, 3d’). Please normalize similarly (if this is what was done) and change the label from "distance" to something like “normalized time”.

- The histograms in Figure 5 are really hard to interpret because there’s too much interleaved information. They should be separated or simplified.

5. The authors oversell the connection between this activity-dependent paradigm for structural plasticity and physiological synaptic growth. Indeed, many mutants with severe defects in acute plasticity (e.g. cortactin and synapsin) and muscle contraction (e.g. Syt1) have normal arbor size and bouton number in third instar larvae. The spaced stimulation paradigm is still very interesting and serves as an excellent system to understand synaptic signal transduction and membrane dynamics, but the authors should soften language throughout. For example, I think it’s still very interesting if the physiological relevance regards membrane tension and mechanical stress at the synapse, rather than actual physiological or activity-dependent bouton growth that would drive synaptic strength in a significant way. An example:

p21. “Without a bleb-like mechanism, we would predict a slower NMJ expansion, not always compatible with the requirements of the neuromuscular system during development or increased activity.”

We don’t know that the rate of bouton growth during development is fast. Further, many mutants that have reduced ghost bouton formation (e.g. Syt1, Synapsin) do not have undergrowth phenotypes, suggesting that these may not be entirely related processes.

6. On a similar note, while the reference list includes some of the previous literature on this stimulation-induced bouton growth, it is not enough to have a list of reference numbers following a broad statement (e.g. “Furthermore, combining these protocols with live imaging studies showed that activity-induced presynaptic morphogenesis may be different from the embryonic GC (Ataman 2009, Piccioli 2014)”, without saying in what way these papers showed they are different – indeed Ataman did suggest there were filopodia-like “synaptopods”). Further, the term that has been used to describe these new boutons for over a decade (“ghost boutons”) is not used at all in the manuscript. This will be confusing for the field, and the authors should use this term, and briefly tell the reader what is known about maturation of a fraction of these structures into functional boutons.

Overall the text does an inadequate job of actually discussing what is known, and putting the current manuscript in context. Neuronal activity, Ca²⁺-dependence, functional actin cytoskeleton and contractile muscle were all previously studied as prerequisites to induce ghost bouton formation. Since the ghost boutons are formed even with severed axons scientists suspected that there was a “local” effect at the synapse too. This does not detract from the novelty of the current study in characterizing blebbing as the mechanism for ghost bouton formation.

Some examples of missed context:

p.18. “We hypothesize that elevated activity, and in response to a still unclear signal to initiate bouton growth, MNs add new boutons by membrane blebbing and muscle contraction is required to increase neuronal confinement, and consequently cortical tension, which powers bouton outgrowth.” “In this scenario, as it has been proposed for cells in compressive 3-D environments⁶⁴, it is possible that a muscle derived signal, released in regions of higher activity, can polarize the recruitment of molecules required to or that facilitate bouton formation to spots of high membrane tension. While the molecular identity of the signal that leads to bouton initiation remains –to be elucidated, ...”

These statements do not take in to account extensive previous findings on bidirectional Wnt signaling (Ataman et al., 2008), Syt4 signaling (Korkut 2013), BMP signaling (Piccioli et al. 2014), integrins (Lee et al 2017), ECM (Dear 2017), CamKII (Nesler 2016) as well as cAMP/PKA-dependent pathway (Vasin et al., 2014), associated with ghost bouton formation, and therefore the findings described in the current manuscript sound (incorrectly) completely novel.

p 20. – “Another question that remains unanswered is related with the membrane source required for structural changes observed in blebs, or, in our case, for neuronal remodeling.”

There is massive synaptic vesicle fusion in the paradigms used in this manuscript, which should be mentioned as a likely “membrane source”, citing Vasin et al., 2014 who show that synapsin mutant NMJs, which have fewer synaptic vesicles, have reduced ghost boutons upon stimulation. The authors do cite here a subsequent synapsin paper with in vivo work, but not the relevant paper using the fillet stimulation paradigm.

Finally, San Martin 2017 is cited in the text but is absent from the references.

Minor concerns:

Methods and Tables

1. The authors state that they use the spaced stimulation protocol from Vasin 2014, and then “Additionally, for live imaging, we also used a mass stimulation procedure of 10- or 16-min High-K+ incubations developed by Martin et. (2017).” The authors must specify which paradigm was used

when (and also include the San Martin paper in their references – it is absent). Since this high-intensity stimulation is the essence of the manuscript, it is critical to clarify in each figure/legend which type was used.

2. It is unclear what “# experimental repeats” means in Table 4 or “biologically independent experiments” in the legends. For example, in the first row, were 13 NMJs from 13 animals imaged 10 times, for 130 independent points? This seems unlikely. What does the 10 refer to?

Figures

3. Not clear what “Pint” is meant to indicate in 1c.

4. Signals in magenta and green are often very different in intensity so it would be more informative to present normalized fluorescence intensity in the kymographs

5. The NASPM experiments lack a NASPM alone control (without stretching). What is the effect of this manipulation on its own, and do the authors expect a difference for acute inhibition compared to chronic GluRIIA mutants tested in Piccioli et al?

6. I would reorder figure 5 in the following way to match the text better:

- G-i to a-c
- D-f no change
- A-c to g-i

Text

7. p. 20 “Our live imaging with CD4-Tom revealed local intracellular vesicle dynamics in MNs at the place of bouton formation.” – This has not been quantified in the paper, and RFP-derived tags such as Tomato tend to aggregate. I suggest removing this statement.

8. p20 “Synaptic boutons have been consistently associated with axonal GCs and actin mediated protrusive processes, such as lamellipodia or filopodia, which are typically seen in cells migrating in 2-D.” This statement needs references or should be removed.

Response to Reviewers

We thank all the reviewers for the time and effort that you have dedicated to providing valuable feedback on our manuscript. We thank the constructive comments and insightful suggestions given by each of you to improve this study. We have made major revisions to our manuscript including the addition of many experiments and re-writing of the document as suggested, which we hope to address the points raised by each of the reviewers. Because all of you raised very important issues, we opted to take longer but to address all the concerns and suggestions raised as completely as possible. Additionally, we provide a document that shows all the changes made to the previous manuscript (named “Fernandes et al_Manuscript changes.pdf”) and another document containing the list and describing the full changes made to the previous figures (named “Fernandes et al_Figure changes.pdf”). Below is our point-by-point response. We hope you find your questions answered and think that the manuscript is ready for publication.

Reviewer #1 (Remarks to the Author):

Fernandes et al., 2021 Motor neuron boutons remodel through membrane blebbing

This manuscript makes the interesting proposal that synaptic bouton addition through a process they term ‘blebbing’ is intimately tied to muscular contraction and synaptic activity at the *Drosophila* larval NMJ terminal. Synaptic boutons are a fundamental organizational unit of both invertebrate and vertebrate peripheral and central synaptic terminals so understanding the mechanisms that govern their development is intrinsically important. However, the manuscript does not study normal bouton development, but rather ‘ghost boutons’, an artificially induced phenomenon whose relationship to normal bouton development is unclear and could be argued to be a pathological phenomena. While the authors may argue, as other studies have done, that ghost boutons could be perhaps be precursors of ‘real boutons’, without definitive establishment that this is actually the case, the import and findings of the manuscript is greatly diminished.

We thank the reviewer for finding our study potentially important to the field. Before we answer the specific concerns raised, we would like to clarify that membrane blebbing is not our terming, it is a well-characterized and widely studied mechanism in the fields of cell migration and cancer invasiveness, as mentioned in the manuscript (Fackler & Grosse, 2002; Charras & Paluch, 2008; Chick & Raz, 2020). Blebbing is characterized by being pressure-driven rather than actin polymerization-based, with some cells being able to alternate between blebbing and lamellipodia-dependent migration, depending on the 3-D environment to which they are exposed. There is an active debate in the field of migration on whether every cell type can engage this mechanism of

blebbing or whether only some cells are “bleb-competent” - a debate that has profound implications on the prediction of how cells adapt to the external cues to which they are exposed at each moment. Our study is the first to report membrane blebbing in neurons in the context of development and structural activity-dependent plasticity.

Regarding our approach to study bouton addition, we do understand the criticism regarding ghost boutons. Below, we present our arguments and data to counterpoint this issue and hope that you are convinced that ghost boutons represent physiologically relevant structures. In addition to what has been shown in studies like Ataman *et al.* (2008), Fuentes-Medel *et al.* (2009), Piccioli & Littleton (2014), amongst others, we conducted a series of new experiments to further support the physiological relevance of ghost boutons. Below is a summary of the experiments we performed and included in the paper:

1. We performed live imaging of the NMJs under unstimulated conditions (at rest) (**Fig. 1d**) and observed new bouton formation with the same morphological features as those induced by high-K⁺ stimulation, although with lower frequency. The fact that it was rare to observe live events in unstimulated NMJs is the main reason why we chose to study bouton formation after high-K⁺ stimulation rather than performing our analysis without stimulation. However, we agree that clarifying whether new bouton formation under unstimulated conditions resembles bouton formation after high-K⁺ stimulation is an important point. Importantly, we showed that even in unstimulated NMJs new boutons formed with low actin and therefore had the hallmark of blebs (**Figs. 2h,i and Supplementary Fig.8a**).
2. We used optogenetics to induce bouton formation as an alternative method to K⁺ stim and observed the same morphological features (with slightly slower kinetics, still compatible with blebbing) (**Supplementary Fig. 3**). Unfortunately, the expression of the channelrhodopsin Chrimson (red light activated channelrhodopsin) is lethal when expressed in all neurons (NSyb-Gal4), even in the absence of light, and slightly noxious if expressed in MNs (with OK6-Gal4) – but we did analyze optogenetics-induced bouton formation with OK6-Gal4. We also noted that the m6/7 NMJs appeared to have smaller boutons, suggesting that Chrimson expression may interfere with normal development. Nevertheless, using optogenetics, we were able to induce new bouton formation in both live and fixed conditions after stimulation, which emerged as round protrusions. On the other hand, it is possible that the observed slower kinetics might be due to the toxicity we observed upon Chrimson expression.
3. We dissected larvae from early 2nd instar onwards (early, midi and late 2nd instar; early mid and late 3rd instar) that have not been subjected to any kind of stimulation and showed that ghost boutons are naturally present throughout normal larvae development, although at a lower

number than what we observe after stimulation with high-K⁺ or optogenetics, as expected (**Supplementary Fig. 4**).

4. Finally, to show that ghost boutons induced by patterned depolarization with K⁺ have the potential for maturation (**Fig. 1g-h**), we analyzed *ex vivo* dissected NMJs and followed pre- and post- synaptic maturation markers (Brp, GluRIIE, and Dlg) by performing long term time-lapse imaging after stimulation. Our data showed that ghost boutons formed by K⁺ stim can acquire these maturation markers even *ex vivo*, suggesting that at least a fraction of ghost boutons has the potential to mature and become functional, as suggested *in vivo* by Ataman *et al.* (2008) and Vasin *et al.* (2019) that showed ghost boutons have maturation potential in intact larvae unstimulated or in response to intense crawling activity.

Altogether, these data and other published data are consistent with a physiological role for ghost boutons, most likely as precursors of mature synaptic boutons, and therefore, a mechanism worthy of study.

Major issues.

1. Artificially induced ghost boutons may bear no relationship to normal bouton formation. The authors interchangeably and incorrectly confuse terminology between 'boutons' and 'ghost boutons'.

We appreciate the comment of the reviewer, and we agree that the previous version of the manuscript was incorrect. In this new version, to avoid confusion, we use the term ghost bouton throughout the manuscript and explain what they are.

Synaptic boutons develop pre and postsynaptic specialisations including active zones for synaptic vesicle release, discrete accumulations of synaptic vesicles, postsynaptic glutamate receptor puncta and 'SSR' elaborations. In contrast, 'ghost boutons' induced by KCL or other treatments have not been shown to produce any of these structures. The blebbing the authors study appears to occur only in ghost boutons which do not show many of the features of normal boutons outlined above suggesting they are not functional and could be an aberrant artificial phenomena. If the manuscripts ultimate findings are that induced extreme hyperactivity causes an increase in non-functional membrane elaborations that never progress to become functional participants in neurotransmission, the relevance is not clear. To address the issue the authors should aim to develop a preparation where normal bouton development can be examined and manipulated. This seems challenging *in vivo*, but perhaps an *ex-vivo* preparation could be used, with a number of solutions that allow long-term *ex-vivo* maintenance of larval preparations (e.g.

HL6 etc.) that might allow normal bouton addition to progress and be manipulated. Without data of this nature, the manuscript findings would need to be specifically (and less interestingly) restricted ghost boutons.

We have briefly described above the series of experiments that we have performed to answer these concerns. Specifically, regarding the physiological relevance of ghost boutons and the use of high-K⁺, which can be considered an extreme method, to induce formation of ghost boutons. We provide below a more comprehensive description of what we did, why, the challenges encountered and the conclusions that we obtained. Altogether, our data (and data from other labs) suggests that ghost boutons have the characteristics to be functional units, with the capacity to exo- and endocytosis being good examples of this as shown by Vasin *et al.* (2019).

1) Ghost bouton maturation

As suggested by the reviewer, we have performed experiments to assess the potential of ghost boutons to acquire maturation markers. We chose to image the active zone marker BRP and postsynaptic markers GluRIIE and DLG. For this, we crossed nSyb-Gal4,UAS-CD4-Tom with UAS-BRP-GFP or with a protein trap line - GluRIIE-GFP or DLG-GFP (FlyFos stocks from VDRC). We performed stimulation with high-K⁺ and imaged the same NMJs at different time points after stimulation. In **Fig. 1g-h**, we show examples of ghost bouton maturation with the 3 markers. We provide examples of ghost boutons that acquire the presynaptic marker BRP and the postsynaptic markers GluRIIE and DLG. The maturation of these markers could be clearly observed from 5h up to 10h post stimulation.

For this set of experiments, we performed high-K⁺ stimulation in HL3.1, as we normally do, but, because HL3.1 is not suitable for long term imaging (Feng *et al.*, 2004), we switched the media after the stimulation to HL6, as suggested by the reviewer. However, in HL6, we observed that the preparations were viable for a maximum of 5h (which coincides with the time reported to be viable for HL6 in Macleod *et al.*, 2002). To prolong the viability of the preparation, we searched the literature and found that *Drosophila* brains could be maintained in Schneider's medium for up to 48h (Rabinovich *et al.* 2015). This medium has also been used in MN axons (Louie *et al.*, 2008), and we therefore decided to test it at the NMJ. With Schneider's media, we could keep the preparation viable for ~10h and in a few cases, we were able to keep the preparation viable for up to 30h, but this was a rare scenario. Unfortunately, even in these media, which are more suitable for long term imaging, the viability of the preparation was significantly variable after 3h post dissection. With this variability, we cannot quantify time of maturation or % of boutons that mature,

but we are confident to say that a fraction of them does mature as some examples shown in the figure (even *ex vivo*), again supporting that these ghost boutons can become functional.

2) The nature of high-K⁺ stimulation and whether ghost boutons represent “an aberrant artificial phenomena”

While we do understand the point raised, our data supports the hypothesis that ghosts boutons are not artificial and have a physiological role (**Fig. 1d-g, Figs. 2h,i, Supplementary Fig. 3, Supplementary Fig. 4, Supplementary Fig. 8**). In addition, we want to highlight that stimulation with high-K⁺ has been successfully used in other systems, showing that high-K⁺ treatment elevates activity globally, but transiently, resulting in the formation of new boutons and new spines, both in mammalian neuronal cultures and in *Drosophila* (Wu *et al.* 2001; Hu *et al.*, 2008; Cai *et al.*, 2007; Li *et al.*, 2005; Yao & Chen, 2006, Maldonado-Díaz *et al.*, 2021). The formation of these ghost boutons by high-K⁺ has been shown to be regulated by genetic pathways involved in bouton and spine formation induced by other methods, with major pathways including both Wnt (Ataman *et al.* 2008; Alicea *et al.* 2017; Gogolla *et al.*, 2009; Tabatadze *et al.*, 2014) and BMP signaling (Piccioli & Littleton, 2014; Withers *et al.* 2000; Horbinski *et al.*, 2001; Lee-Hoeflich *et al.*, 2004), being also involved in developmental synaptic growth (Budnik & Salinas, 2011; Chou & Vactor, 2020). Altogether, these data support that “high-K⁺ stimulation” is a suitable method to easily induce activity-dependent processes.

Having said this, we do agree that high-K⁺ depolarization is an extreme method to increase activity levels, but it allows the induction of activity-dependent pathways easily and reproducibly. Furthermore, in our experiments, most genotypes already carry several genetically encoded genes, making the addition of other constructs to manipulate activity by other means (such as optogenetics) difficult. Also, an excessive load of transgenic constructs present in the parental lines can impact the viability of the parents and progeny.

Below, is a summary of the experiments we performed to answer the criticisms raised concerning the high-K⁺ methodology.

i) Ghost boutons in unstimulated/more physiological conditions *in vivo* and in fixed conditions:

To visualize ghost bouton formation using different methodologies, we used three approaches: 1) live imaging of ghost boutons in unstimulated preparations, 2) optogenetic induction of ghost boutons in whole intact larvae, and 3) optogenetic induction of ghost boutons in semi-dissected larvae, a condition methodologically closer to the high-K⁺ stimulation method

(but possibly more physiological). All these conditions (as mentioned earlier) had the morphological features that support the claim that ghost boutons induced by other means or in unstimulated conditions can be observed and resemble blebs.

ii) Ghost boutons throughout development.

The existence of ghost boutons throughout development (even though in low numbers) indicates that these structures are part of the normal physiology of the neuromuscular junction and that they likely represent precursors of boutons, as suggested by Ataman *et al.* (2008). Concerning bouton genesis, we do not claim that all boutons are formed by blebbing. It is possible that high activity induces the formation of excessive boutons, from which only a fraction matures. It will be very interesting to test whether the boutons that mature are the ones that acquire actin post-stimulation, for example. Also, excess formation of synaptic structures to later be pruned is not an uncommon phenomenon in neuronal development, and a part of ghost boutons can be eliminated through the mechanism previously described by the Freeman lab (Fuentes-Mendel *et al.* 2009). Whether all ghost boutons are “born” identical, and are destined to mature or to be pruned, is a question that remains unanswered and will be an exciting question for the future.

2. Analysis of ghost bouton formation. Although stimulation method the authors used has previously been published on several occasions, they should recognise their preparation is highly artificial due to (i) physiological saline composition and (ii) methodology of stimulation. I would like to see direct interrogation that the bouton addition they observe is not at least in part attributable to their methodological approach and clearly presented. Suggested experiments would include – (a) Observation of the blebbing process at NMJs without addition of KCl and report if they observe similar or lower levels of blebbing – this data is presented in fig.4 as a consequence of another manipulation but would be better tackled directly for validation of the technique in their conditions. (b) Quantify blebbing at incrementally increasing external Ca^{2+} saline concentrations (from 0 mM to physiological levels $\sim 1.5\text{mM}$). If blebbing is a result of muscle contraction due to increased synaptic activity, the authors should observe no or heavily reduced blebbing in Ca^{2+} free saline which should increase with increased Ca^{2+} concentration (c) KCl addition is a far more extreme stimulation than what would be experienced through endogenous motor activity I would suggest authors ideally visualise blebbing through alternative approach, either direct nerve stimulation in line with parameters for known endogenous activity or through pulsed ChR stimulation, live or analysed post hoc to confirm increased blebbing with alternative stimulation parameters.

We thank the reviewer for suggesting these experiments. Some of the points raised here in (a) and (c) were already addressed in the previous point (blebbing in unstimulated conditions and other methodologies to induce bouton formation), so we consider it answered.

Regarding point b), we performed a stimulation experiment at incrementally increasing Ca^{2+} concentrations (0mM, 0.5mM, 1mM and 1.5mM Ca^{2+}) and observed progressively higher levels of ghost bouton formation with increasing Ca^{2+} (**Figs. 7d,e**). As written in the new version of the manuscript: “ We observed a pronounced reduction in GB formation in Ca^{2+} -free saline, as has been previously shown¹, and here we show that both GB number and size steadily increase with incrementally higher Ca^{2+} concentrations (**Figs. 7d-f**), concomitant with higher levels of both neuronal and muscle activity”. In other words, our results show that ghost bouton formation is regulated by synaptic activity and coupled with muscle contraction.

3. Stretching assays are imprecise. The classification of semi stretched, stretched and relaxed is very subjective in my opinion. Can the authors be sure that excessive stretching does not cause ruptures to muscles and thus changes in membrane resistance? How did they ensure that the stretching force is the same across samples? Tools such as GtACR1 or GtACR2 could be used to express in muscles to counteract muscle contractions to independently validate their findings. Do the authors observe any of the reported effects if the cuticle is also stretched?

We agree with this criticism and have now quantified the effects of muscle stretching on contraction efficiency. Below, we summarize the new experiments and quantifications performed, which we trust to address the concerns raised regarding our methodology. The quantifications are shown in **Fig. 6a-d** and **Supplementary Fig. 25**. Briefly, we expressed tropomyosin-GFP in the muscle and performed live imaging of dissected larvae under different stretching and stimulation conditions: 1) relaxed and unstimulated, 2) relaxed and stimulated, and 3) stretched and stimulated (**Fig. 6a**). Details regarding this assay are included in the Methods section. For each condition, we plotted the length of ventral muscle 6 in segment A3 over time, showing that mechanical stretching reduced the amplitude of the muscle contractions and prevented rhythmic contractions (**Fig. 6b**). Additionally, from all animals analyzed, we quantified the mean and maximal contractility levels within the same muscle, with both parameters significantly decreased by larval stretching (**Figs. 6c,d**). The movies and respective quantifications clearly show that this method effectively reduces muscle contraction. To ensure that muscle integrity was preserved in these conditions, we stained the preparations with phalloidin and α -actinin, which allowed visualization of the muscle filamentous actin and z-bands, respectively. Our results showed that the muscles remain intact after manipulations were performed (**Supplementary Fig. 25**).

Regarding the use of optogenetics to prevent muscle contraction, we inquired the authors of the first two papers describing the use of GtACR for neuronal inactivation (Alex Mauss and Farhan Mohd) regarding the possibility that this tool could be used to prevent muscles contraction. While both did not test this possibility directly, they indicated a report that shows that expression of GtACR in the fly cardiac muscle could prevent its contraction (Stanley *et al.* 2019). With this in mind we tested the expression of GtACR1 under the control of a strong muscle driver (mhc-LexA:LHV2, LexAop-GtACR1) in the presence of All Trans Retinal. However, exposure to strong green light did not prevent movement of larvae indicating the inability of this tool to prevent muscle contraction.

Regarding the cuticle being stretched, since the larvae are pinned or glued (live imaging) onto the sylgard, the cuticle is also stretched in our setup. We are not aware of a method in *Drosophila* that allows the stretching of muscles independent of the cuticle.

Additional issues.

Figure 2. Page 10 – line 2 onwards . Authors appear to blame high variability for a lack of effect in JAS treatment on new bouton number. However, in fig.2j, HL3 saline has the highest variability, I would suggest they remove misleading interpretation of data ('although we could observe fewer boutons, we also observed unaffected boutons') this is likely just a description of biological variability in their system and should not be confused with any statistical comparison or real effect.

We removed this sentence and only reported what the statistics show. (Now it reads: "Conversely, treating MNs with JAS, revealed that GB formation after high-K⁺ stimulation was not different from controls (**Figs. 3h,i and Supplementary Fig.13b**). However, we noticed a conspicuous phenotype in the NMJs that responded to the stimulation, which was the presence of very small GBs, usually clustered in discrete regions. While JAS did not change GB frequency after stimulation, we found that GB size was smaller after JAS treatment").

Figure 3. It is not intuitive to me why or how the authors chose to study muscle myosin ii despite reading their rational, the logic seems counterintuitive to the most interesting aspect to the paper – muscle force mediated bouton addition. Regardless, fig 4d shows bouton addition occurs when unstimulated, do they see similar myosin II localisation when boutons form without being stimulated? Visualisation of a similar mechanism when unstimulated will help address the above major critique. Furthermore – would the authors comment on why blebbing is observed when not stimulated and muscles are not contracting?

We apologize for not being clear on why we looked at non-muscle myosin-II (MyoII). We re-wrote the text to make it more clear.

Regarding the specific points raised:

- To answer the question regarding MyoII localization, we now show that ghost boutons from unstimulated and stimulated preparations have similar MyoII localization (**Supplementary Fig. 14 and 15**): MyoII displayed accumulations (suggestive of activation) at the neck and in the cortex. This indicates that MyoII is involved in both types of bouton formation (stimulated and unstimulated).
- To develop on the role of MyoII in this process, our model does not exclude that MyoII participates directly in bouton formation. What we observe is that bouton formation still occurs in conditions of lower MyoII, promoted mostly by muscle contraction around the weakened MN membrane. Interestingly, as it can be seen in **Fig. 5e**, when MyoII is reduced in neurons, we do not observe slowly forming boutons supporting that the boutons that take longer to grow may engage MyoII presynaptically and rely less on muscle-neuron interplay (or it can be a combination of both mechanisms). The main idea is that bouton formation is a pressure-driven mechanism, and the pressure can originate either from muscle or from neuron, or from both. Furthermore, as indicated in the first point, MyoII localized to ghost boutons even without stimulation, further strengthening that MyoII can be engaged in bouton formation in conditions of reduced muscle contraction.
- Regarding the imaging in unstimulated conditions (just membrane or membrane and actin), we observed bouton formation with the presence of a self-generated muscle contraction event, but also observed cases with no visible contraction – both scenarios also observed after stimulation. This suggests that in the dissected animals, there are also self-generated signals to induce muscle contraction (we keep the brain attached). When no visible muscle contraction is observed prior to bouton formation, we hypothesize that 1) a subtle muscle contraction occurred which we were not able to identify, 2) neurons can cell autonomously also provide sufficient internal pressure for bouton formation, or 3) glia can constrict around boutons, therefore providing confinement in ways that can resemble the role the muscle plays (we have movies with labeled glia from another project that may support this idea, but this needs to be extensively tested).

Figure 4. As a further control for SqhDN and SqhCA experiments, authors should upregulate WT Sqh for comparison, expression level and presence of a mutant are confounding in some experiments. Has RNAi-sqh been published before? If not, the efficiency and specificity should be shown, e.g. with a western blot.

We thank the reviewer for these suggestions. Regarding the Sqh-WT, we could not find a publication or available stock that was a UAS-Sqh-WT (tagged or untagged). There are traps and the DN e CA, but no UAS-Sqh-WT (at least to the best of our knowledge), which precluded this experiment. Regarding the RNAi line used: yes, it has been validated before (Nie *et al.*, 2014) and shown to effectively reduce Sqh protein levels, and also to mimic the mutant phenotype in the eye - we now added this reference to the text. In addition, to confirm that the effect we observed with Sqh-IR, Zip-IR and Sqh-DN was specific, we tested other RNAi lines for Sqh and Zip, which confirmed our previous results (**Supplementary Fig. 20**). Furthermore, we stained the NMJ with anti-Zip antibody in control and Neuronal Zip-IR, which showed reduced Zip levels in ghost boutons (**Supplementary Fig. 19**).

Figure 5. Philanthotoxin is more commonly used to block drosophila GluRs, can authors rationalise the use of NASPM vs. PhTx. As the authors note, inhibiting GluRs is not an ideal way to neutralize muscle contractions because glutamate receptor inhibition has been shown to inhibit ghost bouton formation through other mechanisms (see Ref 21 in the manuscript) i.e. by affecting the retrograde signaling from the muscle.

To block muscle contraction without interfering with the neurons directly, we considered all drugs known to block GluRs (and already validated in *Drosophila*) and identified Philanthotoxin (PhTX) and NASPM. PhTX is a drug used for GluR inhibition, which acts by reducing mEPSP amplitude and quantal size, thereby lowering synaptic strength. However, it was shown that NMJs can quickly adapt following acute blockade with PhTX (Davis GW, 2013) due to a homeostatic increase in presynaptic NT release (quantal content). Hence, with PhTX, the reduction in postsynaptic excitability is sensed in the muscle and transduced into a retrograde signal that leads to enhancement in presynaptic release maintaining stable the synaptic strength. On the contrary, NASPM does not adapt (Han *et al.*, 2015) and we therefore chose it, since its pharmacokinetics are more compatible with the duration of our experiments, which last ~1h.

Text. The logical flow of the manuscript could also be improved. For example, it is confusing for the reader to understand whether actin is or is not involved.

We appreciate the suggestion and did major changes to the manuscript to improve it, reducing potential confusion. We also described better the role of actin in ghost bouton formation and remodeling. In summary, our model is that actin polymerization is not required for the expansion phase of ghost bouton formation but is required for bouton stabilization. As such, most boutons form with low actin, and interfering with actin polymerization alters bouton size rather than frequency. We also show that actin polymerization is not the driving force for bouton outgrowth, which clearly is different from other protrusion previously described in neurons, such as growth cones and dendritic filopodia.

Reviewer #2 (Remarks to the Author):

In this paper, Fernandes and co-workers describe the mechanism by which motoneuron (MN)-muscle interactions generate new boutons in the *Drosophila* larval nervous system, allowing MNs to form membrane blebbing as a strategy for local remodeling of presynaptic structures. The authors use a combination of genetics and microscopic approaches to explore bouton genesis at the NMJ of muscles 6/7. They report that an amount of F-actin is relatively low inside new boutons, suggesting that actin polymerization is not essential for bouton formation. They argue in addition that the recruitment of myosin-II to new boutons slows down the growth of new boutons by classical genetic analysis (albeit primary with RNAi and gain-of-function analyses). They further argue that mechanical forces from muscle contraction promote bouton formation at the NMJ. However, the data supporting this contention is not compelling.

I think this paper provides an essential step towards the mechanistic understanding of bouton genesis. However, it requires additional experiments to provide insight into the molecular mechanisms regulating bouton genesis. I have listed my specific concerns about this paper below.

We thank again the reviewer for the nice words regarding the importance of our manuscript and for the helpful comments regarding our findings, and suggestions for additional experiments to support our hypothesis. Our responses to the comments are provided below in a point-by-point format. We hope our answer satisfy your concerns and you find our study suitable for publication.

Specific comments:

[Major comments]

1. Introduction: Given that the trans-synaptic forces mediate synaptic changes (refs 23, 28-35), this must be intensively discussed in the introduction. The findings in the references established through genetics and imaging analyses that a variety of cellular proteins are involved in bouton genesis. Fernandes et al. extend these works to the molecular function of actomyosin networks and make significant contributions towards understanding bouton formation.

We apologize for having not introduced better the role of transsynaptic forces in synaptic remodeling. We now discuss this topic in more depth in both the Introduction and Discussion

sections. Furthermore, we agree that this information will help strengthen how this study represents an important advance in the field of neuroplasticity - increasing *in vivo* evidence that mechanical force is a functional regulator of activity-dependent plasticity, in addition to biochemical signaling.

2. Page 6. The authors point out the potential involvement of a budding mechanism. A quick google search shows an intriguing paper (PMID: 12160747). In the paper, Pennetta et al. have demonstrated that microtubules play an essential role in initiating the budding of boutons. Because there has been an alternative scenario, the authors need to discuss and ideally test it experimentally.

We recognize the relevance of the point identified by the reviewer. It is a great question, and a subject that we want to investigate further.

The role of microtubules (MTs) in bouton formation is a very exciting question that we aim to test. However, a definite answer to this question would require an extensive repertoire of experiments, which would probably be sufficient for an additional publication. Nevertheless, since we present a scenario that does not account for MTs, and MTs have been implicated in bouton outgrowth, we have performed some experiments to address the reviewer's comment, which we will discuss here in more detail. We expressed EB1-GFP, a MT + tip binding protein, under the control of NSyb-Gal4, stimulated these larvae, and checked whether ghost boutons had EB1-GFP, which we used as a marker for MT growth. In a complimentary experiment, we stimulated wild-type larvae and stained the NMJs for Futsch, a MT-binding protein, associated with stable MTs. We observed that 100% of the ghost boutons had EB1-GFP, and most had Futsch (as splattered structures or filamentous, but not Futsch loops typical of mature synaptic boutons), suggesting that, unlike what we observed for actin, ghost boutons are formed with MTs. Additionally, since Futsch has been implicated in MT stabilization and EB1 modulates MTs dynamics, the loss of Futsch-positive loops, together with EB1 enrichment in ghost boutons, suggests that their formation is associated with the formation of dynamic MTs. These results are presented in the images below (**Reviewer Figure 1**).

Reviewer Figure 1. Legend: a, b, Images of synaptic terminals stained for MT-associated proteins, Futsch (a) and EB1 (b). Scale bar, 10 μm . c, d Zoom of examples of ghost boutons (white arrows) highlighting loss of Futsch typical organization (c) and EB1 recruitment (d). e, Confocal image of examples of Futsch staining to reveal MT loops (arrows in Futsch channel) inside mature synaptic boutons. Neuronal membranes were stained with anti-HRP and postsynaptic membranes with anti-Dlg antibodies. GBs were identified by the lack of Dlg. Scale bar, 2 μm .

Furthermore, we discuss here the findings from Pennetta *et al.* (2002) and how those results fit our model. Worthy note, Pennetta *et al.* (2002) does not show directly that MTs are involved in bouton formation. Instead, the authors showed that DVAP-33A (a MT associated protein) was present at the membrane and regulated developmental bouton addition possibly by stabilizing and directing MTs towards boutons. Nevertheless, this model suggested a role for MT during bouton formation, which is in accordance with our observation that new boutons (ghost boutons) induced by acute activity are enriched in MTs, which we confirmed with α -tubulin staining (see **Reviewer Figure 2**, below). However, our data (Futsch and EB1 stainings) suggests that MTs present in these boutons

are not stable, but dynamic, something that Pennetta *et al.* could not directly address since there was no live imaging in their study.

Reviewer Figure 2. Legend: Representative images showing NMJ 6/7 arbor stained with anti- α -tubulin (green) in WT (w1118) larvae. Scale bar, 10 μ M. Below we show Zoom of GBs examples highlighting tubulin aggregates.

Considering that MTs are found in both types of boutons, we question whether MTs share a common role during formation of developmental and activity-dependent boutons. Specifically, one hypothesis is that DVAP-33A role in developmental bouton formation is to fine-tune MT-membrane interactions, allowing first labile interactions and MT growth to initiate bouton expansion, and then, after boutons have formed, stabilizing MTs at the membrane. Supporting this, Pennetta *et al.* (2002) showed that modulation of DVAP-33A levels affects both bouton number and size, although in an inverse manner, with loss of DVPA-33A causing a severe decrease in the number of boutons but a corresponding increase in bouton size, whereas presynaptic overexpression of DVAP-33A induced an increase in the number of boutons and a decrease in their size.

Our results suggest that growing MTs are recruited to the ghost boutons or required for their outgrowth, which we have shown to be *bona fide* blebs. Also, in spreading cells, MTs have been shown to enter large blebs (Tvorogova and Vorobjev, 2013), and MT depolymerizing agents - such as nocodazole - are able to induce blebbing (Tvorogova et al. 2018), although the mechanism regulating blebbing is largely unknown. Overall, these findings suggest that in addition to other

major factors regulating blebbing, such as local actin depolymerization and increased pressure, dynamic MTs are also likely to contribute to bleb formation. Thus, our model of bouton formation is compatible with a role of MTs. Nevertheless, while a key role of reorganization of both actin and MT cytoskeletons in bouton formation is becoming increasingly clear, how different steps of bouton formation are coordinated in space and time by actin-microtubule interactions are not understood and should therefore be addressed by future studies. For instance, it will be interesting to investigate whether MT growth can limit the polymerization of the actin cytoskeleton (as it has been suggested in other cellular processes - Chen et al., 2021) at the initial stages of bouton formation.

We want to emphasize that we consider this debate one of the most exciting next steps.

3. Page 6. Can the F-actin distribution along the length of boutons (Fig 1 d-e) be quantified with a couple of more examples?

As suggested by the reviewer, we have provided an example of F-actin distribution in boutons formed with stimulation in **Supplementary Fig. 7**, in addition to the two examples presented in the main **Fig.2** (b and c). We have the quantification of all data in **Figs. 2f and 2g**. Furthermore, we also show examples of actin in boutons formed without stimulation (resting conditions) in **Fig. 2h and in Supplementary Fig. 8a**.

4. Page 6, 7. Because the authors state “(blebbing) is driven by a flow of cytosolic fluid across a local weakening (rupture or detachment) in the cell actomyosin cortex,” they should provide examples of boutons that show rupture or detachment in the actomyosin cortex using a higher resolution imaging approach.

We appreciate this suggestion and recognize that this would be a very important piece of data to present. However, the detachment or rupture of the actomyosin cortex is too small to be detected by super resolution light microscopy. This detachment, to our knowledge, can only be observed by electron microscopy. Using Platinum Replica Electron Microscopy (PREM), Chikina *et al.*, 2019 proposed gaps in the actin network as suitable sites for bleb formation. We thought about this before, but one of the biggest challenges is the fact that boutons are significantly smaller than cells. In addition, ghost boutons form at unpredictable locations, emerging from a large neuronal area, and it is likely that the local weakening from which the bouton will emerge is transient, and we may not catch it in fixed preparations. And, even if we were able to find such examples, we could not identify the precise localization from which these ghost boutons emerged, making this type of analysis very difficult. Still, we imaged F-actin in ghost boutons by super resolution microscopy (**Reviewer Figure 3**, below) to check if the region from which the bouton originated

could be clearly detected. But, as predicted, we could only hint at potential nucleation sites and further analysis was not reasonable.

Reviewer Figure 3. Figure Legend: a, Representative image of actin organization at *Drosophila* larval NMJ synaptic terminal. Super resolution confocal images (Zeiss Airyscan) of NMJ 6/7 of transgenic larvae expressing UAS-Lifeact-GFP (F-actin reporter) in MNs under the control of NSyb-Gal4. Animals were co-labeled with antibodies against HRP (neuronal membrane, gray), GFP (to enhance the Lifeact-GFP signal, magenta), and Dlg (postsynaptic membrane, cyan). Ghost boutons (GBs) were identified by the lack of Dlg. Scale bar, 10 μ m. a', b, Zoom of GBs examples highlighting the cortical actin cytoskeleton structure. Arrows point to the GBs base and potential nucleation sites at the NMJ terminal.

5. Page 7. What are the controls of overexpression of Lifeact-Ruby? Where is the transgene inserted, and what control transgenes are used? These should be shown at least in Supplementary material. Levels of Lifeact's expression need to be carefully calibrated by proper controls; however, data were not presented to establish this point.

We agree with the need for controls regarding the effect of overexpressing Lifeact-Ruby, which we have now added in **Supplementary Fig. 5**. Briefly, we showed that overexpressing Lifeact-Ruby with different strengths under different neuronal Gal4s does not change the degree of plasticity after high-K⁺ stimulation when compared to WT animals, suggesting that overexpression of Lifeact does not affect ghost bouton formation.

The main reason why we initially chose NSyb-Gal4 to perform our experiments is because it is a strong driver that allows for better visualization of boutons during imaging. Furthermore, the experiment with Lifeact was designed to test if boutons had the hallmark of blebs: little or no actin, and using a strong driver guaranteed that if we could not see Lifeact in ghost boutons, it was not because the levels were too low. Having confirmed that Gal4 lines expressing different levels of Lifeact displayed similar frequencies of ghost bouton formation compared to WT larvae, we decided to keep using NSyb-Gal4, the strongest driver tested to ensure that if actin was low (as we verified being the case), it was not due to having low detection capacity from a low expressing line.

6. Page 9. Please comment on the localization of actin puncta with JAS (Fig 2a). Furthermore, argue how those are correlated with the position of blebbing sites.

The actin puncta before and after JAS treatment, previously shown in Fig.2a, are now shown in **Fig.3b**. After JAS application, F-actin puncta were more intense, appeared mainly at the same location as before drug application, most frequently in branches compared to the main arbor. Sometimes new puncta emerged in regions in the vicinity of more intense puncta. However, from these experiments, we cannot discuss how F-actin puncta resulting from JAS are correlated with the blebbing sites.

To address this question, we used NSyb-Gal4 to express Lifeact-GFP and looked for ghost boutons formed after high-K⁺ stimulation in the presence of JAS, which we present below (**Reviewer Figure 4**). Like in our live imaging experiments, we observed the formation of boutons with or without F-actin puncta at their base. However, we found some examples of intense F-actin puncta (white arrows), sometimes displaying a clustering pattern, at the base of small boutons, which quite often contained F-actin. From the pool of small boutons without any F-actin inside,

many were detached from the main arbor. Importantly, F-actin puncta/clusters were not found near bigger boutons, which, although rare, appeared in regions where the actin cytoskeleton was weaker (yellow arrows). This experiment supports our hypothesis that JAS treatment leads to the formation of smaller boutons by increasing/stabilizing F-actin levels at these regions (**Figs. 3h,j and Supplementary Figs. 13d,e**).

Reviewer Figure 4. Legend: NMJs were co-labeled with antibodies against HRP (neuronal membrane, gray), GFP (to enhance the Lifeact-GFP signal, magenta), and Dlg (postsynaptic membrane, cyan). Ghost boutons (GBs) were identified by the lack of Dlg. Arrows point to the actin at GBs base.

7. Page 13. The authors state, “we opted to use RNAi to decrease ... myosin-II levels”. Are there data establishing that the protein level is reduced in these neurons?

Yes, the line we used was previously validated. However, mistakenly, we did not include this information in the previous version of the manuscript. We have added this information to the revised manuscript. Additionally, we show in **Supplementary Fig. 20** that myosin-II clustering (stained by Zip antibody) in ghost boutons was no longer found with RNAi directed against myosin-II heavy subunit (Zip). Furthermore, we have tested other RNAi lines for each subunit of myosin-II, which confirmed our previous findings (**Supplementary Fig. 19**).

8. Page 13. An essential complement to the RNAi analysis would have been assessing the requirement of myosin-II with loss-of-function mutants. The author should use the MARCM system to study the function of myosin-II in these neurons.

We appreciate this suggestion and agree that MARCM analysis would be an excellent complement to our study. However, given that clones are inherently variable in number and position, and that we do not know the degree of ghost bouton formation in other NMJs (all the quantifications in the manuscript are taken from m6/7), it would be difficult to obtain quantitative information from MARCM analysis of MN clones. Nevertheless, we wanted to try to perform these experiments by ordering the FRTsqh stock from Bloomington (BL25712) to establish the stocks for MARCM. Unfortunately, this line arrived dead several times, was not shipped in another order, and arrived with only a few flies in late February 2022. At this point, we faced serious problems keeping this stock alive. Subsequently, only the amplification of the stock and making of the MARCM stocks would take a few more months, which would then have to be followed by optimization of the generation of MARCM clones at the NMJ (which, contrary to other cell types/tissues, is not commonly used at the NMJ, at least to our knowledge), and its live imaging. In summary, we regret to tell you that we did not succeed at performing Sqh MARCM clones and analysis (and the FRTSqh flies are still sick and not successfully amplified).

However, we performed experiments to assure that myosin-II knockdown is reproducible. We added additional RNAi lines for Sqh and for Zip (Supplementary Fig. 20). The results obtained were consistent with those of our initial analysis (but of course we cannot be sure if null mutations behave differently). We have added in the discussion a point addressing the validity of our conclusions for the reduction of myosin-II could not be extended to a knock-out scenario.

9. Page 16, 17, 18. The addition of physical confinement to MNs is a critical experiment. A more detailed assessment of the frequency, strength, and duration of muscle contractions should be included to confirm that the effect is direct rather than indirect.

We agree with the reviewer's point. To provide a more accurate assessment of contractile activity with muscle contraction blockade, we now show representative contraction kymographs obtained from live larval fillets that were successively subjected to different stretching and stimulation conditions, which is present in Fig.6: 1) relaxed and unstimulated, 2) relaxed and stimulated, 3) stretched and stimulated. Details regarding this assay are included in the Methods section. Briefly, for each condition we plotted the length of ventral muscle 6 in segment A3 over time, showing that mechanical stretching reduced the amplitude of the muscle contractions and prevented rhythmic contractions. Additionally, from all animals analyzed, we quantified mean and maximal contractility

levels within the same muscle, with both parameters being significantly decreased by larval stretching.

It is important to refer that during repetitive muscular contractions, such as those induced by K^+ stimulation, muscle fatigue and potentiation may both impact the resultant contractile response. Therefore, we considered that the strength of muscle contractions was not an appropriate measurement in our system. Additionally, it was difficult to analyze this, and other parameters related to the degree of muscle contraction, such as frequency and duration of muscle contractions, since we cannot control precisely the moment at which contractions start when using high- K^+ .

10. Page 18. The last paragraph of the Results overstated the implication of the results in this paper. They state that "... a balance of mechanical forces is tightly coordinated between NMs and the muscle to regulate remodeling at the NMJ...". The results have not indicated this. They have put together a solid story that myosin-II acting with actin regulates bouton genesis. However, a weakness of the paper is that they have not provided credible evidence of whether muscle contraction locally provides mechanical force to the NMJ.

The reviewer considered that we overstated the implication of our results when we stated that "...a balance of mechanical forces is tightly coordinated between NMs and the muscle to regulate remodeling at the NMJ..." Regarding this criticism, we agree with the reviewer in that we did not provide convincing evidence that muscle contraction provides a local force during NMJ remodeling. However, even though more studies are needed to test and solidify this hypothesis, our results suggest that mechanical forces do participate in NMJ plasticity, and we showed that provoking imbalances in the forces applied by MNs or muscles during activity-dependent plasticity was able to change the dynamics and output of this process. Therefore, to account the role of mechanical force, without overstating the implications of our results we have now reformulated the statement to "...at the *Drosophila* NMJ, in addition to biochemical signaling, mechanical forces are coordinated between MNs and the muscle to regulate neuronal remodeling by blebbing".

Below we discuss in more detail how our results suggest that mechanical force is implicated in the regulation of activity-dependent bouton formation:

-We frequently observed muscle contractions preceding or accompanying bouton formation. Additionally, we observed two patterns of bouton formation: 1) fast bouton formation was frequent and associated with strong muscle contraction; 2) occasionally, slow bouton formation was observed with less muscle movements.

-We showed that in WT NMJs the time of bouton formation was only partially explained by their size, suggesting that in addition to membrane expansion, other factors regulate bouton growth.

-When myosin-II (the main source of cortical tension in cells) was reduced in neurons, we observed boutons formed with an elongated and less defined shape (instead of round), which is suggestive of lower stability against mechanical forces. In this scenario of reduced myosin-II levels, NMJs formed more boutons and bouton formation was faster and occurred while the muscle was contracting. Additionally, we often saw formation of bouton threads (“beads on a string”) with clear muscle contractions.

- Moreover, a clear dynamic shift in myosin-II KD NMJs occurred, with the linear relationship between area and time, being almost lost. We observed that in boutons forming very fast - independently of being large - which suggest that other than neuronal force generation, another factor had a larger effect on bouton growth dynamics.

- We showed that blocking muscle contraction prevented overgrowth of boutons in myosin-II KD NMJs (and the same when we expressed DN form of myosin-II in neurons) suggesting that this effect was dependent on muscle activity. Also, mechanically blocking muscle contraction reduced bouton formation in WT NMJs which suggested that muscle contraction is required to generate force (possibly by compressing MNs) and power bouton formation.

Altogether, we think that it is legitimate to say that mechanical forces are coordinated between MNs and the muscle and have a role in regulation of NMJ remodeling, but we contextualized better this sentence. Nevertheless, we have further tested muscle contraction contribution for activity-dependent plasticity in WT NMJs by manipulating both synaptic activity and/or muscle contraction levels (**Fig. 7**), which resulted in predictable changes in bouton formation in response to stimulation, with a progressive increase in new bouton numbers with higher levels of Ca^{2+} /synaptic activity and/or muscle contraction. Overall, we showed that changing the amplitude of forces at play, which depend on activity levels of both neurons and muscle cells, can change the plasticity levels upon acute stimulation of the NMJ further supporting our hypothesis.

Regarding the hypothesis of mechanical forces, such as muscle contraction, having local effects during plasticity, we consider this question challenging, but also very important to address in future studies, which we have addressed in the Discussion section.

[Minor comment]

11. No explanation is included for Fig 2h, 3b, and 5j in the main text.

We apologize for this error that is now corrected.

Reviewer #3 (Remarks to the Author):

This manuscript provides important mechanistic insight into the basis of acute structural synaptic plasticity at a widely used model synapse, the *Drosophila* neuromuscular junction. A multitude of studies over the last 10+ years have used acute stimulation to delineate signaling pathways important for new synaptic bouton formation in this system, without understanding the cell biological nature of new bouton growth. The current study uses a combination of live imaging and *Drosophila* genetics to suggest that rapid new bouton addition occurs by blebbing, which has been extensively characterized as a mechanism for cell motility, but to my knowledge not previously considered as a mechanism for synapse morphogenesis. The authors go on to show that muscle contraction plays an important role in formation of these blebs, indicating that mechanical forces cooperate with biochemical signals to drive structural plasticity.

The overall message will be of broad interest to the field and is generally supported by the data (though see below for specific questions and concerns). However, the manuscript requires a number of changes before it is accessible to readers and properly acknowledges its impact and place in the field. In particular, the authors fail to provide enough background of previous work studying this paradigm of rapid new bouton addition, and how their results fit in with these previous studies.

We truly want to thank the reviewer for this positive appreciation of our paper. Yes, we do confirm that this study is the first report of blebbing in neurons, which we have now emphasized in the text as well, since it is novel and an important message we want to pass to the readers. Additionally, we appreciate the feedback regarding the lack of background written in the previous version. We thoroughly edited the manuscript according to the reviewer suggestions, and we think is now accessible to all readers and includes the relevant previous findings. Once again, we thank the reviewer for the suggestions.

Major points:

1. The new boutons quantified in the live imaging studies seem extremely heterogeneous. In addition to the expected budding of boutons from pre-existing boutons (Fig 1d) some of them appear to originate from naked axon rather than from existing NMJ (e.g. Fig 1a bottom panel, 305108_0_video_893854_qqj956.avi) or seem to form from boutons splitting or just changing shape (e.g. Fig 1a top panel and Figure 1e, 305108_0_video_893855_qqj9j6.avi). The authors mention that some form quickly and some form slowly (as previously described by Piccioli et al., 2014). Since all these types of events are lumped together in the live imaging analysis without

clarifying their distribution, it is unclear whether certain types of events dominate the quantifications. Also, due to this heterogeneity, it is not clear what fraction of boutons assessed by live imaging relate to the Dlg-negative, HRP-positive structures characterized in this paper and in the literature as “ghost boutons” in fixed, immunolabeled samples. It is particularly hard to imagine that “splitting” boutons in the middle of a string are Dlg-negative.

The authors should explain why it is appropriate to pool these apparently different types of events together, and describe the distribution is between these types of events. They should also more directly relate their live and fixed imaging, by quantifying from two-color live imaging experiments how many presynaptic events (either type of new bouton according to the categories above, or live actin behavior) lack a corresponding Dlg signal. There is a Dlg knockin available that may be suitable. For this experiment, please ensure that these presynaptic events are identified blind to the Dlg channel.

First, we would like to recognize that the reviewer made some very valid points and thank for all the suggestions. Regarding why we pooled all live events together: we thought of doing a cluster analysis of our data, which is the standard statistical method to determine naturally occurring groups (known as clusters) within a dataset, without the need to group data points into any predefined labels or classes. The general idea is that the clustering is performed in a way in which the objects in the same cluster are more similar to each other, than to those in other clusters. Although we agree that this would be a way for us to classify two (or more) populations of boutons, to perform this type of analysis with reliable clustering and maintaining statistical power, we would need a much higher n (bare minimum n= 20 to 30 per expected group). This type of n is hard to obtain in the live imaging, especially with data sets characterized by many features (from our videos we could only analyze at maxima time and size), with each feature (variable) adding an additional dimension to the data. Additionally, the % of “slow” events is very low compared with the “fast” or close to the median value. For these reasons, and since identification of classes could not be done in an unbiased manner, we decided to analyze together these apparently different types of bouton formation events. We have added a simplified sentence to the revised manuscript explaining why we did not separate the events to the general reader.

Regarding this part of the reviewer's comment “Since all these types of events are lumped together in the live imaging analysis without clarifying their distribution, it is unclear whether certain types of events dominate the quantifications”. We agree that, considering only the formation time of boutons, it is unclear whether specific groups of events (and how many groups of events) dominate the quantifications. However, we did present the distribution of all data points across experiments

in **Fig. 1c** in the format of boxplot (min to max), with the median (5 min38s) being represented by the middle line. Importantly, contrary to the mean, the median is not skewed by a small proportion of extremely large or small values, providing a better representation of a “typical” value. Accordingly, most data points concentrate around this value, supporting what we observed in our movies, bouton formation was frequently rapid “ranging from less than 1 min to a few min” as described in the manuscript. Therefore, even though we did not separate the events, we did provide a descriptive and quantitative (time of formation) analysis corroborating that bouton formation is usually a fast process (and certainly faster than budding).

Concerning the question of whether all events that we registered live are ghost boutons, we cannot be 100% sure. We will need to do more experiments with the Dlg Trap. We did several experiments using this line but, unfortunately, all boutons without Dlg formed during the stimulation and we never saw much happening after the stimulation period, which is when we are able to image our preparation. If this lack of events is related with the Dlg-GFP line or just lesser luck, we do not know. Because these experiments are very long, analysis laborious and time consuming, associated with the fact that we cannot predict whether live events will occur during the imaging session, and giving that this paper is already very long, we decided to not proceed for now. Since this is a rather important biological question, beyond the scope of this manuscript, we will address it in the near future.

2. One of the major arguments that the structures are blebs is that they have reduced F-actin and filamin compared to the parent bouton. I do believe the structures are blebs based on their speed of formation, shape, regulation by myosin, and role of external mechanical forces. However, I am less enamored of the actin localization arguments. The authors do not know the identity or function of the major F-actin or filamin structures in the parent boutons, which provide the baseline for their measurements, and this could really matter. For example, if most F-actin in the parent bouton is at sites of compensatory endocytosis (as suggested by the observation that FM dye uptake upon stimulation is actin-dependent (Wang 2010)), then it would not be surprising that these structures are absent from ghost boutons, which only very rarely mature enough to recycle synaptic vesicles (Vasin 2014). On the other hand, any kind of actomyosin cortical cytoskeleton that forms the edges of a bleb or eventually enters into a new bouton might be inherently much less abundant relative to endocytic or other structures. It might be helpful to see filamin and Lifeact co-staining, to demonstrate what proportion of the presynaptic actin co-labels with filamin. The authors should also note that there isn't currently experimental evidence for a cortical actomyosin cytoskeleton in synapses, comparable to the cell cortex of motile cells, so such an apparatus remains hypothetical. Overall, the authors should either justify the importance of the actin localization data, or explain

this major caveat and play down the importance of the actin localization evidence in the manuscript.

Like actin, we previously showed that filamin puncta delineate mature boutons. After stimulation, most boutons formed with low actin suggesting a weaker actin structure. Supporting this claim, filamin was never observed inside ghost boutons, remaining in parental boutons. The reviewer pointed out that the function of the major F-actin or filamin structures in parental boutons could be related to endocytosis, which “would explain why there are absent from ghost boutons, which only very rarely mature enough to recycle synaptic vesicles (Vasin et al., 2014)”. We acknowledge the reviewer’s concern and to test this hypothesis we used NSyb-Gal4 to express Lifeact-GFP (F-actin marker) in neurons and analyzed endocytosis at the NMJ after high-K⁺ stimulation by colocalization with Shibire (dynamin), that we used as a marker for active vesicular endocytosis. The images are presented below in **Reviewer Figure 5**.

As expected, in mature boutons most F-actin is found at sites of compensatory endocytosis, which we identified by colocalization with Shibire antibody staining (**Reviewer Figure 5**). However, we did find several examples of ghost boutons containing actin and Shibire puncta, which suggested that these structures are capable of endocytosis. Furthermore, we always fixed our samples 30 min post-stimulation, and by this time point we never observed filamin being recruited to ghost boutons, although we found different profiles of actin distribution in ghost boutons, and examples of ghost boutons showing putative endocytic capacity (based on Shibire presence). Unfortunately, we were unable to co-stain Filamin with F-actin or Shibire due to incompatibility of fixatives (Bouin’s for Filamin and PFA for Lifeact-GFP/Shibire Ab), which would allow us to confirm that Shibire was found in ghost boutons without filamin. Although more experiments are needed to clarify this question, our data suggests that the presence of filamin in ghost boutons, may not be required for endocytosis to occur. Since filamin was shown to be able to potentiate the dynamin-dependent endocytosis in neurons (Noam *et al.*, 2014) one possibility is that filamin absence in ghost boutons may be linked to reduced endocytosis levels in these structures, which are not fully mature yet.

Reviewer Figure 5. Legend: a, Representative confocal image of m6/7 NMJs of Lifeact-GFP expressing larvae (under control of NSyb-Gal4) stained for Dlg (postsynaptic marker, in cyan) and Shibire (which labels dynamin and the active site for vesicular endocytosis, in magenta). The GFP signal of Lifeact was amplified with anti-GFP antibody. Scale bar is 10 μ m. a', a'' Super-resolution confocal zoom in images of ROI regions showing dynamin patterning in mature boutons (a', "shibire donuts") and an example of dynamin recruitment to ghost boutons (GBs) including F-actin. b, Additional confocal images of actin and dynamin distribution in GBs. Scale bar is 2 μ m. GBs are indicated by arrows and identified by the lack of Dlg.

Another possibility to consider is that filamin-associated F-actin structures are involved in other functions beyond endocytosis. As the reviewer clearly expressed, “On the other hand, any kind of actomyosin cortical cytoskeleton that forms the edges of a bleb or eventually enters into a new bouton might be inherently much less abundant relative to endocytic or other structures. It might be helpful to see filamin and Lifeact co-staining, to demonstrate what proportion of the presynaptic actin co-labels with filamin.”

As we mentioned above, although we would like to present such data, due to incompatibility with fixatives, the co-staining of F-actin and filamin was not possible. Nevertheless, we now discuss here the hypothesis presented by the reviewer. In addition to being involved in endocytosis, filamin is well-known to be a key factor connecting the membrane to the cytoskeleton, and in cells, in addition to the tension produced by actomyosin contractility, filamin is required to develop and maintain an organized actin cytoskeleton (Kelly *et al.*, 2020). From our data, filamin organization, delineating boutons, was always maintained in parental boutons and filamin was never found in ghost boutons. Likewise, the actin cortex remained intact in parental boutons, although it was weakened in ghost boutons. Supporting this, we saw that both actin and myosin were very dynamic during the formation of boutons, being less abundant inside new boutons at the first stages of their development and accumulating once growth stabilized. Altogether, our data indicated that actin cytoskeleton reorganization is required for bouton formation, and filamin complete absence from these structures supports this hypothesis, independently of its endocytic related functions.

Regarding this part of the reviewer’s comment: “The authors should also note that there isn’t currently experimental evidence for a cortical actomyosin cytoskeleton in synapses, comparable to the cell cortex of motile cells, so such an apparatus remains hypothetical. Overall, the authors should either justify the importance of the actin localization data, or explain this major caveat and play down the importance of the actin localization evidence in the manuscript.”

It is true that the experimental evidence for a cortical actomyosin cytoskeleton in synapses, comparable to the cell cortex of motile cells, remains hypothetical, but recent discoveries have made big advances on the knowledge of the structure and components of the neuronal actomyosin network. One important breakthrough finding of Single Molecule Super-resolution Microscopy, was the existence of a periodic membrane associated cytoskeleton (MPS) - composed of a periodic arrangement of ~200 nm spaced actin rings found along axons, that are interconnected by bipolar spectrin tetramers (Mikhaylova *et al.*, 2020 review on the topic; original research in Xu *et al.*, 2013). This structure has never been observed by researchers in decades of investigation of axons with

electron microscopy, most likely due to the very fragile nature of actin filaments. Since then, the periodic MPS has been seen by several microscopy methods and verified in Platinum Replica Electron Microscopy of cultured neurons (Mikhaylova *et al.*, 2020; original research in Vassilopoulos *et al.*, 2019). Furthermore, the MPS structure seems to be conserved from worm to humans, and by now it has been found in a variety of neurons of the central and peripheral nervous system, including in MNs (He *et al.*, 2016; D'Este *et al.*, 2016). Additionally, while the MPS was initially only observed in neuronal axons, it has also been found in dendrites (at the neck of dendritic spines) (Mikhaylova *et al.*, 2020; original research in Bär *et al.*, 2016; Sidenstein *et al.*, 2016), the postsynaptic compartments of most excitatory synapses. Importantly, in neurons myosin-II motor proteins have been recently linked to MPS regulation, being implicated in regulation of axonal initial segment, axonal diameter (by actomyosin contractility) and action potential firing (Costa *et al.*, 2020; Wang *et al.*, 2020). Moreover, the role of actomyosin in the neuronal cytoskeleton goes beyond regulation of axonal stability and propagation of electrical signals, being also involved in LTP-dependent dendritic spine structural plasticity (Rex *et al.*, 2010).

While the presence of an actomyosin cortex at synapses is still in debate, *Drosophila* NMJ synaptic boutons are clearly discernible, comparable to the minute size of mammalian boutons, and our data (**Figs. 4** and **Supplementary Fig.14**) shows that myosin-II is present in mature boutons, sometimes being visible outlining boutons. Furthermore, myosin-II was recruited to ghost boutons, with clear accumulations at the base (strong punctum) or edge (partially or completely delineating the cortex). Therefore, to provide a more detailed view on myosin-II organization, we added to the revised manuscript a super resolution image of myosin-II structure at the cortex of one ghost bouton example (**Fig. 4b**): we can observe myosin-II puncta clearly surrounding the cortex of the ghost bouton. Considering our live data, in which myosin-II remained basal during stages of vigorous growth and accumulated only when growth stalled, this data hints that an actomyosin apparatus likely exists in *Drosophila* NMJ boutons, and eventually accumulates in ghost boutons after their formation.

In the new version of the text, we better discuss the role of actin and myosin-II, and trust that everything we wrote is supported by the data provided.

3. For the actin drug experiments in Fig 2, which are important for the blebbing argument, the authors state that they measure “the number of F-actin puncta” to assess the effectiveness of the drugs at destabilizing or stabilizing actin filament structures. There are a few arrows on n=1 NMJs per treatment, but no quantification is shown as they claimed. This is required. Since we don't know what these structures are, lifeact intensity, number of puncta/NMJ area, and coefficient of variation of lifeact would be an appropriate measure, rather than raw number. For the Jas samples,

a FRAP assay showing slower turnover may also be useful if steady-state quantifications are unclear. The data in Figure 2 are uninterpretable without these measurements, especially given the lack of change in ghost bouton number which is the opposite of the observations in Piccioli et al (reduced ghost boutons in both LatA and Jas). These quantifications should be combined with a repeat of the main results of Figure 2 in parallel, to ensure that the drug treatment worked in the same samples as the bouton size measurements are made.

We acknowledge the problems that the reviewer pointed out. To answer the points raised, we performed the following experiments/quantifications:

1. Quantification of Lifeact puncta number per NMJ area pre and post treatment of NMJs with LatB or JAS (**Figs. 3a-d**).
2. Quantification of Lifeact intensity per NMJ area and per puncta pre and post treatment of NMJs with LatB or JAS (**Supplementary Figs. 11b-e**).
3. FRAP with and without JAS (**Figs.3e-g and Supplementary Figs. 11f,g**).

Briefly, LatB treatment led to a significant decrease in both the number of F-actin puncta per NMJ area and intensity per puncta, while JAS resulted in the increase of these parameters confirming that upon these treatments, F-actin dynamics was shifted towards a more depolymerized or polymerized state, respectively.

Moreover, FRAP confirmed that JAS treatment increases the half-time recovery of Lifeact. Additionally, JAS treatment reduced both total recovery and mobile fraction of Lifeact, further supporting JAS effect in F-actin stabilization.

Importantly, we compared FRAP on 5C-actin and Lifeact (two different F-actin reporters) showing that both probes respond similarly.

We think that the difference with the results obtained by Piccioli & Littleton (2014) may be related to a phenomenon that we describe in the text, which is the fact that we often see many very small boutons together, suggestive of a common event. If the Piccioli & Littleton (2014) study did not catch a comparable number (their n was lower than ours) or if they did not count these small boutons as ghosts, it can explain the differences, since in our experiments the distribution of ghost boutons with JAS is skewed towards low and high numbers. These points were included in new figures and in the text.

4. While I appreciated many of the explanatory diagrams, other parts of the data presentation and analysis were extremely difficult to understand, and must be clarified for a general readership.

We agree with the reviewer regarding the way we present data and its analysis - which had some parts that were difficult to understand. We re-wrote the manuscript and hope that it is now clearer. Still, we address here each of following questions raised by the reviewer (below).

- The language describing kymograph generation was not understandable to me, though I tried very hard. “We draw a line ROI from the base that passed through the bouton reaching the edge and plotting this profile, encoding the intensity in grayscale level (or color for color images) for all slices in the Δt . To adjust the line position, we select a frame where the bouton is still small and draw a line ROI preferentially from the center through the bouton. However, sometimes this was not possible. In these cases, we used the max size frame as reference and draw a line ROI that extended from base to edge during bouton growth.”

What I did understand made the selection of base in particular seem subjective. Is the base at the point of origination of the new bouton (as seems to be the case in 1e, though the “b” end of the line seems arbitrary), or is it the opposite side of the pre-existing bouton (e.g. Fig 1d)? For the $1\mu\text{m} \times 1\mu\text{m}$ boxes used for quantification, it would make a huge difference if it were placed on a bright actin punctum or not. The authors should rewrite these sentences with attention to grammar and clarity, or draw a better diagram to describe how what they did was consistent from bouton to bouton.

- The authors use the label “distance” on the line scans (e.g. 1d’ and 1e’). Methods suggest that this represents “distance” along the time axis in the kymograph, which is an extremely confusing thing to call a time measurement (“In these plots, x-axis reports time (along the line that we draw) and y-axis is fluorescence intensity along this time window.”). Also, is it normalized since the times are different in different boutons? “Distance” in some figures (eg 3c’ and 7a’ is 10x larger than in 1d’, 1e’, 3d’). Please normalize similarly (if this is what was done) and change the label from “distance” to something like “normalized time”.

We recognize this difficulty and hope that our answer clarifies the reviewer. First, we would like to highlight that kymographs are used as graphical representations of a spatial position over time, to visualize motion of fluorescent molecules (or other particles/organelles) moving along a predictable path. In our particular case, the predictable path is the ROI that we draw - a line from the base to the edge of the bouton during a chosen time interval (while the bouton was growing). After generating the kymographs, and as they are oriented in the figures, the y-axis reports space

(distance between base and edge) and the x-axis reports time (from beginning until the bouton reached the maximal size). Since both y- and x- axis lengths vary between different boutons (that formed with different maximal sizes and times), the kymographs we present are a representation of what happened for each example of bouton formation and should be only interpreted as such. We provided the kymographs of bouton formation as a complementary information to the time-lapse images, which do not give information about the continuity of the process. To facilitate the reading of the kymographs we have indicated in each figure the axis reporting to space and time.

Additionally, after the kymographs are generated, we can measure changes that occurred in the membrane and actin or myosin-II channels at the base of at the edge, by tracing a line along the base or the edge for each channel and plotting the fluorescence intensity values for these lines. Again, in these plots, the y-axis gives the fluorescence intensity along edge or base lines, which are in the x-axis represented as distance. To our knowledge, the term distance is widely used in kymographs from time-lapse movie sequences and refers to the starting and ending points of the lines we draw (ROI), and, although it alludes to the time window analyzed it is not equivalent/proportional to it. To clarify this, we changed “distance” to “distance along base or edge line ROI”, and, additionally, we show the time interval in which these changes occurred on top of the plots. We hope this makes the figure more clear.

- The histograms in Figure 5 are really hard to interpret because there’s too much interleaved information. They should be separated or simplified.

We understand this point and show now a simplified version of histograms in which we show relative frequency (as percentage of events) for each class, and grouped the classes in fewer intervals which facilitates the interpretation. We hope the histogram reading is clearer now.

5. The authors oversell the connection between this activity-dependent paradigm for structural plasticity and physiological synaptic growth. Indeed, many mutants with severe defects in acute plasticity (e.g. cortactin and synapsin) and muscle contraction (e.g. Syt1) have normal arbor size and bouton number in third instar larvae. The spaced stimulation paradigm is still very interesting and serves as an excellent system to understand synaptic signal transduction and membrane dynamics, but the authors should soften language throughout. For example, I think it’s still very interesting if the physiological relevance regards membrane tension and mechanical stress at the synapse, rather than actual physiological or activity-dependent bouton growth that would drive synaptic strength in a significant way.

We thank the reviewer for the comment. We now have softened the language, but we think that it is important to note that there is a relationship between these developmental synaptic growth and activity-dependent plasticity, although it is not well understood if they are regulated by distinct mechanisms. We discuss here the relationship between these processes.

While synapse development is mostly genetically determined, established synapses can be subsequently refined through activity-dependent mechanisms. At the *Drosophila* NMJ, developmental expansion occurs primarily through Wnt/Wg- and BMP- canonical signaling, which regulate the addition of new boutons. However, during NMJ development, besides bouton addition, bouton elimination (or pruning) also occurs to refine the synaptic structure and prevent exuberant growth. These sequential morphological processes of bouton addition, expansion to full size and, if necessary, pruning, are modulated by baseline and/or activity-dependent signaling cues, in such a way that ensures a synaptic size and structure that allows proper connectivity and strength. Interestingly, for both Wnt- and BMP-, activity-dependent non-canonical signaling pathways have been described, which are known to regulate acute structural plasticity at the NMJ. This suggests that developmental and activity-dependent mechanisms of synaptic growth share some common signaling mechanisms, although they can be regulated by different temporal requirements (activity-dependent stimulus can trigger local and sustained responses opposed to the more intermittent and global growth seen during development). Given these, we now talk about these points in the introduction and discussion, hopefully softening a potential oversell.

An example:

p21. “Without a bleb-like mechanism, we would predict a slower NMJ expansion, not always compatible with the requirements of the neuromuscular system during development or increased activity.”

We agree that we cannot predict a slower NMJ expansion with only our data, and we therefore removed this sentence from the current manuscript. We thank the reviewer for spotting this issue.

We don't know that the rate of bouton growth during development is fast. Further, many mutants that have reduced ghost bouton formation (e.g. Syt1, Synapsin) do not have undergrowth phenotypes, suggesting that these may not be entirely related processes.

We agree with the reviewer in that we do not know the rate of bouton growth during development. However, based on our unstimulated condition data, together with the report by Vasin *et al.*, 2019, where the authors showed that new boutons can form rapidly in intact larva in response to intense crawling activity, we suggest that bouton growth during development can also be fast. Furthermore,

as we discussed above, although developmental and activity-dependent bouton formation share common elements, they are probably differently regulated and may employ distinct mechanisms. This may help explain why many mutants that have reduced ghost bouton formation, do not have undergrowth phenotypes. In other words, we can speculate that in these mutants, mechanisms activated by basal activity levels are sufficient to ensure normal bouton formation during larval development. However, since in these animals, activity mechanisms are compromised, they cannot properly refine these connections to match intense activity changes and may be less able to cope with environmental stress.

In our opinion, during developmental periods of intense activity, activity-dependent mechanisms may be used to potentiate the rate of growth, or to provide selectivity of sites for increased synaptic growth, overlapping temporarily with mechanisms activated by basal activity. However, future studies will be needed to clearly elucidate these questions. It will also be interesting to test whether neurons (like cancer cells) can adopt different strategies for growth, depending on the 3D/extracellular cues.

In the new version of the text, we discussed developmentally regulated vs activity-dependent bouton formation. We hope it is now clear that our data does not directly address developmental bouton formation and that it is possible that bouton addition may be differentially regulated and may employ distinct mechanisms.

6. On a similar note, while the reference list includes some of the previous literature on this stimulation-induced bouton growth, it is not enough to have a list of reference numbers following a broad statement (e.g. “Furthermore, combining these protocols with live imaging studies showed that activity-induced presynaptic morphogenesis may be different from the embryonic GC (Ataman 2009, Piccioli 2014)”, without saying in what way these papers showed they are different – indeed Ataman did suggest there were filopodia-like “synaptopods”). Further, the term that has been used to describe these new boutons for over a decade (“ghost boutons”) is not used at all in the manuscript. This will be confusing for the field, and the authors should use this term, and briefly tell the reader what is known about maturation of a fraction of these structures into functional boutons.

We agree with the reviewer’s comment, and we have now reformulated the manuscript to settle these issues. We have added the term “ghost boutons” and explained its origin. We also wrote what is known regarding bouton maturation and, because no study had looked at maturation *ex vivo*, we did established a method (**Fig. 1e-h**) to follow maturation in the dissected preparation.

We found several examples of maturation of ghost boutons by acquisition of synaptic components, such as AZs and GluRs (shown in **Fig. 1e-h**).

Relative to filopodia-like “synaptopods”, we discuss below our opinion on their putative role in activity-dependent bouton formation, which we find to be unclear.

Ataman *et al.* (2008) first described that “the most striking structural changes” they observed in response to patterned stimulation of the NMJ with high-K⁺ were “the *de novo* formation of synaptopods and ghost boutons”. However, the authors stated that the nature of these filopodia-like structures (synaptopods) was unclear, and, that although synaptopods could represent an initial stage during synaptic bouton formation (similar to filopodia observed at dendritic spines in normal animals and in response to activity), they did not observe a transition from synaptopod to bouton. Furthermore, the authors argued that synaptopods could be exploratory structures conveying signaling between pre- and/or postsynaptic sites (a function also suggested for dendritic filopodia and for the growth cone filopodia prior to target innervation). On the other hand, ghost boutons, clearly represented rapid *de novo* formation of undifferentiated boutons.

Recently, Vasin *et al.* (2019) also debated the role of synaptopods in bouton formation. The authors described two patterns of bouton formation and maturation at *Drosophila* NMJ: one involving growth of filopodia followed by formation of boutons that were initially devoid of synaptic vesicles but contained a filamentous matrix, and the second was called rapid budding of synaptic boutons packed with synaptic vesicles, which were described as more mature boutons occasionally capable of exocytosis/endocytosis. However, in this study, the authors did not explain how they classified filopodia vs non-filopodia, namely the presence of actin and actin bundling proteins (such as fascin or finbrin) inside these protrusions, was lacking. Without this data, it is hard to be sure on the nature of the protrusions, specially without live imaging to observe the dynamics. Despite this, our data agrees with what these authors report: the majority of boutons form with synaptic vesicles and from non-filopodial precursors. In our live imaging, we sometimes observed filopodia-like structures (like in Ataman *et al.*, 2008), but never saw their conversion into a bouton-like structures. Instead, in many hours of movies analyzed, boutons always formed as spherical protrusions arising from the MN membrane, sometimes leaving thin threads behind them, which maintained boutons connected to the main arbor. The main difference we observed between fast and slow bouton formation was the presence of visible muscle contractions associated with the first type.

In conclusion, although in our study we present blebbing as a new mechanism of bouton formation, we do not exclude other types of bouton formation. But we did not observe any bouton formation

from filopodia. Since filopodia are structures sensitive to light we rarely observe them with live microscopy (which required considerable laser power to observe the membrane in new boutons - sometimes dimmer compared to parental boutons) and upon chemical fixations (very rarely being preserved in fixed samples).

We also added data and discussion regarding the maturation potential of ghost boutons.

Overall the text does an inadequate job of actually discussing what is known, and putting the current manuscript in context. Neuronal activity, Ca²⁺-dependence, functional actin cytoskeleton and contractile muscle were all previously studied as prerequisites to induce ghost bouton formation. Since the ghost boutons are formed even with severed axons scientists suspected that there was a “local” effect at the synapse too. This does not detract from the novelty of the current study in characterizing blebbing as the mechanism for ghost bouton formation.

We agree with the reviewer that we did not thoroughly describe what is known about activity-dependent bouton formation, which represents a vast and dense repertoire of studies. We have now significantly updated the text to according to the reviewer suggestions.

Some examples of missed context:

p.18. “We hypothesize that elevated activity, and in response to a still unclear signal to initiate bouton growth, MNs add new boutons by membrane blebbing and muscle contraction is required to increase neuronal confinement, and consequently cortical tension, which powers bouton outgrowth.” “In this scenario, as it has been proposed for cells in compressive 3-D environments⁶⁴, it is possible that a muscle derived signal, released in regions of higher activity, can polarize the recruitment of molecules required to or that facilitate bouton formation to spots of high membrane tension. While the molecular identity of the signal that leads to bouton initiation remains to be elucidated, ...”

These statements do not take in to account extensive previous findings on bidirectional Wnt signaling (Ataman et al., 2008), Syt4 signaling (Korkut 2013), BMP signaling (Piccioli et al. 2014), integrins (Lee et al 2017), ECM (Dear 2017), CamKII (Nesler 2016) as well as cAMP/PKA-dependent pathway (Vasin et al., 2014), associated with ghost bouton formation, and therefore the findings described in the current manuscript sound (incorrectly) completely novel.

It was not our intention to oversell our findings. Our intention was to emphasize our main findings: membrane blebbing and MN confinement powered by muscle contraction as a new mechanism of remodeling. To address this problem, we have re-written the text taking into account previous data and our current findings. We thank the reviewer for this suggestion.

p 20. – “Another question that remains unanswered is related with the membrane source required for structural changes observed in blebs, or, in our case, for neuronal remodeling.”

There is massive synaptic vesicle fusion in the paradigms used in this manuscript, which should be mentioned as a likely “membrane source”, citing Vasin et al., 2014 who show that synapsin mutant NMJs, which have fewer synaptic vesicles, have reduced ghost boutons upon stimulation. The authors do cite here a subsequent synapsin paper with in vivo work, but not the relevant paper using the fillet stimulation paradigm.

We find this question of the membrane source for bouton outgrowth very exciting and that is why we decided to discuss it. However, given that we do not do any experiments to directly test this, we decided to remove this paragraph.

Additionally, we added the points and respective references suggested by the reviewer to the text.

Finally, San Martin 2017 is cited in the text but is absent from the references.

This is now corrected.

Minor concerns:

Methods and Tables

1. The authors state that they use the spaced stimulation protocol from Vasin 2014, and then “Additionally, for live imaging, we also used a mass stimulation procedure of 10- or 16-min High-K+ incubations developed by Martin et. (2017).” The authors must specify which paradigm was used when (and also include the San Martin paper in their references – it is absent). Since this high-intensity stimulation is the essence of the manuscript, it is critical to clarify in each figure/legend which type was used.

The reviewer is right. Although we mainly used spaced stimulation in our live movies, we also used massed stimulation, and, since the dynamics of bouton formation we observed between the two protocols were identical, we did not discriminate before. We have now clearly identified the type of stimulation used (spaced vs massed) stimulation in the methods.

2. It is unclear what “# experimental repeats” means in Table 4 or “biologically independent experiments” in the legends. For example, in the first row, were 13 NMJs from 13 animals imaged 10 times, for 130 independent points? This seems unlikely. What does the 10 refer to?

We apologize for not being clear, we reformulated the writing and hope it is clearer now.

In **Fig 1b**, “# experimental repeats” refers to the quantification of the time for bouton formation, assessed by live imaging where we imaged 13 larvae, 1 NMJ per larvae (13 NMJs), from 10 different crosses made throughout the time of the project. For live imaging, we only do 1 NMJ per animal to assure that the integrity of the larvae is good. The number of boutons plotted in **Fig1b** (46 boutons), reflects the fact that more than one ghost bouton formed per NMJ in response to the stimulation. In fact, and as reported by the Littleton lab, most ghost boutons form during the stimulation, being therefore hard to catch these events live post stimulation. Still, there is ~30% probability of observing bouton addition post stim (1 out of 3 movies have events), and whenever there are events, it is likely that more than one bouton forms. Mechanistically this is interesting as it may suggest that priming for new boutons is coordinated in the whole arbor – but this needs to be tested.

Figures

3. Not clear what “Pint” is meant to indicate in 1c.

It means Internal (int) Pressure (P) – we have written this in the figure (which is now Fig. 2a instead of Fig. 1c) legend and have put int as a superscript to be clearer.

4. Signals in magenta and green are often very different in intensity so it would be more informative to present normalized fluorescence intensity in the kymographs.

We now present actin and membrane normalized fluorescence intensity (min-max normalization in each time interval) in the kymographs, and show graphs with comparison between base and edge levels for each channel.

5. The NASPM experiments lack a NASPM alone control (without stretching). What is the effect of this manipulation on its own, and do the authors expect a difference for acute inhibition compared to chronic GluRIIA mutants tested in Piccioli et al?

We have added this control (NASPM alone without stretching) in **Supplementary Fig. 23**, which showed that muscle inactivation with NASPM alone was also able to reduce bouton formation in the control (NSyb > +), and at intermediate levels in MyoII-IR, since this decrease was further accentuated with mechanical stretching. In the semi-stretched condition, there were still differences observed between GB formation in control and reduced Myo-II in neurons, suggesting

that residual muscle contraction allowed GB formation when the membrane cortex in neurons was weaker.

Although chronic genetic loss of GluRIIA subunit (in mutants and in postsynaptic RNAi) leads to a reduction in mEPSP, with corresponding decrease in synaptic strength, adaptation over chronic time scales has been reported (Li *et al.*, 2018) with EPSP amplitudes being restored to baseline values despite continued diminishment of GluRIIA functionality, due to a homeostatic increase in presynaptic NT release. In theory, if the dosage of NASPM was able to inhibit all GluRs, and since acute inhibition with NASPM does not lead to adaptation in our experiments, we would expect a more pronounced reduction of synaptic strength, and, hence, of muscle activity when compared with GluRIIA KD (or GluRIIB). In the paper by Piccioli et al (2014), in the postsynaptic KD of GluRIIA, the authors observed a reduction in the formation of GBs after high K^+ . It has also been shown that a retrograde signal is required for structural plasticity (Piccioli & Littleton, 2014; Sulkowski & Serpe, 2014). Putting all data together with our results, we propose a model where synaptic activity activates a GluRIIA-dependent signal in the muscle, which is released acting retrogradely in the MN to prime places for new bouton addition. Hence, all are required: synaptic activity, GluRIIA-dependent retrograde signal, and muscle contraction.

6. I would reorder figure 5 in the following way to match the text better:

- G-i to a-c
- D-f no change
- A-c to g-i

We have reorganized previous the figure 5 in the revised manuscript (now designated **figure 6**), with all the panels being referenced in order in the text.

Text

7. p. 20 “Our live imaging with CD4-Tom revealed local intracellular vesicle dynamics in MNs at the place of bouton formation.” – This has not been quantified in the paper, and RFP-derived tags such as Tomato tend to aggregate. I suggest removing this statement.

We accepted the reviewer’s suggestion and removed this statement.

8. p20 “Synaptic boutons have been consistently associated with axonal GCs and actin mediated protrusive processes, such as lamellipodia or filopodia, which are typically seen in cells migrating in 2-D.” This statement needs references or should be removed.

After re-working on this manuscript, we decided to remove this sentence.

References

1. Fackler, O. T. & Grosse, R. Cell motility through plasma membrane blebbing. 879–884 (2002). doi:10.1083/jcb.200802081
2. Charras, G. & Paluch, E. Blebs lead the way: How to migrate without lamellipodia. *Nature Reviews Molecular Cell Biology* 9, (2008).
3. Schick, J. & Raz, E. Blebs-Formation, Regulation, Positioning, and Role in Amoeboid Cell Migration. *Front. cell Dev. Biol.* 10, 926394 (2022).
4. Ataman, B. et al. Rapid Activity-Dependent Modifications in Synaptic Structure and Function Require Bidirectional Wnt Signaling. *Neuron* 57, 705–718 (2008).
5. Fuentes-Medel, Y. et al. Glia and Muscle Sculpt Neuromuscular Arbors by Engulfing Destabilized Synaptic Boutons and Shed Presynaptic Debris. *PLoS Biol.* 7, e1000184 (2009).
6. Piccioli, Z. D. & Littleton, J. T. Retrograde BMP Signaling Modulates Rapid Activity-Dependent Synaptic Growth via Presynaptic LIM Kinase Regulation of Cofilin. *J. Neurosci.* 34, 4371–4381 (2014).
7. Vasin, A. et al. Two Pathways for the Activity-Dependent Growth and Differentiation of Synaptic Boutons in *Drosophila*. *eNeuro* 6, (2019).
8. Feng, Y., Ueda, A. & Wu, C.-F. A modified minimal hemolymph-like solution, HL3.1, for physiological recordings at the neuromuscular junctions of normal and mutant *Drosophila* larvae. *J. Neurogenet.* 18, 377–402 (2004).
9. Macleod, G. T., Hegström-Wojtowicz, M., Charlton, M. P. & Atwood, H. L. Fast calcium signals in *Drosophila* motor neuron terminals. *J. Neurophysiol.* 88, 2659–2663 (2002).
10. Rabinovich, D., Mayseless, O. & Schuldiner, O. Long term ex vivo culturing of *Drosophila* brain as a method to live image pupal brains: insights into the cellular mechanisms of neuronal remodeling. *Front. Cell. Neurosci.* 9, 327 (2015).
11. Louie, K., Russo, G. J., Salkoff, D. B., Wellington, A. & Zinsmaier, K. E. Effects of imaging conditions on mitochondrial transport and length in larval motor axons of *Drosophila*. *Comp. Biochem. Physiol. Part A, Mol. Integr. Physiol.* 151, 159–172 (2008).
12. Wu, G.-Y., Deisseroth, K. & Tsien, R. W. Spaced stimuli stabilize MAPK pathway activation and its effects on dendritic morphology. *Nat. Neurosci.* 4, 151–158 (2001).
13. Hu, X., Viesselmann, C., Nam, S., Merriam, E. & Dent, E. W. Activity-Dependent Dynamic Microtubule Invasion of Dendritic Spines. *J. Neurosci.* 28, 13094 LP – 13105 (2008).
14. Cai, Q., Pan, P.-Y. & Sheng, Z.-H. Syntabulin–Kinesin-1 Family Member 5B-Mediated Axonal Transport Contributes to Activity-Dependent Presynaptic Assembly. *J. Neurosci.* 27, 7284 LP – 7296 (2007).
15. Li, Z., Okamoto, K.-I., Hayashi, Y. & Sheng, M. The Importance of Dendritic Mitochondria in the Morphogenesis and Plasticity of Spines and Synapses. *Cell* 119, 873–887 (2005).

16. Yao, J., Qi, J. & Chen, G. Actin-Dependent Activation of Presynaptic Silent Synapses Contributes to Long-Term Synaptic Plasticity in Developing Hippocampal Neurons. *J. Neurosci.* 26, 8137 LP – 8147 (2006).
17. Maldonado-Díaz, C., Vazquez, M. & Marie, B. A comparison of three different methods of eliciting rapid activity-dependent synaptic plasticity at the *Drosophila* NMJ. *PLoS One* 16, e0260553 (2021).
18. Gogolla, N., Galimberti, I., Deguchi, Y. & Caroni, P. Wnt signaling mediates experience-related regulation of synapse numbers and mossy fiber connectivities in the adult hippocampus. *Neuron* 62, 510–525 (2009).
19. Tabatadze, N., McGonigal, R., Neve, R. L. & Routtenberg, A. Activity-dependent Wnt 7 dendritic targeting in hippocampal neurons: plasticity- and tagging-related retrograde signaling mechanism? *Hippocampus* 24, 455–465 (2014).
20. Alicea, D., Perez, M., Maldonado, C., Dominicci-cotto, C. & Marie, B. Cortactin is a regulator of activity-dependent synaptic plasticity controlled by Wingless. *J. Neurosci.* (2017).
21. Withers, G. S., Higgins, D., Charette, M. & Banker, G. Bone morphogenetic protein-7 enhances dendritic growth and receptivity to innervation in cultured hippocampal neurons. *Eur. J. Neurosci.* 12, 106–116 (2000).
22. McCabe, B. D. et al. The BMP Homolog Gbb Provides a Retrograde Signal that Regulates Synaptic Growth at the *Drosophila* Neuromuscular Junction. *Neuron* 39, 241–254 (2003).
23. Budnik, V. & Salinas, P. C. Wnt signaling during synaptic development and plasticity. *Curr. Opin. Neurobiol.* 21, 151–159 (2011).
24. Chou, V. T., Johnson, S. A. & Van Vactor, D. Synapse development and maturation at the *drosophila* neuromuscular junction. *Neural Dev.* 15, 11 (2020).
25. Lee-Hoeflich, S. T. et al. Activation of LIMK1 by binding to the BMP receptor, BMPRII, regulates BMP-dependent dendritogenesis. *EMBO J.* 23, 4792–4801 (2004).
26. Stanley, C. E., Mauss, A. S., Borst, A. & Cooper, R. L. The Effects of Chloride Flux on *Drosophila* Heart Rate. *Methods Protoc.* 2, (2019).
27. Davis, G. W. Homeostatic signaling and the stabilization of neural function. *Neuron* 80, 718–728 (2013).
28. Han, T. H., Dharkar, P., Mayer, M. L. & Serpe, M. Functional reconstitution of *Drosophila melanogaster* NMJ glutamate receptors. *Proc. Natl. Acad. Sci.* 112, 6182–6187 (2015).
29. Pannetta, G., Hiesinger, P. R., Fabian-fine, R., Meinertzhagen, I. A. & Bellen, H. J. *Drosophila* VAP-33A Directs Bouton Formation at Neuromuscular Junctions in a Dosage-Dependent Manner. 35, 291–306 (2002).
30. Tvorogova, A. V & Vorobjev, I. A. Microtubules suppress blebbing and stimulate lamella extension in spreading fibroblasts. *Cell tissue biol.* 7, 43–53 (2013).

31. Tvorogova, A., Saidova, A., Smirnova, T. & Vorobjev, I. Dynamic microtubules drive fibroblast spreading. *Biol. Open* 7, (2018).
32. Chen, A. et al. Inhibition of polar actin assembly by astral microtubules is required for cytokinesis. *Nat. Commun.* 12, 2409 (2021).
33. Chikina, A. S., Svitkina, T. M. & Alexandrova, A. Y. Time-resolved ultrastructure of the cortical actin cytoskeleton in dynamic membrane blebs. *J. Cell Biol.* 218, 445–454 (2019).
34. Kelley, C. A. et al. FLN-1/filamin is required to anchor the actomyosin cytoskeleton and for global organization of sub-cellular organelles in a contractile tissue. *Cytoskeleton (Hoboken)*. 77, 379–398 (2020).
35. Mikhaylova, M., Rentsch, J. & Ewers, H. Actomyosin Contractility in the Generation and Plasticity of Axons and Dendritic Spines. *Cells* 9, (2020).
36. Xu, K., Zhong, G. & Zhuang, X. Actin, spectrin, and associated proteins form a periodic cytoskeletal structure in axons. *Science* 339, 452–456 (2013).
37. Vassilopoulos, S., Gibaud, S., Jimenez, A., Caillol, G. & Leterrier, C. Ultrastructure of the axonal periodic scaffold reveals a braid-like organization of actin rings. *Nat. Commun.* 10, 5803 (2019).
38. He, J. et al. Prevalent presence of periodic actin–spectrin-based membrane skeleton in a broad range of neuronal cell types and animal species. *Proc. Natl. Acad. Sci.* 113, 6029–6034 (2016).
39. D’Este, E. et al. Subcortical cytoskeleton periodicity throughout the nervous system. *Sci. Rep.* 6, 22741 (2016).
40. Bär, J., Kobler, O., van Bommel, B. & Mikhaylova, M. Periodic F-actin structures shape the neck of dendritic spines. *Sci. Rep.* 6, 37136 (2016).
41. Sidenstein, S. C. et al. Multicolour Multilevel STED nanoscopy of Actin/Spectrin Organization at Synapses. *Sci. Rep.* 6, 26725 (2016).
42. Costa, A. R. et al. The membrane periodic skeleton is an actomyosin network that regulates axonal diameter and conduction. *Elife* 9, (2020).
43. Wang, T. et al. Radial contractility of actomyosin rings facilitates axonal trafficking and structural stability. *J. Cell Biol.* 219, (2020).
44. Rex, C. S. et al. Article Myosin IIb Regulates Actin Dynamics during Synaptic Plasticity and Memory Formation. *Neuron* 67, 603–617 (2010).
45. Li, X. et al. Synapse-specific and compartmentalized expression of presynaptic homeostatic potentiation. *Elife* 7, e34338 (2018).
46. Sulkowski, M., Kim, Y.-J. & Serpe, M. Postsynaptic glutamate receptors regulate local BMP signaling at the *Drosophila* neuromuscular junction. *Development* 141, 436–447 (2014).

REVIEWERS' COMMENTS

Reviewer #1 (Remarks to the Author):

The authors are to be commended for a very thorough and complete effort to address my criticisms and, in my opinion, also those of the other reviewers. All have been satisfactorily addressed. I support publication of the revised manuscript.

Reviewer #2 (Remarks to the Author):

The authors have addressed all my comments and questions. Therefore, I recommend the revised manuscript for publication in Nature Communications.

Reviewer #3 (Remarks to the Author):

We appreciate the authors' extensive efforts to improve the manuscript, which makes an important and novel contribution to the field. Most of the major concerns have been resolved. However, on a few important issues the authors were not entirely responsive to the questions posed, or chose to address different questions (detailed below, in the order of the points in our original review). We think they can all be addressed in writing or with minimal additional analysis or removal of existing data.

1. Heterogeneity of ghost boutons in live imaging: The authors have done a satisfactory job of describing the distribution of events in terms of their lifetimes (slow vs fast) and eventual size of the nascent bouton. However, it is still unclear what fraction of events occur from budding off an existing bouton, the axon, or splitting of an existing bouton. The "new bouton" in Fig 2b is just doubling in size of the existing bouton, so it doesn't fit in any of those categories. The only mention of this in the text is "primarily formed from pre-existing boutons", which is insufficient – this could mean 55% or 99%. Unbiased clustering is not necessary; a simple descriptive grouping is sufficient and absolutely necessary for the reader to interpret and possibly reproduce the results.

Relationship between live imaging and fixed imaging: We asked the authors to conduct two-color imaging to relate the Dlg-negative boutons in the fixed imaging to the boutons they are imaging live. They state in their rebuttal that "We will need to do more experiments with the Dlg Trap. We did several experiments using this line but, unfortunately, all boutons without Dlg formed during the stimulation and we never saw much happening after the stimulation period, which is when we are able to image our preparation." This is concerning, and suggests that the vast majority of boutons seen in the fixed imaging "GB" throughout the paper occur far before those visualized live after the stimulation period (as shown in Piccioli 2014) and measured in all the live imaging in this paper. This is a caveat of comparing the two datasets and should be disclosed loudly and clearly in the paper at the end of page 4.

Overall there is also an enormous amount of variability in control GB numbers from experiment to experiment, and between different control strains in a single experiment (eg The w¹¹¹⁸ flies seem very different in S1d vs S1f). It's important for the authors to comment on possible sources of this variability.

2. Diverse nature of synaptic actin structures: The authors did not address our major point, which is that comparing all the actin structures in the mother and nascent bouton is not meaningful because there are many different types of actin structures in boutons, only a subset of which is likely to be involved in blebbing and bleb growth (as described in these papers PMID 16914524, 34324418, 23593037 and bioRxiv 2022.05.18.492480, which the authors should cite as previous descriptions of synaptic actin structures). Instead they took a long journey in their response into endocytic actin structures in the bleb, which does not address the concern (Though they should note that Shi is distributed much more broadly at the synapse than predicted frequency of endocytic events, and has many functions in membrane remodeling and actin regulation. It is therefore not likely a good marker for "active endocytosis").

The authors also made no attempt to address our serious concern about the arbitrary nature of the line drawn for analysis "the selection of base in particular seem subjective. Is the base at the point of origination of the new bouton (as seems to be the case in 1e, though the "b" end of the line seems arbitrary), or is it the opposite side of the pre-existing bouton (e.g. Fig 1d)? For the 1 μ m x 1 μ m boxes used for quantification, it would make a huge difference if it were placed on a bright actin punctum or not." Figures noted in the quote above is from the original manuscript, but in current figure 2 the "base" is in random different places in 2b and 2c, and not even on a bouton in S8b.

Overall, I don't think it makes any difference for the conclusions of the paper, but I find the quantifications in Fig 2f and g to be not very meaningful (and also the split axis is likely not permitted). Unless the authors can explain that the base-edge measurements aren't arbitrarily affected by position and very bright actin structures (which may have nothing to do with bouton growth eg endocytic patches), these figures and the detailed supplements should be removed. The quantifications of MyoII are less concerning since that signal seems more uniform across the mother or daughter bouton.

3. Quantification of actin drug experiments. My concerns have been satisfied. It's a bit surprising that the Actin-5C and Lifeact showed similar turnover rates given that actin can only turn over by depolymerization and repolymerization, while lifeact can unbind and bind filaments, but perhaps this just suggests that turnover is very rapid. Fig S11fg figure and legend should note which actin reporter is being shown.

4. Connection between activity-dependent paradigm for structural plasticity and physiological synaptic growth: I am satisfied by the authors' explanations and additional experiments.

5. Clarification of data presentation and analysis: Other than the selection of "base" (see above), I am satisfied by the data presentation and clarification of analyses.

6. Relationship to the existing literature: The authors have satisfactorily adopted the previously used nomenclature to describe NMJ blebbing (aka "ghost boutons"), as suggested by us and the other reviewers. They improved the contextualization of their findings with the existing literature.

Minor points:

Table 4 still says "# repeats". If this is number of independent crosses, please just say that directly.

We recommend checking the supplemental figures and graph axes for typos (eg Parcial should be partial)

Abstract: "promoting bouton addition by increasing MNs confinement" replace with "which we hypothesize promotes bouton addition by increasing MN confinement."

Response to Reviewers

We are grateful to the reviewers for their contribution in improving our paper and are very enthusiastic about their consideration of the revised manuscript for publication in Nature Communications. Here, we have addressed the remaining points raised by Reviewer 3, which can be found below in our point-by-point response (reviewer points in black, our answer in blue). We also made the necessary alterations to the text and supplementary figure 2. We hope that Reviewer 3 finds the questions answered and considers our manuscript appropriate for publication.

Reviewer #3 (Remarks to the Author):

We appreciate the authors' extensive efforts to improve the manuscript, which makes an important and novel contribution to the field. Most of the major concerns have been resolved. However, on a few important issues the authors were not entirely responsive to the questions posed, or chose to address different questions (detailed below, in the order of the points in our original review). We think they can all be addressed in writing or with minimal additional analysis or removal of existing data.

1. **Heterogeneity of ghost boutons in live imaging:** The authors have done a satisfactory job of describing the distribution of events in terms of their lifetimes (slow vs fast) and eventual size of the nascent bouton. However, it is still unclear what fraction of events occur from budding off an existing bouton, the axon, or splitting of an existing bouton. The "new bouton" in Fig 2b is just doubling in size of the existing bouton, so it doesn't fit in any of those categories. The only mention of this in the text is "primarily formed from pre-existing boutons", which is insufficient – this could mean 55% or 99%. Unbiased clustering is not necessary; a simple descriptive grouping is sufficient and necessary for the reader to interpret and possibly reproduce the results.

We thank the reviewer for pointing this out in our revised manuscript. The reviewer is correct that we did not categorize the events observed in the live imaging experiments based on the type of morphological alterations, and we only quantified their time of formation. As requested, we now provide a description of the fraction of boutons forming from the axon or from a preexisting bouton, for both tags used to visualize bouton outgrowth in our movies (CD4-Tom and mCherry). The analysis is presented below and included in **Supplementary Figure 2a,b**.

Figure Legend: Characterization of types of bouton formation based on origin and type of morphological alteration. Left: analysis of movies from 3rd instar larvae expressing CD4-Tom under the control of the neuronal driver NSyb-Gal4. 96% of boutons emerge from pre-existing boutons, 4% from the axon. From the boutons emerging from other boutons, 70% give rise to more boutons, and 26% go through a transition phase where two boutons are visible but that resolve into one resulting in the increase in size of the mother bouton (elongation). Right: analysis of movies from 3rd instar larvae expressing mCherry under the control of the driver DV-Glut-LexA. 98% of boutons emerge from pre-existing boutons, 2% from the axon. From the boutons emerging from other boutons, 74% give rise to more boutons, and 24% go through a transition phase where two boutons are visible but that resolve into one resulting in the increase in size of the mother bouton (elongation). Note: percentages are displayed considering the total number of boutons 100%, with the total boutons being 46 for CD4-Tom and 43 for mCherry.

Overall, most “nascent boutons” develop from preexisting boutons (or “maternal boutons”). Within the boutons that form from pre-existing boutons, we subdivided in boutons that form by splitting or less frequently by elongation. What we considered as “boutons formed by splitting” corresponds to events that, by the end of our movies, we can clearly identify the “nascent bouton” dividing from the “maternal bouton” (example in **Figure 1a**). In contrast, “boutons formed by elongation” (example in **Figure 2b**) include events that started with the formation of a discrete bouton that could clearly be distinguished from the “mother bouton” at some point, but that throughout the movie resolved into a single bouton, which could no longer be distinguished from the mother bouton. In this category, we observed the formation of a bleb-like bouton that grows (when two boutons can be counted) and then retracts onto the original one, giving rise to the elongation/growth of the mother bouton. This category of boutons likely represents blebbing as a mechanism for bouton expansion. We included this type in the analysis because in the fixed samples, we do not know in which phase of the process each bouton is, and in the live imaging we cannot be 100% certain of the fate of each bouton if we were to image for longer periods. Furthermore, we find very interesting that blebbing with or without retraction (as described in other cell types) is used for remodeling either to allow net growth of each bleb or to increase the number of blebs/boutons. Not often, boutons elongated directly from the axon (example in **Figure 1b**), and in these cases, “nascent boutons” always exhibited slow and sustained growth. We have

provided this information in **Supplementary Figure 2** and reformulated the writing of the manuscript to include these observations and clarifying the methodology used for the quantifications.

Relationship between live imaging and fixed imaging: We asked the authors to conduct two-color imaging to relate the Dlg-negative boutons in the fixed imaging to the boutons they are imaging live. They state in their rebuttal that “We will need to do more experiments with the Dlg Trap. We did several experiments using this line but, unfortunately, all boutons without Dlg formed during the stimulation and we never saw much happening after the stimulation period, which is when we are able to image our preparation.” This is concerning, and suggests that the vast majority of boutons seen in the fixed imaging “GB” throughout the paper occur far before those visualized live after the stimulation period (as shown in Piccioli 2014) and measured in all the live imaging in this paper. This is a caveat of comparing the two datasets and should be disclosed loudly and clearly in the paper at the end of page 4.

We realize that we may have not been clear in our previous answer concerning this point, we hope we can explain it better now.

First, we want to clarify that our observations regarding bouton formation are similar to what was previously reported by Piccioli and Littleton (2014), which stated that “vigorous GB formation occurs during the K⁺-stim, with many of these boutons developing during K⁺ pulses”. The difference between Piccioli and Littleton protocol and ours is that after the 3rd pulse of high K⁺, the authors allow the larvae to rest in HL3 solution for 2 min before being stretched and fixed, while we established that we fix the larvae 30 min after stimulation. We do this because we often have several different conditions/genotypes running in parallel and wanted to always take the same time from stimulation to fixation, independently of the speed of stretching the larvae (and of the number of larvae at play during each experiment). Likewise, in Ataman et. al. (2008) experimental preparations remained in normal HL3 15 min after the stimulation protocol to complete dissection prior to immunohistochemical analysis.

Regarding the reviewer’s comment of “ This is concerning, and suggests that the vast majority of boutons seen in the fixed imaging “GB” throughout the paper occur far before those visualized live after the stimulation period (as shown in Piccioli 2014) and measured in all the live imaging in this paper” we will now explain our view on this issue:

It is true that many of the events occur during the stimulation period, which we hypothesize to be promoted by higher muscle contraction.

However, regarding the live imaging, although monitoring of NMJs throughout the duration of the stimulation protocol is possible, these movies show very strong movements, and their analyses were not viable. For this reason, we could only image our preparation in HL3.1 solution right after the

stimulation, which we often did for 30 min-1h. Even though live imaging was performed post-stimulation the frequency of bouton formation observed during this period in stimulated NMJs was significantly higher than unstimulated NMJs (we observed new events in ~30% of the larvae imaged after stimulation compared with ~5% of the larvae imaged without stimulation or at rest). This indicates that GBs still develop considerably beyond the stimulation period, which is also a reason why we always awaited 30 min post-stimulation before fixing our samples.

In the manuscript we did not compare the fixed and live imaging datasets, but we also did not explicitly write that many boutons form during the stimulation period. Therefore, we took the reviewer's suggestion and added a sentence to the main text where it reads "Even though many boutons form during the high K⁺ pulses as reported, live imaging of NMJs immediately after stimulation allowed us to identify two main types of bouton appearances". Additionally, in our movies most of the events of bouton outgrowth occurred during self-generated muscle contractions that suggests an analogous mechanism of bouton formation during or post- stimulation.

However, even if post-stimulation significant GB formation is still occurring, and often during muscle contractions, in our live imaging experiences we are only able to register a fraction of these events when compared to fixed samples, leaving open the possibility that they may not be representative of all types of bouton formation. Regarding the two-color imaging experiment asked by the reviewer to relate the Dlg-negative boutons identified in the fixed imaging to the boutons we imaged live, we performed this experiment with a different line because we already had a few movies with the Dlg Flyfos line (where there is an extra copy of Dlg) without detecting any events post-stimulation. Now, we chose to change the line to one that expresses a chimeric CD8-GFP-Shaker (C-terminal sequence of Shaker potassium channel) under the control of myosin heavy chain (MHC) promoter (Zito *et.*, 1999). This line has been previously used to label the postsynaptic densities via interaction of Shaker with postsynaptic scaffold Dlg. Additionally, it is well-established that MHC-CD8-GFP-Shaker recapitulates presynaptic arbor morphology during larval development, allowing visualization of bouton formation during this period (Zito *et.*, 1999). This time, we were able to observe new bouton formation (and also to observe that some GBs were already present when we started the imaging session) (examples are shown below).

As recommended by the reviewer, new boutons were identified using the presynaptic membrane, as performed in our movie analysis throughout the manuscript, blinded to postsynaptic Dlg, after which we checked for the presence of Dlg in these events. With this novel approach we observed that in most cases, boutons formed without any Dlg signal (12 boutons out of 14), which suggests that boutons we observed in fixed and live samples are comparable. Only on a few occasions (2 boutons out of 14), we found very small boutons stretching at the neuronal arbor which contained postsynaptic Dlg again suggestive that Dlg presence may not favor bouton formation. Overall, we have analyzed 18 larvae and observed new events in 5 larvae (~27%). In the figures below we show a summary of the data we observed. But this question will be interesting to follow in detail in future studies.

Example 1 – Fast bouton formation without Dlg

Example 2 – Slow bouton formation without Dlg

Example 3 – Bouton expansion followed by remodeling without Dlg

Example 4 – Another example of bouton expansion and remodeling without Dlg

Example 5 – Small bouton expanding with Dlg

Figures legend: Membrane was labeled with CD4-Tom under the control of NSyb-Gal4 (neuronal driver) and the postsynaptic density was visualized with MHC-CD8-GFP-Shaker (interacts and colocalizes with Dlg). Arrow indicates where bouton emerges or expands. Complete line circle signals “mother boutons”. Dotted line circle shows nascent or remodeling boutons. Scale bar is 2 μm .

Overall there is also an enormous amount of variability in control GB numbers from experiment to experiment, and between different control strains in a single experiment (eg The w1118 flies seem very different in S1d vs S1f). It's important for the authors to comment on possible sources of this variability.

We think that when the reviewer stated that w^{1118} flies were very different in S1d VS S1f the reviewer probably meant **Supplementary Figures 5d** and **5f** since current Supplementary Figure 1 contains only schematics regarding methodology. We will discuss below Supplementary Figures 5d and 5f.

These figures correspond to two independent experiments, where our objective was to have a notion of whether different tags interfered with bouton addition. In both Supplementary Figure 5d and 5f, the n is not very high, it is possible that the differences observed are due to variations in the intensity of response to stimulation. Supporting this, w^{1118} and the other genotypes in Supplementary Figure 5d all show a lower range of GB numbers, suggesting less efficient response to the stimulation, when comparing with controls and remaining genotypes in Supplementary Figure 5f. Despite this, if we analyze the results of the w^{1118} group shown in Supplementary Figure 5f as a 4th experimental group in Supplementary Figure 5d, it is not different from the other group with w^{1118} (by ANOVA), which suggests that the variability observed is within the considered normal values. Regarding possible

sources of variability in our experiments, while inter-individual factors can contribute that larvae can respond differently to the stimulation, external factors, such as temperature, slight changes in pH, can also cause significant changes in neuronal activity, and thereby influence the response to the stimulation protocols.

To try to minimize the sources of variation in our experiments, we always performed our experiments side by side with a control. Therefore, within each experience in Supplementary 5d and Supplementary 5f we compared controls and genotypes that were subjected to roughly to the same conditions and that replied to the stimulation. Importantly, in both cases, independently of lower or higher intensity of response to stimulation, GB numbers were not affected by overexpression of actin reporters, compared to WT control (w^{1118}).

2. Diverse nature of synaptic actin structures: The authors did not address our major point, which is that comparing all the actin structures in the mother and nascent bouton is not meaningful because there are many different types of actin structures in boutons, only a subset of which is likely to be involved in blebbing and bleb growth (as described in these papers PMID 16914524, 34324418, 23593037 and bioRxiv 2022.05.18.492480, which the authors should cite as previous descriptions of synaptic actin structures). Instead, they took a long journey in their response into endocytic actin structures in the bleb, which does not address the concern (Though they should note that Shi is distributed much more broadly at the synapse than predicted frequency of endocytic events, and has many functions in membrane remodeling and actin regulation. It is therefore not likely a good marker for “active endocytosis”).

We first want to thank the reviewer for directing us towards these papers. Having read this literature carefully - Nunes et. al. (2006); Del Signore et. al. (2021); Zhao et. al. (2013); Bingham et. al. (BioRxiv 2022) - we agree that we should introduce and discuss the possibility that the actin puncta we observe can represent places of endocytosis. We added this information to the manuscript, pg 7. Altogether, these papers established an important role for actin in the regulation of endocytosis in synaptic boutons. Of more direct relevance to our work, in Del Signore et. al. (2021) the authors showed that dynamic actin patches partially colocalize with clc, suggesting that a fraction of actin puncta at the *Drosophila* NMJ are endocytic spots. In the manuscript Bingham et. al. (BioRxiv 2022) the authors show beautiful super-resolution microscopy images where three types of presynaptic actin structures were identified: a branched actin mesh at the active zone that regulates access to the plasma membrane for exocytosis; linear actin rails which help vesicles move between the exocytosis, endocytosis zones and intracellular pools; peri-synaptic actin corrals that function as scaffolds for the vesicular pool and could also help at endocytic zones. Furthermore, a study by Gong et al (2018), reported that endocytosis can regulate the localization of factors required for the control of the place where membrane blebbing will occur. Altogether, we think that it is in fact a possibility to be considered that the actin puncta we observe in ~26% of bouton formation are places of endocytosis. However, since not all actin patches are endocytic, they can also represent exocytic places akin to what was

suggested by Piccioli and Littleton (2014). This latter scenario is also corroborated by our data with MyoII, where we observe puncta at the base of nascent boutons in similar % compared with actin (**Supplementary Figures 9I and 18I**), and in agreement with what was reported by Piccioli et. al. (2014). Whether these puncta represent endo or exocytic places (or both) is still unclear at this point and will be investigated in future studies. Still, we want to emphasize that we accept the criticism, which we now discuss in the context of our results and find quite interesting.

The authors also made no attempt to address our serious concern about the arbitrary nature of the line drawn for analysis “the selection of base in particular seem subjective. Is the base at the point of origination of the new bouton (as seems to be the case in 1e, though the “b” end of the line seems arbitrary), or is it the opposite side of the pre-existing bouton (e.g. Fig 1d)? For the 1 μ m x 1 μ m boxes used for quantification, it would make a huge difference if it were placed on a bright actin punctum or not.” Figures noted in the quote above is from the original manuscript, but in current figure 2 the “base” is in random different places in 2b and 2c, and not even on a bouton in S8b. Overall, I don’t think it makes any difference for the conclusions of the paper, but I find the quantifications in Fig 2f and g to be not very meaningful (and also the split axis is likely not permitted). Unless the authors can explain that the base-edge measurements aren’t arbitrarily affected by position and very bright actin structures (which may have nothing to do with bouton growth eg endocytic patches), these figures and the detailed supplements should be removed. The quantifications of MyoII are less concerning since that signal seems more uniform across the mother or daughter bouton.

First, we want to explain that our intention with the quantifications of actin was to ask whether the nascent bouton had high or low levels of actin, since this is a hallmark of blebs. To do so, we thought that quantifying the levels of actin in the growing bouton in comparison with the level of actin at the place from which it emerged was the best option. Although we do recognize that the actin structures observed at the NMJ may not be all related to bouton formation, generally actin found in “mother boutons” was noticeably more intense when compared to actin in “nascent boutons”, which was the point we wanted to convey. We have re-written the manuscript so that this is clearly defined and explained in detail how the quantifications were performed (in the methods and by adding a panel in Figure 2).

On the other hand, we also want to clarify our methodology, which is related with the lines used to build the kymographs versus the boxes used to quantify actin or MyoII fluorescence intensity at the base or edge of new boutons (which are not the same).

Some confusion probably arose because we represented the lines used for kymographs in the time-lapse images of bouton formation with letter b (base) and e (edge) to represent the starting and ending points of these lines. Importantly, there is not a direct correspondence between the lines for kymographs generation and the boxes used to quantify intensity. To avoid misinterpretation, we have

now changed the way kymograph lines are indicated and changed their designation. We apologize that this was not made understandable before. Moreover, we better describe here (see below) and in the methods section the difference between actin/MyoII kymographs and actin/MyoII intensity measurements.

Actin or MyoII Kymographs

The kymographs were based on our movies and were built to visualize the motion or recruitment of actin or MyoII into boutons during a selected Δt - while these boutons were growing or remodeling. To create these graphs, we drew a line (ROI) that extended from where the bouton was emerging and crossed the bouton to the end of the growth (or that spanned a remodeling bouton) for the chosen Δt . We have re-written this part in the methods to make this clear and to provide a more detailed explanation of kymograph generation.

Actin or MyoII fluorescence intensity

We used maximum intensity projections from Z-stacks acquired in our movies to probe fluorescence intensity of actin and MyoII at the base and edge of growing boutons. For each event, the intensity (integrated density) was measured in a $1 \times 1 \mu\text{m}$ square ROI placed at the base (closest point of origin of the bouton) and in the edge of the new bouton (farthest point parallel to the base). Base-edge pairs were measured in the beginning of bouton's expansion (the frame when the bouton started to emerge) and after expansion ceased (maximum size frame). We now better explain this in the methods. Importantly, analysis base-edge measurements were not randomly determined and are comparable across experiments.

Regarding the bright actin or MyoII puncta, when present at the base of new boutons: these structures were included in the analysis since we hypothesize that they may participate in bleb formation. However, to acknowledge the reviewer's concern regarding the significance of these quantifications: "Unless the authors can explain that the base-edge measurements aren't arbitrarily affected by position and very bright actin structures (which may have nothing to do with bouton growth eg endocytic patches), these figures and the detailed supplements should be removed. The quantifications of MyoII are less concerning since that signal seems more uniform across the mother or daughter bouton.", we now also performed the same analysis excluding base-edge values where actin puncta were observed which did not change our results (see below). For the reasons reported above related with bleb biology, we think that this data is valid and should be maintained in the paper as it is important to our message that boutons form with lower actin content, like blebs do (or else we would have to say that actin was low without any sort of quantification). In any case, we added a sentence to the text explicitly stating that 1) the puncta were included in the quantifications when present, 2) that we favor a model where these puncta represent places of high actin/MyoII contractility that can contribute for outgrowth, but that they can represent alternative actin structures involved in different processes, and that they can equally represent places of active endocytosis.

Lastly, we previously presented these graphs with YY-axis segmented into discrete intervals just to allow better range visualization of the data and was not of our knowledge that split axis was not permitted. Hence, we apologize for that and now display the graphs with normal axis.

Figure Legend: Quantification of actin intensity in new boutons, excluding the cases where a punctum of actin was present at the base of the nascent bouton.

3. Quantification of actin drug experiments. My concerns have been satisfied. It's a bit surprising that the Actin-5C and Lifeact showed similar turnover rates given that actin can only turn over by depolymerization and repolymerization, while Lifeact can unbind and bind filaments, but perhaps this just suggests that turnover is very rapid. Fig S11fg figure and legend should note which actin reporter is being shown.

We thank the reviewer for pointing this out. The actin reporter used in Supplementary Figures 11f and g was Lifeact-GFP. We have now indicated this in the figure legend and agree with the interpretation. This had been reported previously (Gokhale et. al., 2016), but we also had the same question and hence repeated the experiment with Actin-5C.

4. Connection between activity-dependent paradigm for structural plasticity and physiological synaptic growth: I am satisfied by the authors' explanations and additional experiments.

5. Clarification of data presentation and analysis: Other than the selection of "base" (see above), I am satisfied by the data presentation and clarification of analyses.

6. Relationship to the existing literature: The authors have satisfactorily adopted the previously used nomenclature to describe NMJ blebbing (aka "ghost boutons"), as suggested by us and the other reviewers. They improved the contextualization of their findings with the existing literature.

Points 4-6: We thank again for the reviewer feedback and hope that we have now answered the remaining questions/concerns raised.

Minor points:

Table 4 still says "# repeats". If this is number of independent crosses, please just say that directly. We recommend checking the supplemental figures and graph axes for typos (eg Parcial should be partial)

Abstract: "promoting bouton addition by increasing MNs confinement" replace with "which we hypothesize promotes bouton addition by increasing MN confinement."

We thank the reviewer for indicating these points. We have performed all the corrections necessary, and changed the n of each experiment to the figure legends, as requested by the Editorial staff of Nature Communications, providing all the details.

We hope we answered your questions and that you find our paper suitable for publication.

References:

- Ataman B, Ashley J, Gorczyca M, Ramachandran P, Fouquet W, Sigrist SJ, Budnik V. Rapid activity-dependent modifications in synaptic structure and function require bidirectional Wnt signaling. *Neuron*. 2008 Mar 13;57(5):705-18. doi: 10.1016/j.neuron.2008.01.026. PMID: 18341991; PMCID: PMC2435264.
- Bingham, D. et al. Distinct nano-structures support a multifunctional role of actin at presynapses. *bioRxiv* 2022.05.18.492480 (2022). doi:10.1101/2022.05.18.492480.
- Del Signore SJ, Kelley CF, Messelaar EM, Lemos T, Marchan MF, Ermanoska B, Mund M, Fai TG, Kaksonen M, Rodal AA. An autoinhibitory clamp of actin assembly constrains and directs synaptic endocytosis. *Elife*. 2021 Jul 29;10:e69597. doi: 10.7554/eLife.69597. PMID: 34324418; PMCID: PMC8321554.
- Gokhale, A. *et al.* The Proteome of BLOC-1 Genetic Defects Identifies the Arp2/3 Actin Polymerization Complex to Function Downstream of the Schizophrenia Susceptibility Factor Dysbindin at the Synapse. *J. Neurosci.* **36**, 12393–12411 (2016).
- Gong, X., Didan, Y., Lock, J. G. & Strömblad, S. KIF13A-regulated RhoB plasma membrane localization governs membrane blebbing and blebby amoeboid cell migration. *EMBO J.* **37**, (2018).
- Nunes P, Haines N, Kuppaswamy V, Fleet DJ, Stewart BA. Synaptic vesicle mobility and presynaptic F-actin are disrupted in a N-ethylmaleimide-sensitive factor allele of *Drosophila*. *Mol Biol Cell*. 2006 Nov;17(11):4709-19. doi: 10.1091/mbc.e06-03-0253. Epub 2006 Aug 16. PMID: 16914524; PMCID: PMC1635382.
- Piccioli ZD, Littleton JT. Retrograde BMP signaling modulates rapid activity-dependent synaptic growth via presynaptic LIM kinase regulation of cofilin. *J Neurosci*. 2014 Mar 19;34(12):4371-81. doi: 10.1523/JNEUROSCI.4943-13.2014. PMID: 24647957; PMCID: PMC3960475.
- Zhao L, Wang D, Wang Q, Rodal AA, Zhang YQ. *Drosophila* cyfip regulates synaptic development and endocytosis by suppressing filamentous actin assembly. *PLoS Genet*. 2013 Apr;9(4):e1003450. doi: 10.1371/journal.pgen.1003450. Epub 2013 Apr 4. PMID: 23593037; PMCID: PMC3616907.
- Zito K, Parnas D, Fetter RD, Isacoff EY, Goodman CS. Watching a synapse grow: noninvasive confocal imaging of synaptic growth in *Drosophila*. *Neuron*. 1999 Apr;22(4):719-29. doi: 10.1016/s0896-6273(00)80731-x. PMID: 10230792.